# Disentangling molecular mechanisms regulating sensitization of interferon alpha signal transduction

Frédérique Kok[1,2,†] (iD), Marcus Rosenblatt[3,4,†], Melissa Teusel[1,2,†], Tamar Nizharadze[1,2] (iD), Vladimir Gonçalves Magalhães[5] (iD), Christopher Dächert[2,5], Tim Maiwald[3], Artyom Vlasov[1,2], Marvin Wäsch[1], Silvana Tyufekchieva[6], Katrin Hoffmann[6], Georg Damm[7], Daniel Seehofer[7], Tobias Boettler[8] (iD), Marco Binder[5], Jens Timmer[3,4,9,10,*,‡] (iD), Marcel Schilling[1,**,‡] (iD) & Ursula Klingmüller[1,***,‡] (iD)

## Abstract

Tightly interlinked feedback regulators control the dynamics of intracellular responses elicited by the activation of signal transduction pathways. Interferon alpha (IFNα) orchestrates antiviral responses in hepatocytes, yet mechanisms that define pathway sensitization in response to prestimulation with different IFNα doses remained unresolved. We establish, based on quantitative measurements obtained for the hepatoma cell line Huh7.5, an ordinary differential equation model for IFNα signal transduction that comprises the feedback regulators STAT1, STAT2, IRF9, USP18, SOCS1, SOCS3, and IRF2. The model-based analysis shows that, mediated by the signaling proteins STAT2 and IRF9, prestimulation with a low IFNα dose hypersensitizes the pathway. In contrast, prestimulation with a high dose of IFNα leads to a dose-dependent desensitization, mediated by the negative regulators USP18 and SOCS1 that act at the receptor. The analysis of basal protein abundance in primary human hepatocytes reveals high heterogeneity in patient-specific amounts of STAT1, STAT2, IRF9, and USP18. The mathematical modeling approach shows that the basal amount of USP18 determines patient-specific pathway desensitization, while the abundance of STAT2 predicts the patient-specific IFNα signal response.

**Keywords** dynamic pathway modeling; feedback control; interferon; personalized treatment; signal transduction

**Subject Categories** Computational Biology; Immunology; Signal Transduction

Mol Syst Biol. (2020) 16: e8955

## Introduction

Cells rely on the transient activation of signal transduction pathways to rapidly adapt to changes in the environment. Molecular mechanisms that enable strong, but transient activation of signal transduction include the induction of positive regulators, such as the activation or induction of transcriptional co-regulators, and of negative feedback mechanisms, such as transcriptional induction of genes encoding negative feedback proteins or the post-translational activation of negative feedback regulators.

The innate immune response is the first line of defense against pathogens. Upon infection, pathogens are rapidly detected, followed by the activation of signal transduction pathways that stimulate the production of interferons. Interferons are cytokines with antiviral

1 Division Systems Biology of Signal Transduction, German Cancer Research Center (DKFZ), Heidelberg, Germany
2 Faculty of Biosciences, Heidelberg University, Heidelberg, Germany
3 Institute of Physics, University of Freiburg, Freiburg, Germany
4 FDM - Freiburg Center for Data Analysis and Modeling, University of Freiburg, Freiburg, Germany
5 Research Group "Dynamics of Early Viral Infection and the Innate Antiviral Response", Division Virus-Associated Carcinogenesis, German Cancer Research Center (DKFZ), Heidelberg, Germany
6 Department of General, Visceral and Transplantation Surgery, Ruprecht Karls University Heidelberg, Heidelberg, Germany
7 Department of Hepatobiliary Surgery and Visceral Transplantation, University of Leipzig, Leipzig, Germany
8 Department of Medicine II, University Hospital Freiburg—Faculty of Medicine, University of Freiburg, Freiburg, Germany
9 Signalling Research Centres BIOSS and CIBSS, University of Freiburg, Freiburg, Germany
10 Center for Biological Systems Analysis (ZBSA), University of Freiburg, Freiburg, Germany
*Corresponding author. Tel: +49 761 203 5829; Fax: +49 761 203 8541; E-mail: jeti@fdm.uni-freiburg.de
**Corresponding author. Tel: +49 6221 42 4485; Fax: +49 6221 42 4488; E-mail: m.schilling@dkfz.de
***Corresponding author. Tel: +49 6221 42 4481; Fax: +49 6221 42 4488; E-mail: u.klingmueller@dkfz.de
†These authors contributed equally to this work
‡These authors contributed equally to this work as last authors

and immunomodulatory effects (Tang *et al*, 2018). Due to these immunomodulatory functions, interferon pathway activation has recently been discussed to enhance the impact of checkpoint inhibitors in immunotherapies (Wang *et al*, 2019). Type I interferons are secreted and act via autocrine and paracrine induction of signal transduction. Thereby, an antiviral state is established both in infected and uninfected cells. In addition to antiviral genes acting directly on viruses, many of the induced genes are involved in pathogen recognition and the production of interferons as well as having immunomodulatory functions, giving rise to a highly dynamical system that consists of multiple waves of interferon production and release (Marie *et al*, 1998; Sato *et al*, 1998).

IFNα belongs to the type I interferons and signals via the interferon alpha receptors 1 and 2 (IFNAR1 and IFNAR2) (Novick *et al*, 1994; Domanski *et al*, 1995). Upon binding of IFNα, IFNAR1 and IFNAR2 dimerize, leading to the activation of the associated Janus kinases JAK1 and TYK2 and tyrosine phosphorylation of the receptors. Subsequently, the latent transcription factors signal transducer and activator of transcription 1 (STAT1) and STAT2 (Platanias *et al*, 1994) are recruited to the phosphorylated receptors and are activated by tyrosine phosphorylation. Phosphorylated STAT1 can form homodimers (Decker *et al*, 1991), STAT1:STAT2 heterodimers by binding to phosphorylated STAT2 (Li *et al*, 1996) or, by binding to interferon response factor 9 (IRF9) and phosphorylated STAT2, can form the interferon-stimulated gene factor 3 (ISGF3), which translocate to the nucleus and induce expression of antiviral, immunomodulatory, and feedback genes (Schindler *et al*, 1992). Positive feedback proteins include IRF9, STAT1, and STAT2 (Lehtonen *et al*, 1997), and negative feedback proteins include SOCS1, SOCS3, USP18, and IRF2 (Harada *et al*, 1989; Song & Shuai, 1998; Malakhova *et al*, 2006).

The extent of the overall response depends on the balance of the induced feedback mechanisms. Pre-exposure to a ligand can result in three scenarios: (i) desensitization of the pathway defined as lower activation upon stimulation, (ii) the same activation upon stimulation as without prestimulation, or (iii) hypersensitization of the pathway defined as higher activation of the pathway upon stimulation.

Recombinant IFNα has been used as treatment against chronic viral infections such as infection with the hepatitis B virus (HBV) and as an anti-tumor drug (Friedman, 2008). However, it was observed that many patients do not respond to the therapy (Suk-Fong Lok, 2019). Non-responsiveness was correlated with pre-activation of the endogenous IFNα signal transduction pathway (Chen *et al*, 2005) showing elevated levels of ISGs in liver biopsies of patients with chronic HCV infection (Sarasin-Filipowicz *et al*, 2008) or in hepatocytes isolated from patients with chronic HBV infection (Zhu *et al*, 2012). This desensitization of the pathway by pre-activation of the IFNα signal transduction pathway, also called refractoriness, was confirmed both in cell culture and *in vivo* experiments in mice (Larner *et al*, 1986; Makowska *et al*, 2011). USP18 was proposed as a factor contributing to pathway desensitization (Sarasin-Filipowicz *et al*, 2009).

Despite the reported evidence for an impact of pre-activation of IFNα signal transduction on the responsiveness of the IFNα signal transduction pathway, the specific conditions that result in pathway desensitization remained unclear. Further, it has not yet been explored whether also hypersensitization of the pathway might be possible. To unravel the molecular mechanisms that determine how pre-activation of the IFNα signal transduction pathway impacts the response to further ligand exposure, a mathematical model of the IFNα signal transduction pathway was established that comprises multiple feedback loops. The model was calibrated with quantitative time-resolved measurements of pathway components and target genes for different IFNα dose combinations using Huh7.5 and HepG2-hNTCP cell lines and primary human hepatocytes as cellular model systems. With this approach, we showed that while prestimulation with a high dose of IFNα results in desensitization of the signal transduction pathway, prestimulation with a low dose of IFNα can hypersensitize the pathway. Model simulations and experimental evidence revealed that not only USP18 but also SOCS1 are required for pathway desensitization, while induction of IRF9 and STAT2 contributes to pathway hypersensitization and the basal amount of IRF9 controls the dynamics of the ISGF3 transcription factor complex formation. Analysis of primary human hepatocytes from different donors identified patient-to-patient variability of basal USP18 levels as the key determinant controlling the patient-specific pathway desensitization threshold. Mathematical model simulations exploring a virtual patient cohort demonstrated that the abundance of STAT2 determines the patient-specific extent of the antiviral response.

# Results

### Prestimulation with a low IFNα dose hypersensitizes the pathway and prestimulation with a high IFNα dose desensitizes the pathway

To examine the impact of prestimulation with IFNα on the dynamics of IFNα signal transduction, we first established IFNα concentrations that resulted in low, intermediate, and high pathway activation. We exposed growth factor-depleted Huh7.5 cells for 1 h to different concentrations of IFNα and monitored by quantitative immunoblotting the amount of tyrosine-phosphorylated STAT1 and STAT2 ($pSTAT1_{Cyt}$, $pSTAT2_{Cyt}$) in cytoplasmic lysates (Fig 1A). At a dose of 2.8 pM IFNα (equal to 10 international units (IU)), 50% of maximal STAT1 phosphorylation and 10% of maximal STAT2 phosphorylation were reached, at 28 pM IFNα (equal to 100 IU) 90% of maximal STAT1 phosphorylation and 80% of maximal STAT2 phosphorylation were achieved, and at 1,400 pM IFNα (equal to 5000 IU) 100% of maximal STAT1 phosphorylation and 100% of maximal STAT2 phosphorylation were observed. For subsequent experiments, we selected 2.8 pM IFNα (low), 28 pM IFNα (intermediate), and 1,400 pM IFNα (high) as prestimulation doses. Since STAT proteins translocate to the nucleus upon activation, we additionally measured pSTAT1 in nuclear lysates (Appendix Fig S1A) as well as in total cell lysates (Appendix Fig S1B), showing a comparable dose–response behavior. To ensure the linearity of detection in the enzymatic assays (chemiluminescence) employed for quantitative immunoblotting, we not only measured the abundance of pSTAT1 in total cellular lysates by chemiluminescence employing a CCD camera-based device (Appendix Fig S1B), but also by fluorescence using a near-infrared fluorescence scanner (Appendix Fig S1C). The comparison of the chemiluminescence-based quantifications with the fluorescence-based quantifications revealed a Pearson correlation coefficient of 0.99, showing a comparable detection range for both methods (Appendix Fig S1D). To assess the

impact of IFNα prestimulation on the dynamics of STAT1 and STAT2 phosphorylation, growth factor-depleted Huh7.5 cells were either left untreated or were prestimulated with 2.8 pM or 1,400 pM IFNα. After 24 h of prestimulation, the cells were stimulated with 1,400 pM IFNα and were lysed every 10 min for up to 1 h (Fig 1B). The dynamics of STAT1 and STAT2 tyrosine phosphorylation (pSTAT1, pSTAT2) in cytoplasmic (exemplified in Fig 1C) and nuclear extracts were examined by quantitative immunoblotting and subsequently quantified (Fig 1D). The quantitative analysis revealed that in cells without IFNα-prestimulation, addition of IFNα resulted in a sharp increase of both pSTAT1 and pSTAT2 reaching maximal levels 30 min after stimulation, but a higher fold change of pSTAT1 compared with pSTAT2 was observed, both in cytoplasm and nucleus. The levels of pSTAT1 and pSTAT2 remained elevated for the entire observation time. Surprisingly, we observed in cells prestimulated with the low IFNα dose that stimulation with 1,400 pM IFNα resulted at 30 min in elevated phosphorylation of primarily STAT2 in the nucleus compared to the phosphorylation levels observed in the nucleus of untreated cells. The observed increased phosphorylation levels were sustained for the entire observation period suggesting hypersensitization of the IFNα signal transduction pathway. On the contrary, in cells prestimulated with the high dose of IFNα, the stimulation with 1,400 pM IFNα did not elicit phosphorylation of STAT1 and STAT2 neither in the cytoplasm nor the nucleus, suggesting pathway desensitization (Fig 1D). To determine the impact of residual IFNα from the prestimulation, we removed the ligand after 20.5 h prestimulation by washing, reapplied the ligand after 3 h growth factor-depletion and monitored the dynamics of IFNα induced STAT1 and STAT2 phosphorylation by quantitative immunoblotting (Appendix Fig S2A). These experiments demonstrated that ligand removal after 20.5 h of prestimulation resulted in the same level of hyper- or desensitization of the IFNα pathway as achieved by 24 h of prestimulation without ligand removal. However, ligand removal after only 1 h of prestimulation prevented the establishment of altered sensitivity toward IFNα stimulation, which correlated with incapacity to induce expression of positive and negative feedback proteins during 1-h IFNα stimulation (Appendix Fig S2B).

To closer investigate the impact of prestimulation with a low or high IFNα dose on IFNα-induced gene expression, we selected twenty interferon-stimulated genes (ISGs) with different dynamics in gene expression based on a previously published microarray analysis performed with IFNα-stimulated Huh7.5 cells (Maiwald *et al*, 2010). We focused our in depth analysis shown in Fig 1E on three ISGs representing an early, an intermediate, and a late dynamics. As an ISG with early transient dynamics, we selected the C-X-C motif chemokine 10 (*CXCL10*). The interferon-induced GTP-binding protein MX1 (*MX1*) was selected for its intermediate transient dynamics and interferon alpha-inducible protein 6 (*IFI6*) as a gene with a late response. The mRNA expression of these three ISG in Huh7.5 was investigated by qRT-PCR analysis for a total observation time of 48 h comprising 24 h of prestimulation with the low or high dose of IFNα or no prestimulation and 24 h of stimulation with 1,400 pM IFNα. Prestimulation with 1,400 pM IFNα induced a strong activation of *CXCL10*, *MX1,* and *IFI6* during the first 24 h (Fig 1E). During the prestimulation phase, both *CXCL10* and *MX1* showed a peak of maximal mRNA expression at 8 h and a subsequent decline of mRNA expression either to basal levels for *CXCL10,*

or to 60% of maximal expression for *MX1,* whereas the mRNA expression of *IFI6* increased during the entire observation period. Stimulation with 1,400 pM IFNα after 24 h of prestimulation with the high dose of IFNα did not result in a further increase of *IFI6* mRNA expression, *MX1* mRNA expression levels were marginally elevated and the mRNA expression of *CXCL10* remained at basal levels. These results showed that the pathway desensitization observed at the signal transduction level established by prestimulation with the high IFNα dose propagates to the expression of target genes. These findings also held true for the early transcripts *SOCS3*, *IRF1*, *IFIT2*, *IRF2*, *SOCS1,* and *CXCL11* (Appendix Figs S3A and S4A), for the intermediate transcripts *ZNFX1*, *NMI*, *STAT2*, *TRIM21*, *STAT1*, *IFIT1*, *USP18,* and *EIF2AK2* (Appendix Figs S3B and S4B) as well as for the late transcripts *ISG15*, *IRF9,* and *IFITM3* (Appendix Figs S3C and S4C). Prestimulation with 2.8 pM IFNα induced lower gene expression compared to prestimulation with the high dose of IFNα (Fig 1E versus F). However, cells prestimulated for 24 h with the low dose of IFNα responded to stimulation with 1,400 pM IFNα and responded faster compared to cells that had not been prestimulated with IFNα albeit with lower maximal mRNA levels. For example for CXCL10 and MX1 maximal peaks of gene expression were already observed at 4 h after stimulation of cells prestimulated with 2.8 pM IFNα (Fig 1F), compared to maximal gene expression observed at 8 h after stimulation of cells without prestimulation (Fig 1E).

In summary, prestimulation with a low dose of IFNα resulted in hypersensitization of signal transduction and accelerated target gene expression, while prestimulation with a high dose of IFNα caused pathway desensitization and prevented the induction of target gene expression.

### Establishment of a mathematical model of IFNα-induced signal transduction and gene expression to unravel the mechanisms of IFNα dose-dependent pathway sensitization

To elucidate how prestimulation with a low dose of IFNα generates hypersensitization of signal transduction, while prestimulation with a high dose of IFNα results in pathway desensitization, we established an ordinary differential equation (ODE) model (Fig 2). Rate equations were derived from the law of mass-action according to chemical reaction network theory, including Michaelis–Menten kinetics. The ODE model incorporates IFNα-induced signal transduction starting with activation of the receptors IFNAR1 and IFNAR2, followed by the phosphorylation of STAT1 and STAT2, complex formation of the phosphorylated STAT proteins as well as their translocation to the nucleus and induction of feedback proteins. It integrates the prestimulation as well as the stimulation with different IFNα doses over time.

In brief, IFNAR1 and IFNAR2 in complex with JAK1 and TYK2 are summarized as one species termed Receptor (Rec). Upon binding of IFNα, the receptor becomes phosphorylated and therefore activated (aRecIFN). The activated receptor phosphorylates cytoplasmic STAT1 (STAT1c) or STAT2 (STAT2c). Upon phosphorylation, STAT1 homodimers (pSTAT1dimc) can be formed. Similarly, pSTAT1pSTAT2 heterodimers (pSTAT1pSTAT2c) can be formed upon phosphorylation of STAT1 and of STAT2. In the mathematical model, phosphorylation and dimerization of STAT1 and STAT2 were approximated by single reactions in which the active receptor

complex forms and induces dimer formation directly. Binding of IRF9c to pSTAT1pSTAT2c heterodimers results in the formation of ISGF3c. The three complexes pSTAT1dimc, pSTAT1pSTAT2c, and ISGF3c translocate to the nucleus and induce expression of target genes. In the nucleus, pSTAT1 homodimers (pSTAT1dimn) induce the expression of *SOCS3* mRNA by binding to STAT1 transcription factor binding sites called occupied gamma-activated sequence-binding sites (OccGASbs). The promoters of the genes encoding the

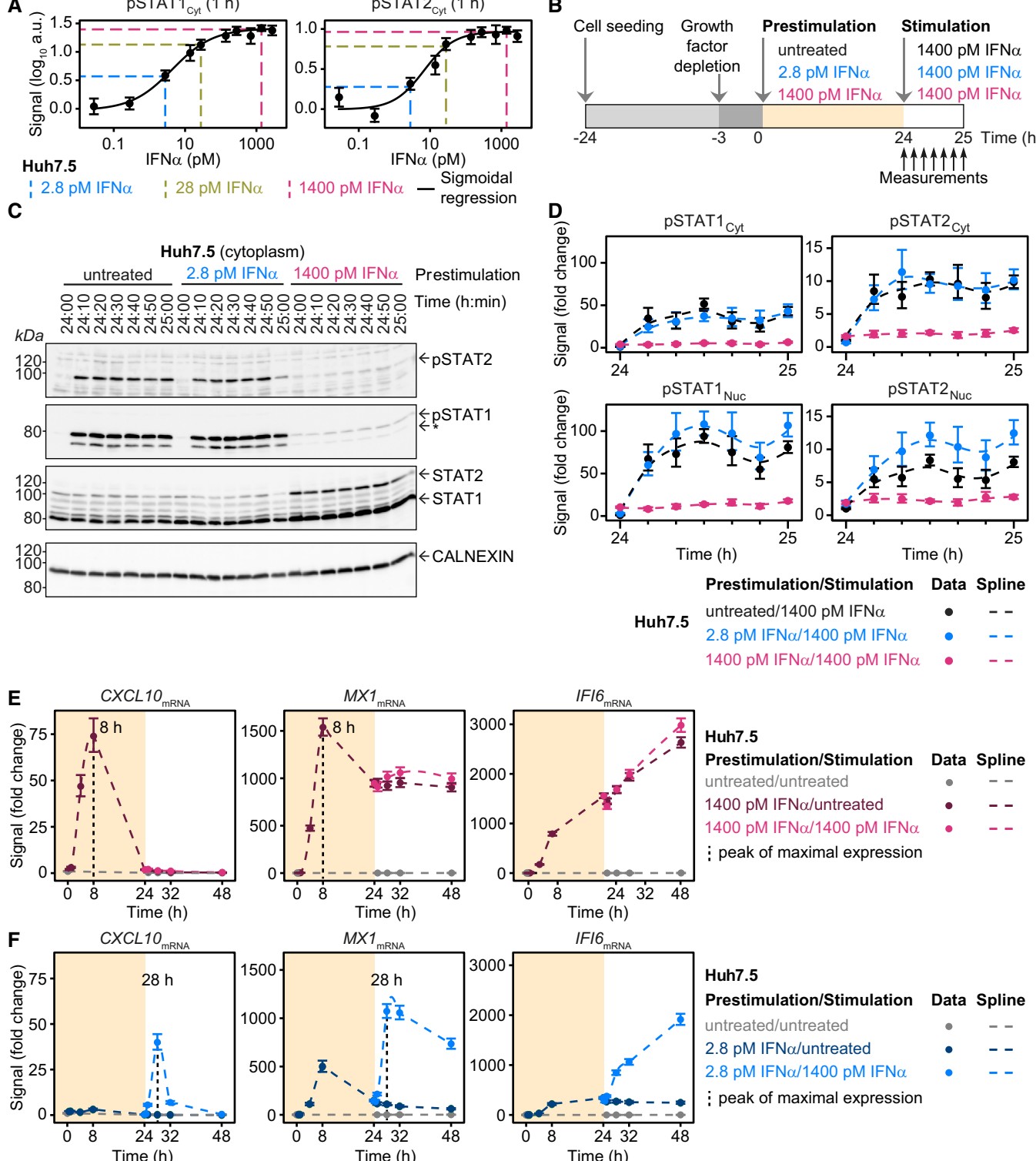

**Figure 1.**

**Figure 1. Dose-dependent sensitization of IFNα signal transduction.**

A IFNα dose dependency of STAT1 and STAT2 phosphorylation in Huh7.5 cells. Cells were seeded 24 h prior to the start of the experiment. Three hours before stimulation, cells were growth factor-depleted and were subsequently stimulated with the indicated concentrations of IFNα. Cytoplasmic protein lysates were collected 1 h after the stimulation and phosphorylation of STAT1 and STAT2 was detected by immunoblotting utilizing antibodies recognizing STAT1 phosphorylated on tyrosine residue 701, or STAT2 phosphorylated on tyrosine residue 690. Data points are displayed as dots with 1σ confidence interval estimated from biological replicates ($N = 1$ to $N = 38$) using a combined scaling and error model. Data are approximated with a sigmoidal function and signals corresponding to a low dose (2.8 pM IFNα), a medium dose (28 pM IFNα), and a high dose (1,400 pM IFNα) are displayed with dashed lines.

B Experimental design of IFNα sensitization experiment in Huh7.5. Cells were seeded 24 h prior to the start of the experiment. Three hours before prestimulation, cells were growth factor-depleted and were subsequently prestimulated with 2.8 pM IFNα, 1,400 pM IFNα, or were left untreated. After 24 h, cells were stimulated with 1,400 pM IFNα. Cytoplasmic and nuclear protein lysates were collected at indicated time points.

C Representative immunoblot of IFNα-induced phosphorylation of STAT1 and STAT2 upon stimulation of Huh7.5 cells prestimulated for 24 h with 2.8 pM IFNα, 1,400 pM IFNα or without prestimulation. Time points after prestimulation are indicated. 20 μg of cytoplasmic lysates were analyzed using antibodies for the indicated targets. Phosphorylation of STAT1 and STAT2 was detected by immunoblotting utilizing antibodies recognizing STAT1 phosphorylated on tyrosine residue 701, or STAT2 phosphorylated on tyrosine residue 690. An asterisk indicates pSTAT1β. Calnexin served as loading control. Molecular weights are indicated on the left. Immunoblot detection was performed with chemiluminescence employing a CCD camera-based device (ImageQuant).

D Quantification of immunoblots of IFNα-induced phosphorylation of cytoplasmic and nuclear STAT1 and STAT2 in Huh7.5 cells prestimulated with 2.8 pM IFNα, 1,400 pM IFNα or without pretreatment. Time points after prestimulation are displayed. Data are displayed as fold change relative to untreated cells. Errors were estimated with a combined scaling and error model, comprising 1σ confidence interval estimated from biological replicates ($N = 3$). Dashed lines indicate smoothing splines.

E Induction of interferon-stimulated genes upon prestimulation with 1,400 pM IFNα (yellow background) and stimulation with 1,400 pM IFNα (white background) in Huh7.5 cells, assessed by qRT-PCR. RNA levels were normalized to the geometric mean of reference genes GAPDH, HPRT, and TBP and were displayed as fold change. Peak of gene expression is indicated. Data points displayed as dots with 1σ confidence interval estimated from biological replicates ($N = 4$ to $N = 14$) using a combined scaling and error model. Dashed lines indicate smoothing splines.

F Induction of interferon-stimulated genes upon prestimulation with 2.8 pM IFNα (yellow background) and stimulation with 1,400 pM IFNα (white background) in Huh7.5 cells, assessed by qRT-PCR. RNA levels were normalized to the geometric mean of reference genes GAPDH, HPRT, and TBP and were displayed as fold change. Peak of gene expression is indicated. Data points are displayed as dots with 1σ confidence interval estimated from biological replicates ($N = 4$ to $N = 6$) using a combined scaling and error model. Dashed lines indicate smoothing splines.

positive feedback proteins IRF9, STAT1, and STAT2 as well as the negative feedback proteins USP18, SOCS1, and IRF2 harbor gamma interferon-activated sites (GAS) in combination with interferon-stimulated response elements (ISRE). Since pSTAT1:pSTAT2 heterodimers and ISGF3 bind to these combined GAS and ISRE sites, both, nuclear pSTAT1pSTAT2n and ISGF3n, contribute to the formation of occupied GAS- and ISRE-binding sites (OccGASbs + OccISREbs) in the promoters of these genes. By means of the model, the gene induction by ISGF3n was estimated to be stronger than by pSTAT1pSTAT2n, which is in agreement with literature showing that IRF9, STAT1, and STAT2 all contribute to binding to the ISRE (Qureshi *et al*, 1995). Inside the nucleus, all transcription factor complexes can dissociate into their individual components that can translocate back to the cytoplasm. The individual components STAT1, STAT2, and IRF9 have the freedom to shuttle between cytoplasm and nucleus (Meyer *et al*, 2002; Banninger & Reich, 2004). The induced feedback mRNAs are translated into proteins, taking gene-specific time delays for translation into account, which were incorporated via linear chains between mRNA and protein targets (MacDonald, 1976). For both the transcriptional and translational processes, gene-specific saturation levels were taken into account. Signal termination involves SOCS3, USP18, and SOCS1. USP18 binds to IFNAR2 and thereby inhibits downstream substrate phosphorylation (Malakhova *et al*, 2006), while SOCS proteins act at the receptor level directly inhibiting formation of the active receptor-IFN complex (aRecIFN) by inhibiting JAK family members (Chen *et al*, 2000). SOCS1 additionally mediates degradation of the activated receptor complexes (Piganis *et al*, 2011). Additionally, the transcriptional modulator IRF2 was incorporated to capture transient dynamics of SOCS1mRNA (Harada *et al*, 1989). Turnovers of all species include basal production and degradation.

To capture dynamic properties of the system, the IFNα-signal transduction model was calibrated with 1,918 data points generated under 25 experimental conditions, comprising quantitative time-resolved data obtained at the protein and the RNA level. Identifiability of model parameters was addressed by computing the profile likelihood (Raue *et al*, 2009) for each parameter (Appendix Fig S10). Out of 85 model parameters, 74 parameters were identifiable, i.e., finite confidence intervals were obtained (Appendix Table S3). From the remaining eleven parameters, three showed confidence intervals open to minus infinity and eight open to plus infinity. However, no further model reduction was applied due to the biological relevance of these parameters (see Materials and Methods).

We tested three additional mechanisms, (i) a cytoplasmic phosphatase dissociating pSTAT1dimc, pSTAT1pSTAT2c, and ISGF3c, (ii) STAT2 functioning as an adapter for USP18 (Arimoto *et al*, 2017), and (iii) pSTAT1dimn inducing OccGASbs+OccISREbs by formulating alternative mathematical models (Appendix Fig S5A). We re-estimated the model parameters for each of these three hypotheses and calculated the Bayesian information criterion (BIC). In all three cases, the goodness-of-fit was nearly the same, however, due to the additional parameters, the BIC was significantly worse and these additional mechanisms were rejected (Appendix Fig S5B).

In summary, we established an ODE model of IFNα signal transduction consisting of seven feedback proteins that could contribute to pathway sensitization to different extents.

## IFNα dose-dependent induction of feedback proteins and pathway sensitization

We hypothesized that sensitization of the IFNα signal transduction pathway is determined by the IFNα-induced feedbacks differing in their dose–response behavior as well as in their dynamics. To address this assumption, we performed time-resolved analysis of the protein levels of the known negative feedback regulator USP18, the positive feedback regulator IRF9 and additionally accounted for

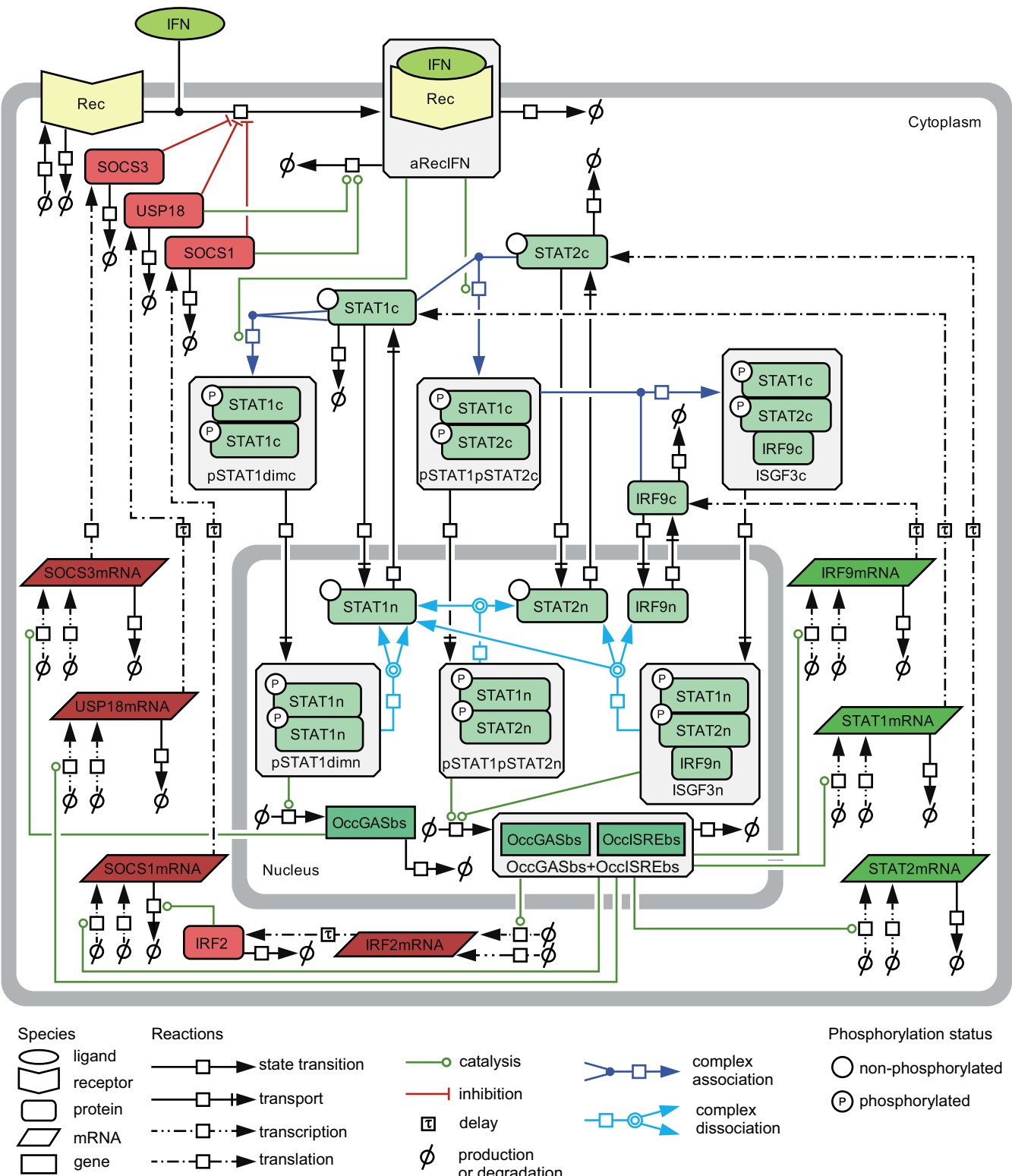

**Figure 2. Mathematical model structure of IFNα-induced JAK/STAT signal transduction pathway.**

The model structure is represented by a process diagram displayed according to Systems Biology Graphical Notation (Le Novere *et al*, 2009). Negative regulators are depicted in red. c: cytoplasm, n: nucleus, dim: dimer, rec: receptor, a: active, OccGASbs: occupied binding sites containing gamma-activated sequence, OccGASbs + OccISREbs: occupied binding sites containing gamma-activated sequence and interferon-stimulated response element.

the total protein amount of STAT1 and STAT2 (tSTAT1 and tSTAT2) that we assumed to act as additional positive regulators of the system in response to IFNα. To obtain a quantitative understanding of the influence of these signal transduction components on pathway activation, we in parallel performed time-resolved analysis of pSTAT1 and pSTAT2. Growth factor-depleted Huh7.5 cells were left untreated or were stimulated with different IFNα doses ranging from 2.8 to 2,800 pM (for untreated, 2.8, 28, and 1,400 pM IFNα see Figs 3A and EV1A; for 8.4, 280, and 2,800 pM IFNα see Fig EV1A). The changes of concentrations of signal transduction components were monitored by quantitative immunoblotting at different time points up to 32 h and were used to calibrate the mathematical model. For all investigated doses, both, data points and model trajectories, revealed transient phosphorylation of cytoplasmic and nuclear STAT1 (pSTAT1) and STAT2 (pSTAT2) that returned close to basal levels 8 h after stimulation. In addition, stimulation with 1,400 pM IFNα resulted in a dampened second peak of pSTAT1 and pSTAT2 around 12 h as visible in the data and captured by the model (Fig 3A). The model suggested that the second peak of pSTAT1 and pSTAT2 is already triggered by stimulation with IFNα doses above 2.8 pM (Fig EV1A). Interestingly, we observed in both model and data a different time-dependent and dose-dependent induction of the feedback proteins IRF9, USP18 as well as of the total STAT1 and STAT2 proteins. While saturation levels of IRF9 protein were already detected after 6 h of stimulation with 2.8 pM IFNα (Figs 3A and EV1A), the other feedback proteins showed a more graded IFNα dose-dependent increase in the maximal responses. USP18 protein reached maximal induction only upon stimulation with 140 pM IFNα for 24 h (Fig EV1B, USP18$_{Cyt}$) and reached a plateau at 8 h of stimulation (Figs 3A and EV1A), while total STAT1 and STAT2 protein levels reached a plateau only at 14–16 h after stimulation with IFNα doses of 140 pM or more (Figs 3A, and EV1A and B, tSTAT1$_{Cyt}$, tSTAT2$_{Cyt}$, tSTAT1$_{Nuc}$, tSTAT2$_{Nuc}$). Subsequently, the expression of feedback proteins remained sustained. Thus, we detected an IFNα dose-dependent increase of USP18, tSTAT1, and tSTAT2 at 24 h of stimulation. Maximal levels of IRF9 protein were reached at 8 h upon stimulation with as little as 2.8 pM IFNα and were maintained at 24 h. Further, after 4 h of IFNα stimulation, we detected an induction of IRF9 already in response to stimulation with 2.8 pM IFNα, whereas for STAT1, STAT2, and USP18 only minor increases were observed even for the highest IFNα doses tested (Fig EV1B). One hour of stimulation sufficed to induce pSTAT1 and pSTAT2 as well as tSTAT1$_{Nuc}$ and tSTAT2$_{Nuc}$, but not the other components.

To examine the impact of different IFNα doses on mRNA expression profiles, mRNA expression was monitored by qRT-PCR at different time points after stimulation with 2.8, 28, and 1,400 pM IFNα (Fig 3B) and the impact of prestimulation with IFNα was examined (Fig EV2A). Similar to the findings at the protein level, sustained expression profiles for *STAT1*, *STAT2*, *IRF9*, and *USP18* mRNAs were observed for 24 h of stimulation with IFNα. For *STAT1*, *STAT2*, and *USP18* mRNA, a gradual increase in mRNA expression in response to rising IFNα dose was detected, whereas for *IRF9* again mRNA expression levels close to saturation were already detected with as little as 2.8 pM IFNα (Fig 3B). On the other hand, a more transient expression dynamics was observed for *IRF2*, *SOCS1*, and *SOCS3* mRNA, with *SOCS1* and *IRF2* showing mRNA levels still above basal expression after 24 h of IFNα stimulation (Fig 3B).

Interestingly, transient high levels of *SOCS3* mRNA that returned within 8 h to basal mRNA levels were only observed in cells stimulated with 1,400 pM IFNα, whereas in cells stimulated with 2.8 or 28 pM IFNα only a minor induction of *SOCS3* mRNA was detectable. Further, IFNα stimulation of prestimulated cells only showed an induction of *STAT1*, *STAT2*, *USP18*, *SOCS1*, and *SOCS3* mRNA upon prestimulation with 2.8 pM IFNα, but not upon prestimulation with 1,400 pM IFNα (Fig EV2A). *IRF9* mRNA expression remained at maximal levels that was already induced by prestimulation with 2.8 pM IFNα. Likewise, at the protein level, prestimulation with as little as 2.8 pM resulted in maximal IRF9 levels that did not further increase upon stimulation with high IFNα concentrations in the observed time period of 8 h (Fig EV2B). For USP18, STAT1, and STAT2, prestimulation with IFNα doses higher than 28 pM for 24 h resulted in saturated levels that could not be further increased by stimulation with high IFNα doses. Altogether, the differences in timing and IFNα dose-dependent induction of the feedbacks on protein and mRNA level were accurately captured by our ODE model of IFNα signal transduction.

The results shown in Figs 3A and B, and EV1A, and B only display IFNα dose-dependent relative changes in the expression of feedback proteins. However, to dissect the specific contribution of each component, absolute values are essential. Therefore, the amount of STAT1, STAT2, IRF9, and USP18 molecules per cell was determined experimentally in growth factor-depleted Huh7.5 cells that were either left untreated or were stimulated with 2.8, 28, or 1,400 pM of IFNα for 24 h (Fig 3C). This quantitative analysis revealed that the feedback proteins had a different abundance and again showed a different dose dependency among each other. The amount of STAT1 protein molecules per cell ranged from 500,000 to 1,000,000 molecules under basal conditions and gradually increased in an IFNα dose-dependent manner. In contrast, the amount of STAT2 protein was only 50,000 molecules at basal level and thus one order of magnitude lower compared to STAT1. However, STAT2 protein levels showed a much higher fold change upon treatment with 1,400 pM IFNα. The feedback proteins IRF9 and USP18 were present at very low levels under basal conditions. Upon treatment with IFNα, IRF9 protein levels increased starting from treatment with 2.8 pM IFNα, while USP18 protein levels only showed a minor increase after stimulation with 2.8 and 28 pM IFNα. The protein abundances determined by the calibrated model were very well in line with the experimental data.

To evaluate whether our mathematical model is capable to capture hypersensitization and desensitization of the pathway, growth factor-depleted Huh7.5 cells were prestimulated with 2.8, 28, and 1,400 pM IFNα or left untreated and were subsequently stimulated with 1,400 pM IFNα at 24 h. As shown in Fig 3D, and for additional IFNα doses in Fig EV2B, the calibrated mathematical model was able to describe the experimental data. In summary, experimental data and model trajectories revealed that at IFNα concentrations of 28 pM and below, hypersensitization of STAT2 phosphorylation in the nucleus was observed in the first 2 h of the observation period. Only a very small hypersensitization effect was detected on nuclear STAT1 phosphorylation with 2.8 pM IFNα prestimulation after 1 h (Figs 3D and EV2B). A gradual increase in desensitization of STAT1 and STAT2 phosphorylation in the cytoplasm and nucleus was observed for prestimulation with IFNα concentrations higher than 28 pM (Fig EV2B).

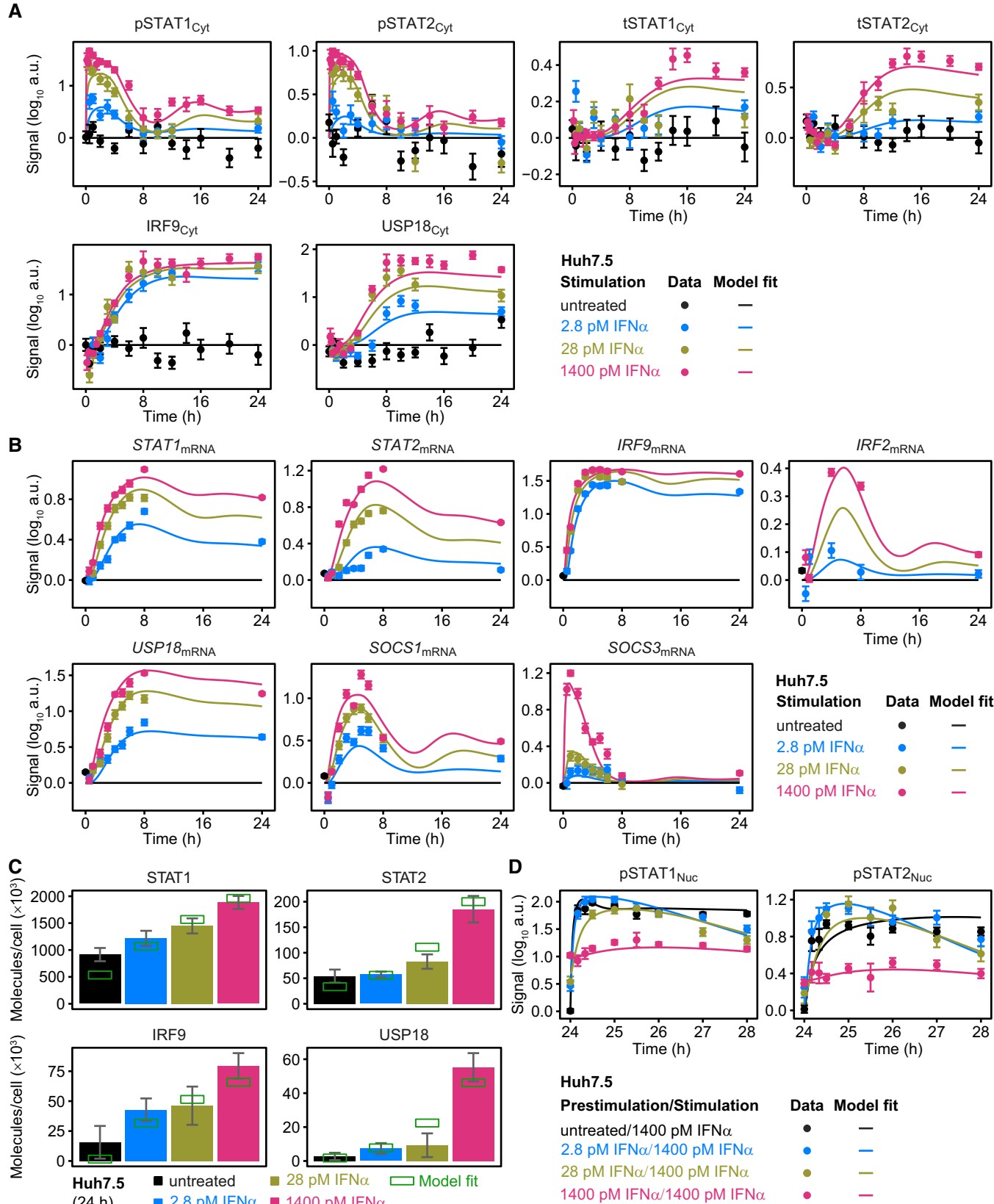

**Figure 3.**

◄

**Figure 3. Model calibration with IFNα-induced signal transduction in Huh7.5 upon prestimulation and stimulation with IFNα.**

Growth factor-depleted Huh7.5 cells were prestimulated with 2.8, 28, 1,400 pM IFNα or left untreated and were stimulated with 1,400 pM IFNα 24 h later. IFNα-induced signaling was measured by time-resolved quantitative immunoblotting and detected with chemiluminescence using a CCD camera-based device. Data were normalized to reference proteins Calnexin or HDAC1, scaled and subjected to model calibration.

A   Model calibration with time-resolved IFNα-induced phosphorylation of STAT1 and STAT2 and induced feedback proteins upon prestimulation with 0, 2.8, 28, or 1,400 pM IFNα. Cytoplasmic lysates were subjected to quantitative immunoblotting. Experimental data were represented by filled circles with errors representing 1σ confidence intervals estimated from biological replicates ($N = 3$ to $N = 23$) using a combined scaling and error model. Model trajectories are represented by lines. pSTAT1, pSTAT2 represent phosphorylated STAT1 and STAT2 on residue Tyr701 and Tyr690, respectively. tSTAT1 and tSTAT2 represent total form of STAT1 and STAT2 comprising both phosphorylated and unphosphorylated STAT1, STAT2, respectively.

B   Model calibration with time-resolved IFNα-induced feedback transcripts upon prestimulation with 0, 2.8, 28, or 1,400 pM IFNα, assessed by qRT-PCR. mRNA levels were normalized to the geometric mean of reference genes *GAPDH*, *HPRT*, and *TBP*. Experimental data are represented by filled circles with errors representing 1σ confidence intervals estimated from biological replicates ($N = 3$ to $N = 14$) using a combined scaling and error model. Model trajectories are represented by lines.

C   Model calibration with the amount of molecules per cell of STAT1, STAT2, IRF9, and USP18 determined 24 h after prestimulation with 0, 2.8, 28, or 1,400 pM IFNα. Calibrator proteins were spiked into 10 μg of total protein lysate and subjected to quantitative immunoblotting. Immunoblot detection was performed by chemiluminescence using a CCD camera-based device. Averaged values ($N = 4$) are displayed with standard deviations. Green squares indicate amounts estimated by the mathematical model.

D   Model calibration of IFNα-induced phosphorylation of STAT1 and STAT2 upon stimulation of Huh7.5 prestimulated with 0, 2.8, 28, or 1,400 pM IFNα. Nuclear lysates were subjected to quantitative immunoblotting. Experimental data are represented by filled circles with errors representing 1σ confidence interval estimated from biological replicates ($N = 4$ to $N = 22$) using a combined scaling and error model. Model trajectories are represented by lines.

Taken together, the mathematical model that was calibrated based on the experimental data revealed that the feedback components of IFNα signal transduction differ with respect to their IFNα dose dependencies and their induction dynamics. Prestimulation with doses below 28 pM IFNα resulted in hypersensitization of the pathway as indicated by elevated phosphorylation of STAT1 and STAT2, while IFNα doses higher than 28 pM established a gradual IFNα dose-dependent desensitization of the pathway. Our calibrated mathematical model was able (i) to simultaneously describe dose-dependent activation of the pathway, (ii) to characterize time-dependent induction of feedback components over a measurement period of 24 h, and (iii) to capture both dose-dependent hypersensitization and desensitization of the pathway.

**Model-based analysis of the dynamics of pSTAT1 complex formation and model validation**

We employed the mathematical model of IFNα-signal transduction to examine the impact of the IFNα prestimulation dose on the dynamics of the formation of the transcriptionally active pSTAT1- and pSTAT2-containing complexes in the nucleus upon stimulation with a high dose of IFNα. In principle pSTAT1 homodimers, pSTAT1:pSTAT2 heterodimers and pSTAT1:pSTAT2:IRF9 (ISGF3) trimers can form. The model-based analysis revealed that in cells without prior exposure to IFNα, total STAT1 levels are in excess compared to STAT2 and IRF9 and therefore initially primarily pSTAT1 homodimers are formed in the nucleus that bind to gamma interferon-activated site (GAS) elements in promoter regions (Fig 4A). The mathematical model indicated that simultaneously pSTAT1:pSTAT2 heterodimers are formed, albeit with slower dynamics. Finally, after 4 h, ISGF3 complexes are formed that bind to interferon-stimulated response element (ISRE) and become the dominant transcription factor complexes. This delay in formation of ISGF3 is caused by IFNα-induced IRF9 upregulation and coincides with the dynamics of the induction of IRF9 (Fig 3A). Interestingly, the model indicated that upon prestimulation with 28 pM IFNα, which increases the abundance of IRF9 (Fig 3C), stimulation with 1,400 pM IFNα resulted in an immediate rise in ISGF3 complexes as well as a much reduced formation of pSTAT1 homodimers. This effect is even more pronounced upon prestimulation with 280 pM

followed by stimulation with 1,400 pM IFNα, triggering primarily an immediate increase in ISGF3 complexes. Therefore, we hypothesized that the formation of pSTAT1 homodimers is reduced as a function of an increasing prestimulation dose. Consequently, the expression of GAS-controlled genes should be reduced upon stimulation with 1,400 pM IFNα in cells prestimulated with low to intermediate IFNα doses, because under these conditions primarily ISGF3 complexes are formed that bind to ISRE sequences.

To experimentally verify the model-predicted consecutive occurrence of the different transcription factor complexes, we performed electrophoretic mobility shift assays (EMSA) as previously reported (Forero *et al*, 2019). Experiments using a probe comprising the GAS-binding region of the *IRF1* promoter (Fig EV3A, left panel) showed that in mock prestimulated Huh7.5 cells an early DNA:protein complex is induced after 1 h (corresponding to 25 h after mock prestimulation) of stimulation with 1,400 pM IFNα. This DNA:protein complex is absent at 4 and 6 h post-IFNα stimulation of Huh7.5 cells (corresponding to 28 and 30 h after mock prestimulation, respectively). As shown in Fig EV3A, right panel, incubation of the lysate-DNA mixture with an antibody recognizing STAT1 led to a supershift, which was absent upon addition of antibodies detecting STAT2 or IRF9, confirming the specificity of the detected complex as pSTAT1 homodimer. In accordance with our assumption in the mathematical model, no major binding of the pSTAT1:pSTAT2 heterodimer to the GAS region was observed. On the contrary to non-prestimulated Huh7.5 cells, formation of pSTAT1 homodimeric complexes induced by stimulation with 1,400 pM IFNα is much reduced in cells prestimulated with 280 pM IFNα for 24 h. To quantitatively compare the obtained results with our model predictions, we predicted the dynamics of occupied GAS-binding sites induced by 1,400 pM IFNα in untreated Huh7.5 cells and in Huh7.5 cells prestimulated with 280 pM IFNα for 24 h (Fig 4B, left panel). The corresponding 68%-confidence intervals were computed as proposed in Kreutz *et al* (2012). The simulation showed a steep increase of occupied GAS-binding sites within the first hour after stimulation, which was suppressed upon prestimulation of cells with IFNα. As shown in Fig 4B, left panel, the mean values of pSTAT1 homodimeric complexes detected by EMSA ($N = 3$) were in agreement with the model prediction and experimentally confirmed a rapid but transient formation of pSTAT1 homodimers in response

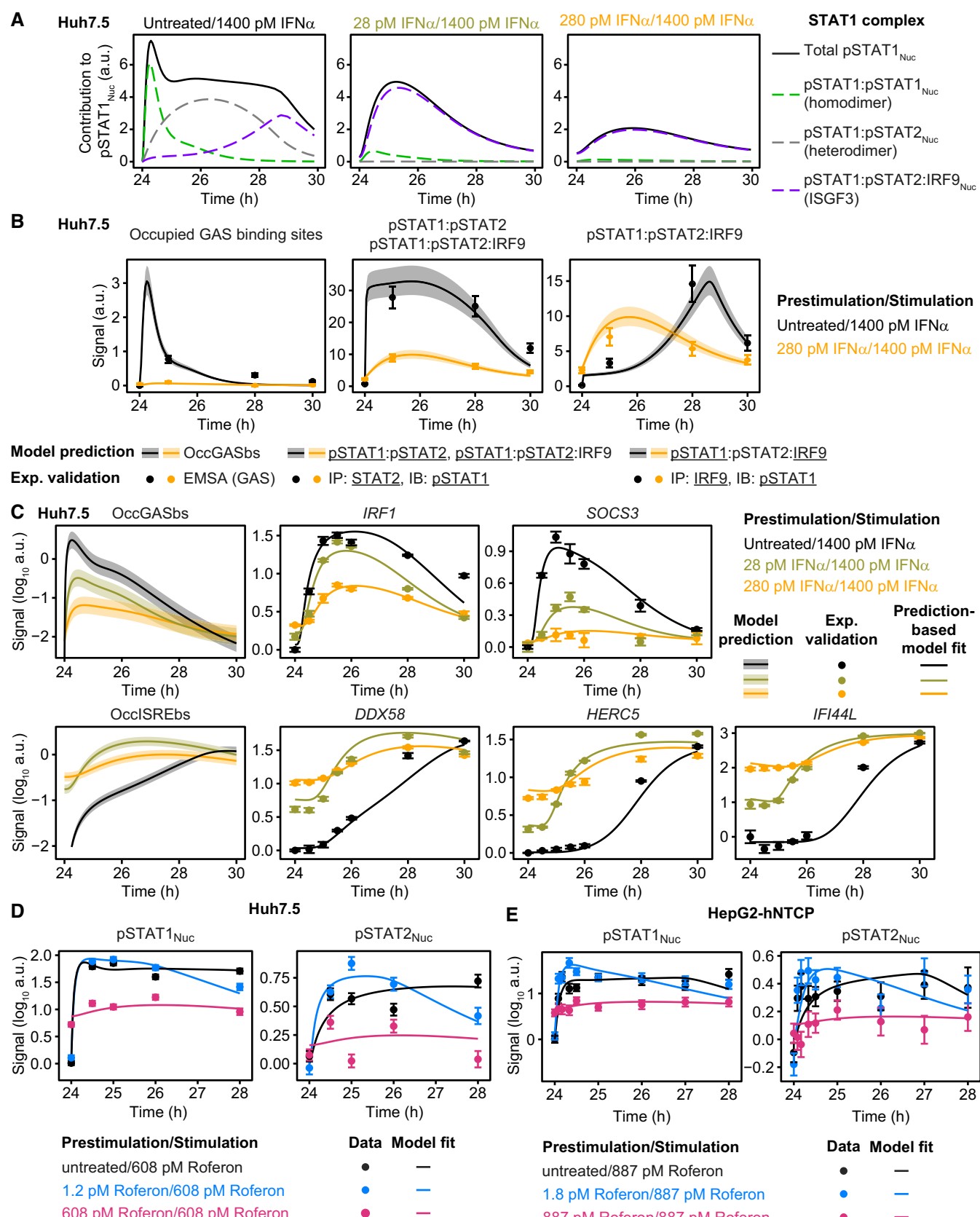

**Figure 4.**

**Figure 4. Model analysis of the dynamics of pSTAT1 complex formation and model application to Roferon and HepG2-hNTCP cells.**

A   Model analysis reveals impact of different prestimulation doses on the dynamics of pSTAT1-containing nuclear complexes. The time-resolved amounts of nuclear pSTAT1 homodimers, pSTAT1:pSTAT2 heterodimers, and pSTAT1:pSTAT2:IRF9 trimers were calculated by the mathematical model. Simulations were performed for Huh7.5 cells stimulated with 1,400 pM IFNα that were either untreated or prestimulated with 28 pM IFNα or 280 pM IFNα for 24 h. Different STAT1 comprising transcription factor complexes are indicated.

B   Model predictions of IFNα-induced dynamics of occupied GAS-binding sites (OccGASbs) (left panel), of the sum of the pSTAT1:pSTAT2 heterodimers and the pSTAT1:pSTAT2:IRF9 trimers (middle panel) and of the pSTAT1:pSTAT2:IRF9 trimers (right panel) in untreated Huh7.5 cells and in Huh7.5 cells prestimulated for 24 h with 280 pM IFNα that were subsequently stimulated with 1,400 pM IFNα. Lines with shading represent model predictions with 68% confidence intervals using the prediction profile likelihood method. For experimental validation of the dynamics of OccGASbs, electrophoretic mobility shift assays (EMSA) were performed using nuclear protein lysates obtained from untreated Huh7.5 cells or Huh7.5 cells that were prestimulated for 24 h with 280 pM IFNα and then stimulated with 1,400 pM IFNα. Lysates were incubated with radioactively labeled oligonucleotides probes harboring the GAS-binding region of the human *IRF1* promoter. Samples were resolved on a native polyacrylamide gel, and radioactivity was visualized and quantified from three independent experiments (left panel). For experimental validation of the dynamics of the sum of the pSTAT1:pSTAT2 heterodimers and the pSTAT1:pSTAT2:IRF9 trimers, immunoprecipitations (IP) were performed using total cell lysates obtained from untreated Huh7.5 cells or Huh7.5 cells that were prestimulated for 24 h with 280 pM IFNα and then stimulated with 1,400 pM IFNα. Lysates were subjected to immunoprecipitation with antibodies recognizing STAT2 and phosphorylated STAT1 was detected with quantitative immunoblotting (IB) (middle panel). For experimental validation of the dynamics of the pSTAT1:pSTAT2:IRF9 trimers, immunoprecipitations were performed using total cell lysates obtained from untreated Huh7.5 cells or Huh7.5 cells that were prestimulated for 24 h with 280 pM IFNα and then stimulated with 1,400 pM IFNα. Lysates were subjected to immunoprecipitation (IP) with antibodies recognizing IRF9 and phosphorylated STAT1 was detected with quantitative immunoblotting (IB) (right panel). Antibodies and the corresponding proteins in the complexes are underlined. Experimental data are represented by filled circles with errors representing 1σ confidence intervals estimated from biological replicates (N = 3) using a combined scaling and error model.

C   Model predictions of IFNα-induced dynamics of occupied GAS-binding sites (OccGASbs) and of occupied ISRE-binding sites (OccISREbs) in Huh7.5 cells without prestimulation and in cells prestimulated for 24 h with 28 and 280 pM IFNα that were subsequently stimulated with 1,400 pM IFNα. Model predictions were performed using the prediction profile likelihood method. Lines with shading represent model predictions with 68% confidence intervals. For experimental validation, growth factor-depleted Huh7.5 cells were prestimulated with 0, 28, and 280 pM IFNα. After 24 h, cells were stimulated with 1,400 pM IFNα and IFNα-induced expression of target genes was measured by qRT–PCR. RNA levels were normalized to the geometric mean of reference genes *GAPDH*, *HPRT*, and *TBP*, averaged and displayed as fold change, represented by filled circles with errors representing standard error of the mean calculated from biological replicates (N = 3). Except for gene-specific parameters (mRNA synthesis and degradation rates, time delay parameter and Hill coefficient), qRT–PCR data were used for model validation but not for model calibration.

D   Dose-dependent sensitization of signal transduction induced by the therapeutic interferon α Roferon. Growth factor-depleted Huh7.5 were prestimulated with 0, with 1.2 or 608 pM Roferon and stimulated with 608 pM Roferon 24 h later. Concentrations of Roferon correspond to equipotent concentrations of IFNα. Nuclear lysates were subjected to quantitative immunoblotting and Roferon-induced phosphorylation of STAT1 and STAT2 was detected by chemiluminescence utilizing a CCD camera-based device (ImageQuant). Filled circles represent scaled data with errors representing 1σ confidence intervals estimated from biological replicates (N = 3 to N = 8) using a combined scaling and error model. Model trajectories are represented by lines.

E   Dose-dependent sensitization of signal transduction induced by Roferon in HepG2-hNTCP cells. Growth factor-depleted HepG2-hNTCP were prestimulated with 0, 1.8 or 887 pM Roferon and stimulated with 887 pM Roferon 24 h later. Nuclear lysates were subjected to quantitative immunoblotting and Roferon-induced phosphorylation of STAT1 and STAT2 was detected by chemiluminescence utilizing a CCD camera-based device (ImageQuant). Filled circles represent scaled data with errors representing 1σ confidence intervals estimated from biological replicates (N = 3) using a combined scaling and error model. Model trajectories are represented by lines.

to stimulation with IFNα, which was much reduced upon IFNα prestimulation, validating the model-predicted early occurrence of pSTAT1 homodimers in response to IFNα stimulation.

To investigate the IFNα-induced dynamics of the formation of the other STAT1-containing transcription factor complexes, we performed co-immunoprecipitation (co-IP) experiments. We stimulated non-prestimulated Huh7.5 cells or Huh7.5 cells prestimulated with 280 pM IFNα for 24 h for 1–6 h (corresponding to 25–30 h after prestimulation) with 1,400 pM IFNα. The cellular lysates were used for immunoprecipitation experiments using antibodies recognizing STAT2 and co-immunoprecipitated pSTAT1 was detected by quantitative immunoblotting (Fig EV3B). With these co-IP experiments the dynamics of the sum of IFNα-induced formation of pSTAT1:pSTAT2 heterodimers and pSTAT1:pSTAT2:IRF9 trimers (ISGF3) was detected. In non-prestimulated Huh7.5 cells the signal for co-immunoprecipitating pSTAT1 was maximal after 1 h of stimulation with 1,400 pM IFNα and slowly decreased thereafter but not reaching baseline after 6 h of stimulation (25–28 h after mock prestimulation). Upon prestimulation of Huh7.5 cells with 280 pM IFNα for 24 h, higher levels of total STAT2 were observed, while the overall signal of co-immunoprecipitated pSTAT1 was lower. Distinctively, it was already detectable after 24 h of prestimulation and increased to a much lower extent by stimulation with 1,400 pM IFNα compared to the amount detected in untreated cells. To

compare the experimental results obtained by the quantification of co-immunoprecipitated pSTAT1 (N = 3) to the predictions by our mathematical model, we calculated the dynamics of the sum of pSTAT1:pSTAT2 heterodimers and the pSTAT1:pSTAT2:IRF9 trimers (ISGF3) induced by 1,400 pM IFNα in non-prestimulated Huh7.5 cells and Huh7.5 cells prestimulated with 280 pM IFNα for 24 h and computed 68%-confidence intervals. As shown in Fig 4B, middle panel, the model-predicted broad peak of the sum of pSTAT1:pSTAT2 heterodimers and ISGF3 was in line with the experimental data and was reduced to around one-third upon prestimulation of the cells.

Additionally, to quantify in non-prestimulated and prestimulated Huh7.5 cells the dynamics of IFNα-induced ISGF3 complex formation, co-IP experiments were performed using antibodies recognizing IRF9 and co-immunoprecipitated pSTAT1 was detected by quantitative immunoblotting (Fig EV3C). The signal for IRF9-precipitated pSTAT1 increased upon stimulation of non-prestimulated Huh7.5 cells with 1,400 pM IFNα with a peak at 4 h post-IFNα treatment (28 h after mock prestimulation). Upon prestimulation of Huh7.5 cells with 280 pM IFNα for 24 h, IRF9 levels were strongly increased and co-immunoprecipitated pSTAT1 was now already peaking at around 1 h after IFNα stimulation (25 h after prestimulation). The mean values for pSTAT1 (N = 3) at the different time points of IFNα stimulation of untreated and prestimulated Huh7.5

cells were in line with the model-predicted dynamics of ISGF3 complex formation in response to IFNα stimulation confirming the late increase of ISGF3 transcription factor complexes in untreated cells and acceleration of the formation to 1 h after IFNα stimulation in cells prestimulated with 280 pM IFNα. Overall, these results confirmed our model predictions that whereas pSTAT1 homodimers are rapidly and transiently formed in response to IFNα stimulation, it takes several hours until enough IRF9 protein has been produced to assemble significant amounts of the ISGF3 complex before it becomes the dominant transcription factor complex.

To experimentally demonstrate the impact of these transcription factor complexes on the expression dynamics of target genes, we had to first identify interferon target genes whose expression is primarily regulated by the presence of GAS- or ISRE-binding sites. As shown in Appendix Fig S6A, bioinformatics analyses revealed that promoter regions of the *IRF1* and *SOCS3* genes harbor primarily putative GAS sequences, while the *DDX58*, *HERC5*, and *IFI44L* genes contain primarily putative ISRE sites in proximity to the transcription start site. To experimentally verify that these genes are primarily driven by either GAS or ISRE sites, we stimulated Huh7.5 cells with 5,000 IU/ml of either IFNγ or IFNα (corresponding to 1,400 pM IFNα) (Appendix Fig S6B). In line with the promoter analysis, stimulation with IFNγ, which only induces phosphorylation of STAT1 and therefore the formation of pSTAT1 homodimers, resulted in a sustained expression of *IRF1* and *SOCS3*, but not of the other genes. In agreement with our model-based prediction that IFNα triggers the transient formation of pSTAT1 homodimers and the sustained formation of ISGF3 complexes, IFNα induced a transient expression of *IRF1* and *SOCS3* with a peak around 1 h after IFNα stimulation, while it induced a sustained expression for *DDX58*, *HERC5*, and *IFI44L* in the timeframe of 4 h. In sum, these experiments established that the expression of *IRF1* and *SOCS3* is controlled by the presence of GAS sequences, whereas the expression of *DDX58*, *HERC5*, and *IFI44L* is primarily dependent on the presence of ISRE sites in Huh7.5 cells.

To further evaluate the distinct IFNα dose-dependent formation of IFNα-induced transcription factor complexes as predicted by the mathematical model (Fig 4A), we simulated the dynamics of the occupancy of GAS-binding sites upon stimulation with 1,400 pM IFNα after prestimulation with 28 and 280 pM IFNα. Prestimulation with these IFNα doses was predicted by the mathematical model to reduce pSTAT1 homodimer formation and consequently the occupancy of GAS-binding sites in a dose-dependent manner upon stimulation with 1,400 pM IFNα (Fig 4C, upper left panel). The corresponding 68%-confidence intervals (shaded areas in Fig 4C) were computed as proposed in Kreutz *et al* (2012). To verify this model prediction, we examined the dynamics of the production of the GAS-dependent transcripts *IRF1* and *SOCS3* upon stimulation with 1,400 pM IFNα in untreated cells and after prestimulation with 28 and 280 pM IFNα (Fig 4C, symbols in upper right panels). In accordance with the mathematical model, the dynamics of the expression of these genes was reduced by the prestimulation with 28 and 280 pM IFNα and reflected the predicted reduced formation of the pSTAT1 homodimers. Conversely, the dynamics of the formation of ISGF3 complexes and consequently the occupancy of ISRE-binding sites upon stimulation with 1,400 pM IFNα after prestimulation with 28 and 280 pM IFNα was predicted. In contrast to the GAS- and ISRE-binding sites controlling the expression of *STAT1*,

*STAT2*, *IRF9*, *IRF2*, *USP18*, and *SOCS1* that are occupied by ISGF3 and pSTAT1:pSTAT2 heterodimers, the ISRE-binding sites are only occupied by ISGF3. The model predicted that the prestimulation with 28 pM IFNα resulted in a higher initial level of occupied ISRE-binding sites at 24 h and an accelerated increase of occupied ISRE-binding sites upon stimulation with 1,400 pM IFNα. Prestimulation with 280 pM IFNα was predicted to further increase the initial occupancy of ISRE-binding sites after 24 h and the maximum occupancy of ISRE-binding sites upon stimulation with 1,400 pM IFNα was predicted to be similar as in cells that were not prestimulated (Fig 4C, lower left panel). The experimental analysis of the dynamics of the ISRE-dependent transcripts *DDX58*, *HERC5*, and *IFI44L* confirmed upon prestimulation with 28 pM IFNα a higher basal expression at 24 h and an accelerated gene induction compared to cells that were not prestimulated. Upon prestimulation with 280 pM IFNα, the basal expression of *DDX58*, *HERC5*, and *IFI44L* was even higher at 24 h and the maximum expression upon stimulation with 1,400 pM IFNα was similar to cells that were not prestimulated, reflecting the predicted dynamics of the occupied ISRE-binding sites (Fig 4C, symbols in lower right panels). These measured transcripts were linked to the amount of occupied binding sites predicted by the mathematical model by estimating gene-specific parameters, i.e., mRNA synthesis and degradation rate, time delay of mRNA production, and the Hill coefficient, while all remaining model parameters were fixed. This allowed to overlay the measured dynamics of this gene set with the simulated model trajectories (Fig 4C, lines in upper and lower right panels).

We further validated our model prediction using SOCS3 protein dynamics. We first predicted the IFNα-induced dynamics of occupied GAS-bindings sites (OccGASbs) in Huh7.5 cells without prestimulation and in cells prestimulated for 24 h with 2.8 and 28 pM IFNα that were subsequently stimulated with 1,400 pM IFNα (Fig EV3C). Also with these lower prestimulation doses, we observed that the amount of the OccGASbs was reduced in a dose-dependent manner. In Fig EV3D, the experimental data points of quantified SOCS3 protein expression were overlaid with simulated model trajectories. The corresponding 68%-confidence intervals (shaded areas in Fig EV3D) were computed as proposed in Kreutz *et al* (2012). The results showed that in untreated cells and cells prestimulated with 2.8 pM IFNα, SOCS3 expression was rapidly induced upon stimulation with 1,400 pM IFNα, while only a minor induction of the SOCS3 protein was observed in cells prestimulated with 28 pM IFNα, thereby confirming our model-based hypothesis and thus validating the capacity of the model to predict the dynamics of IFNα-induced formation of the different STAT1 transcription factor complexes in Huh7.5 cells.

Next, we investigated whether the model could be applied to analyze the dynamic behavior of signal transduction induced by a therapeutic agent such as Roferon (interferon α-2a, Roche). First, equipotent doses of Roferon and the research grade IFNα, which we utilized in this study, were determined by a subgenomic HCV replicon assay (Fig EV4A). According to the obtained IC$_{50}$ values (Fig EV4B) that were verified in a dose–response experiment measuring pSTAT1 and pSTAT2 by quantitative immunoblotting (Fig EV4C), Huh7.5 cells were prestimulated with a low dose of 1.2 pM Roferon (corresponding to 2.8 pM IFNα), a high dose of 608 pM Roferon (corresponding to 1,400 pM IFNα), or were left untreated. After 24 h, the cells were stimulated by adding a high dose of

 

608 pM Roferon and phosphorylation of cytoplasmic (Fig EV4D) and nuclear (Fig 4D) STAT1 and STAT2 was investigated for up to 4 h by quantitative immunoblotting. The experimental data and the model trajectories were highly similar to those obtained in the experiments performed with the research grade IFNα (Fig 1D). Again, prestimulation with a high dose of Roferon resulted in pathway desensitization as indicated by lower levels of both nuclear and cytoplasmic pSTAT1 and pSTAT2, while prestimulation with a low dose of Roferon generated pathway hypersensitization, most evident for nuclear pSTAT2, confirming that dose-dependent sensitization of IFNα signal transduction in Huh7.5 cells is also established by Roferon.

To address whether the extent of pathway sensitization is cell type-specific, we examined Roferon-induced phosphorylation of STAT1 and STAT2 in the nucleus of HepG2-hNTCP cells, which are commonly used to study infection of hepatitis B virus (Hoh *et al*, 2015). We measured the abundance of STAT1, STAT2, IRF9, and USP18 in HepG2-hNTCP cells (Fig EV5A), which were significantly different from the corresponding abundances detected in Huh7.5 cells (Fig 3C). While STAT1 and STAT2 were of lower abundance in HepG2-hNTCP cells than in Huh7.5 cells, the number of molecules per cell of IRF9 and USP18 were higher in untreated HepG2-hNTCP cells. To determine differences in parameters of the IFNα signal transduction model between HepG2-hNTCP and Huh7.5 cells, we employed our previously established method to identify cell type-specific parameters based on $L_1$ regularization (Merkle *et al*, 2016). This analysis revealed that the basal synthesis rate of *STAT1* mRNA and the parameter comprising phosphorylation and association of STAT1 and STAT2 were different in HepG2-hNTCP cells compared to Huh7.5 cells. 301 data points generated for three experimental conditions were used for calibration of these additional model parameters, and the protein abundance of STAT1, STAT2, IRF9, and USP18 determined for HepG2-hNTCP cells were incorporated (Figs 4E and EV5B). The other dynamic parameters were fixed to parameter values estimated from the Huh7.5 IFNα dataset. Distinct from Huh7.5 cells that were kept in 1.5 ml medium, HepG2-hNTCP cells were cultivated in 1 ml and therefore the applied Roferon doses were adjusted accordingly (see Materials and Methods). HepG2-hNTCP cells were prestimulated for 24 h with a low dose of 1.8 pM Roferon, a high dose of 887 pM Roferon or were left untreated and were subsequently stimulated with a high dose of 887 pM Roferon. The experimental results (Fig 4E) revealed that prestimulation with a high Roferon dose very much decreased the Roferon-induced presence of pSTAT1 and pSTAT2 in the nucleus of HepG2-hNTCP cells confirming desensitization of the pathway. However, hypersensitization of pSTAT2 phosphorylation upon prestimulation with a low Roferon dose was less pronounced compared to Huh7.5 cells (Fig 4D). As shown in Fig 4E, the obtained model trajectories were in line with the experimental data confirming that the mathematical model is capable to represent the IFNα dose-dependent sensitization of the IFNα signal transduction pathway independent of the cell type.

To investigate the impact of the different ratio between the STAT proteins and IRF9 in HepG2-hNTCP cells on formation of pSTAT1-containing transcription factor complexes, we simulated with our mathematical model the dynamics of pSTAT1:pSTAT1 homodimers, pSTAT1:pSTAT2 heterodimers, and ISGF3 complexes in the nucleus of HepG2-hNTCP cells (Appendix Fig S7A). Unlike Huh7.5 cells, the

mathematical model predicted a rapid formation of ISGF3 complexes within 30 min due to the higher amounts of IRF9 compared to STAT1 being present in untreated HepG2-hNTCP cells. Stimulation with IFNα results in a gradual increase in IRF9 production and therefore in a further increase in the formation of ISGF3 complexes 2 h later. Further, the mathematical model suggested that in HepG2-hNTCP cells pSTAT1:pSTAT1 homodimers and pSTAT1:pSTAT2 heterodimers are formed with a similar dynamics as in Huh7.5 cells in response to IFNα stimulation, but the amounts of these complexes are lower. Additionally, our mathematical model simulations indicated that prestimulation of HepG2-hNTCP cells with 28 or 280 pM IFNα for 24 h reduces the formation of these complexes in a dose-dependent manner, while the formation of the ISGF3 complex is, similar to Huh7.5 cells, much accelerated.

To experimentally validate in HepG2-hNTCP cells the impact of the formation of these transcription factor complexes on the expression dynamics of the target genes selected for Huh7.5 cells, we first stimulated HepG2-hNTCP cells with 5,000 IU/ml of either IFNγ or IFNα (corresponding to 1,400 pM IFNα) (Appendix Fig S7B). Similar to Huh7.5 cells, stimulation with IFNγ induced in HepG2-hNTCP cells the expression of *IRF1* and *SOCS3*, but not of *DDX58*, *HERC5*, and *IFI44L*. In line with the mRNA expression dynamics observed in Huh7.5 cells, IFNα stimulation of HepG2-hNTCP cells resulted after 4 h in the induction of the expression of *DDX58*, *HERC5*, and *IFI44L* and in an immediate increase in the expression of *SOCS3* with a peak at around 1 h after IFNα stimulation. For *IRF1* an approximately 50-fold increase of mRNA expression was observed within 1 h after IFNα stimulation, which was in line with the mRNA expression dynamics in Huh7.5 cells and confirmed *IRF1* as an immediate early gene of IFNα-induced responses. However, in HepG2-hNTCP cells the IFNα-induced expression of *IRF1* was more sustained suggesting that the down regulation of the *IRF1* expression in HepG2-hNTCP is potentially modulated by the cell context-specific activation of other transcription factors. This is in line with previous reports that the expression of *IRF1* can also be regulated by the NFκB and MAP-kinase pathways (Yarilina *et al*, 2008), and therefore, we did not include *IRF1* in our further analyses.

We simulated with our mathematical model the dynamics of the occupancy of GAS-binding sites induced in HepG2-hNTCP cells either untreated or prestimulated with 28 or 280 pM IFNα by stimulation with 1,400 pM IFNα. Prestimulation with 28 pM IFNα was predicted to have little impact on the peak amplitude of the occupancy of GAS-binding sites, while prestimulation with 280 pM IFNα reduced the peak amplitude of occupied GAS-binding sites by an order of magnitude (Appendix Fig S7C, upper left panel). In accordance with the model prediction, the dynamics of the expression of experimentally measured expression of *SOCS3* was only slightly reduced by the prestimulation with 28 pM IFNα, but almost completely abolished by prestimulation with 280 pM IFNα (Appendix Fig S7C, upper right panel). Moreover, similar to Huh7.5 cells, the mathematical model predicted for HepG2-hNTCP cells a higher initial level of occupied ISRE-binding sites at 24 h of prestimulation and an accelerated increase of occupied ISRE-binding sites upon stimulation with 1,400 pM IFNα (Appendix Fig S7C, lower left panel), which was in agreement with the experimentally observed expression of the ISRE-dependent transcripts *DDX58*, *HERC5*, and *IFI44L* (Appendix Fig S7C, lower right panel). These experiments demonstrate that the mathematical model can also predict the

dynamics of IFNα-induced transcription factor complex formation in another liver cell line and thus confirm the broader applicability of the mathematical model.

To conclude, upon stimulation of untreated cells pSTAT1 primarily contributes to the presence of pSTAT1 homodimers and subsequently pSTAT1:pSTAT2 heterodimers. In cells prestimulated with a low IFNα dose, IRF9 levels are increased and therefore ISGF3 complexes prevail. Furthermore, an IFNα dose-dependent pathway sensitization was also confirmed for the therapeutic agent Roferon and the sequential formation of STAT1-containing transcription factor complexes is not specific for a certain cell type, but rather observed for both Huh7.5 and HepG2-hNTCP cells.

## USP18 is not sufficient to desensitize IFNα-induced signal transduction

Since it was previously reported that USP18 acts as key negative regulator of IFNα signal transduction and we observed that an increase of USP18 correlated with an increase of pathway desensitization (Figs 3C compared to D, and EV1), we investigated whether USP18 alone is sufficient to establish pathway desensitization.

First, we studied the impact of knockdown of USP18 on the effect of prestimulation with IFNα and on the establishment of pathway desensitization. One day prior to the start of the experiment, Huh7.5 cells were transfected with siRNAs targeting USP18. Subsequently, the cells were prestimulated with 1,400 pM IFNα for 24 h and were stimulated with 1,400 pM IFNα or were left untreated. IFNα-induced phosphorylation and feedback induction was investigated by quantitative immunoblotting (Fig 5A, Appendix Fig S8B) and qRT-PCR (Appendix Fig S8C). As shown in Appendix Fig S8A, upon stimulation with IFNα, cells transfected with the non-targeting control siRNA showed a comparable dynamics of activation of the IFNα signal transduction pathway relative to the dynamics observed in untransfected Huh7.5 cells, indicating that the method did not interfere with the dynamics of the IFNα signal transduction pathway. Therefore, the data obtained in cells transfected with control siRNA were scaled together with data obtained in untransfected Huh7.5 cells. The quantitative analysis shown in the first panel of Fig 5A identified an average knockdown efficiency of 94.5 ± 2% at the USP18 protein level after 24 h of stimulation with 1,400 pM IFNα. In control cells incubated with 1,400 pM IFNα, a transient increase of pSTAT1 and pSTAT2 in the cytoplasm was observed at 3 h that returned close to basal levels after 8 h and showed a dampened second peak beyond 12 h (Fig 5A, right panel). On the contrary, in cells transfected with USP18 siRNA, pSTAT1, and pSTAT2 levels in the cytoplasm remained sustained beyond 24 h. Nuclear pSTAT1 and pSTAT2 levels showed a comparable sustained dynamic behavior upon USP18 knockdown (Appendix Fig S8B). The experimentally observed levels of pSTAT1 and pSTAT2 in cytoplasm and nucleus in USP18 knockdown cells were captured by the calibrated mathematical model and confirmed the role of USP18 as a negative regulator of IFNα signal transduction (Fig 5A and Appendix Fig S8B). In addition, USP18 knockdown resulted in an increased induction of the mRNAs of feedback genes (Appendix Fig S8C) and of the feedback proteins STAT1 (Fig 5A, tSTAT1$_{Cyt}$), STAT2, IRF9, and USP18 (Appendix Fig S8B). As described by the data and the model trajectories, stimulation of control and USP18 knockdown cells 24 h after prestimulation with a high IFNα dose triggered only a weak

increase in pSTAT1 and pSTAT2. The knockdown experiments confirmed USP18 as a major negative feedback, which is also captured by the calibrated mathematical model.

To investigate whether an increase in the abundance of USP18 alone is sufficient to cause desensitization of the IFNα signal transduction pathway, we established a stable Huh7.5 cell line with inducible USP18 expression. Instead of prestimulation with IFNα, cells were pretreated with doxycycline for 24 h to induce the expression of USP18 protein. Figure 5B (first panel) shows that USP18 protein expression was induced to a similar extent as in parental Huh7.5 cells stimulated for 24 h with 1,400 pM IFNα. In control cells, Huh7.5 cells transduced with the empty TetON vector, doxycycline treatment did not induce the expression of USP18 protein. In these control cells, a similar dynamics of IFNα-induced signal transduction was observed as in untreated parental Huh7.5 cells and therefore data of both conditions were scaled together (Appendix Fig S9A). Stimulation of the inducible USP18 overexpressing cell line (Huh7.5-TetON-USP18) with 1,400 pM IFNα for up to 6 h resulted in a slightly lower activation of pSTAT1 and pSTAT2 relative to cells that had not been prestimulated with IFNα, which is in agreement with USP18 acting as a negative regulator of the pathway (Fig 5B). Unexpectedly, distinct from cells prestimulated for 24 h with 1,400 pM IFNα that showed desensitization of pSTAT1 and pSTAT2 activation in the cytoplasm, the USP18 overexpressing cell line that contained comparable amounts of USP18 did not show this strong desensitization but rather displayed a significantly higher extent of pathway activation than the prestimulated cells (Fig 5B and Appendix Fig S9B). The experimental data and the mathematical model together indicated that USP18 alone is not sufficient to cause pathway desensitization. Therefore, we tested whether in addition SOCS1 is required. To identify possible mechanisms that could explain pathway desensitization and the USP18 overexpression data, different model structures were tested. While USP18 and SOCS1 both inhibit activation of the receptor, the impact of these molecules on internalization of the active receptor was unclear. Internalization of the active receptor could be enhanced by SOCS1, by USP18 or only if both molecules were binding. We tested different model structures and evaluated the resulting goodness-of-fit by means of the Bayesian information criterion (BIC) (Fig 5C). As exemplified by the fit of nuclear pSTAT1 (Fig 5C) and the lowest BIC value (Fig 5D), the model captured the experimental data best when taking both SOCS1-mediated and SOCS1:USP18-mediated degradation of the activated receptor into account.

We concluded that, USP18 indeed acts as a negative regulator and impacts signal attenuation, but since USP18 overexpression alone was not sufficient to induce pathway desensitization, we propose that USP18 acts as a cofactor for SOCS1-mediated degradation of active IFNα receptor complexes.

## Identification of components predictive for sensitization of the IFNα signal transduction pathway in cell lines and primary human hepatocytes

To dissect mechanisms that regulate dose-dependent sensitization of IFNα signal transduction in hepatoma cell lines, we employed the mathematical model to simulate in Huh7.5 and HepG2-hNTCP cell lines the amounts of STAT1, STAT2, IRF9, and USP18 protein for untreated cells and for cells prestimulated with 1.2 pM or 608 pM

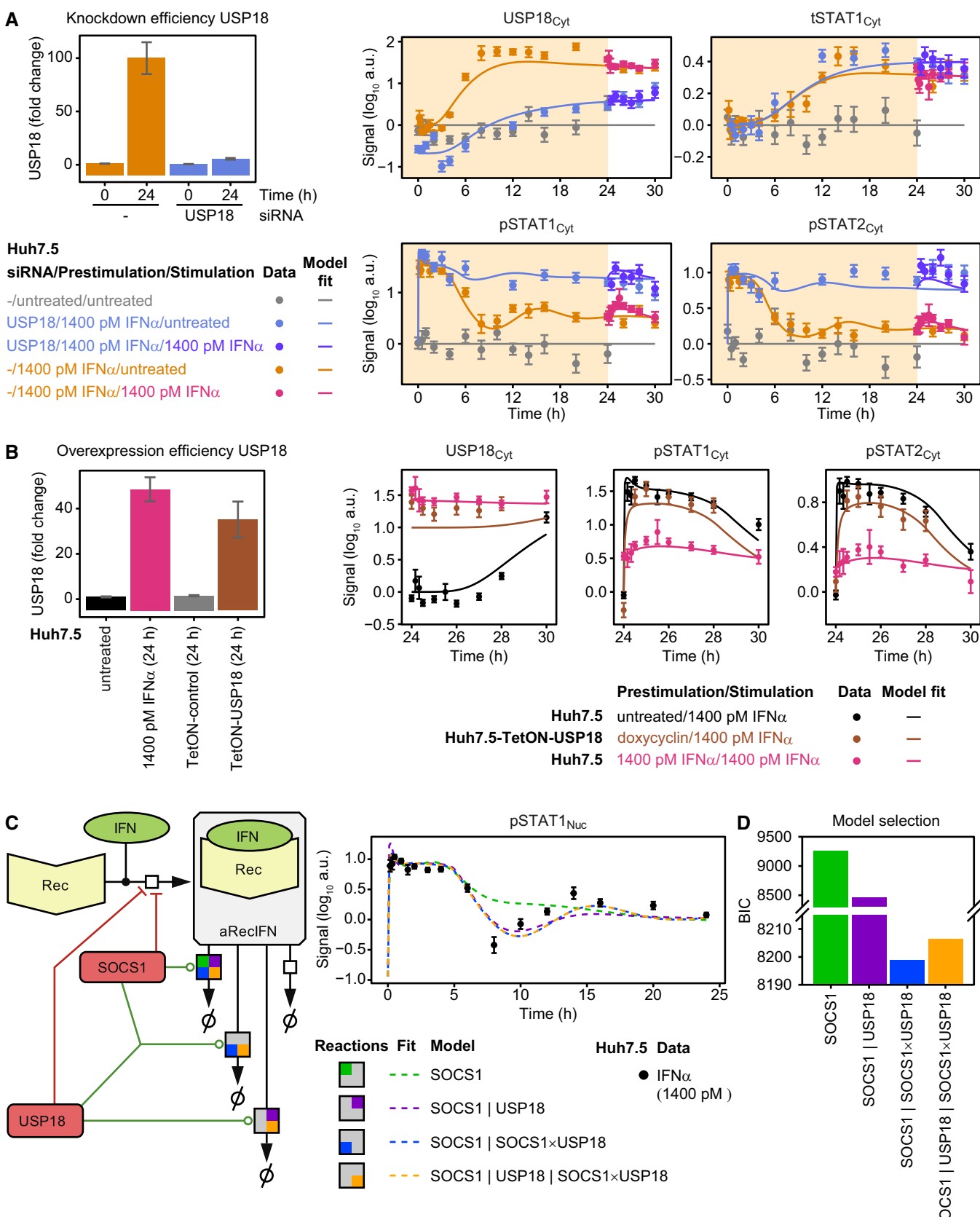

**Figure 5.**

◀

**Figure 5.  USP18 alone is not sufficient to induce desensitization.**

A  Knockdown of USP18 results in sustained signaling. USP18 knockdown efficiency determined by quantitative immunoblotting of cytoplasmic lysates of Huh7.5 cells transfected with USP18 siRNA relative to parental Huh7.5 cells and Huh7.5 cells transfected with non-targeting siRNA. Cells were stimulated with 1,400 pM IFNα for 24 h. Error bars represent 1σ confidence intervals estimated from biological replicates (*N* = 6 to *N* = 10) (left). Model fit and experimental data of Huh7.5 cells transfected with control siRNA or USP18 siRNA are shown. Cells were transfected with USP18 siRNA or control siRNA 1 day prior to growth factor-depletion. Next, cells were prestimulated with 1,400 pM IFNα (yellow background) and stimulated with 1,400 pM IFNα at 24 h or untreated (white background). IFNα-induced phosphorylation of STAT1 and STAT2 and induction of USP18 and tSTAT1, comprising both phosphorylated and unphosphorylated STAT1 protein, are displayed. Experimental data were obtained by quantitative immunoblotting using chemiluminescence and CCD camera-based device (ImageQuant). For model purposes, data in control siRNA and untransfected Huh7.5 are combined to one condition. Data from multiple time courses scaled together are displayed as filled circles with errors representing 1σ confidence intervals estimated from biological replicates (*N* = 2 to *N* = 10) using a combined scaling and error model. Lines represent model trajectories.

B  Overexpression of Huh7.5 is not sufficient to explain desensitization. Induced expression of USP18 after treatment of Huh7.5-TetON-USP18 and Huh7.5-TetON-control cells with doxycycline for 24 h in comparison with parental Huh7.5 cells treated with 1,400 pM IFNα for 24 h. Analysis was performed by quantitative immunoblotting. Error bars represent 1σ confidence intervals estimated from biological replicates (*N* = 3 to *N* = 5) (left). Model fits and experimental data of Huh7.5-TetON-USP18 treated with doxycycline for 24 h and stimulated with 1,400 pM IFNα or parental Huh7.5 cells prestimulated with 0 or 1,400 pM IFNα and stimulated with 1,400 pM IFNα after 24 h are shown. The dynamics of IFNα-induced phosphorylation of STAT1 and STAT2 and induction of USP18 are depicted. Experimental data were obtained by quantitative immunoblotting using chemiluminescence and CCD camera-based device (ImageQuant). For modeling purposes, data from Huh7.5-TetON empty vector control and untransduced Huh7.5 are combined to one condition. Data are displayed as filled circles with errors representing 1σ confidence intervals estimated from biological replicates (*N* = 3 to *N* = 4) using a combined scaling and error model. Line represents model trajectories.

C  Scheme of possible mechanisms for SOCS1- and USP18-induced receptor degradation. Models with different structures concerning SOCS1- and USP18-catalyzed degradation of active receptor complexes (aRecIFN) were tested. Vertical lines denote separate enzymatic reactions, multiplication sign denotes cooperative enzymatic reactions. For each of the four different model structures, the resulting model trajectories of the best fit of pSTAT1 in the nucleus are exemplarily shown as dashed lines. Data are displayed as filled circles with errors representing 1σ confidence intervals estimated from biological replicates (*N* = 1 to *N* = 38).

D  Both SOCS1- and SOCS1:USP18-induced receptor degradation is required. Bayesian information criterion (BIC) was used to evaluate goodness-of-fit for the four different models shown in (C). The model "SOCS1 | SOCS1 × USP18" that shows the best performance comprises degradation of the active receptor complexes by both SOCS1 and a SOCS1:USP18 complex.

Roferon for 24 h (Fig 6A). The model simulations were in line with the corresponding experimental data (Figs 3C and EV5A) and revealed the presence of slightly lower amounts of IRF9 and USP18 in untreated Huh7.5 cells compared to untreated HepG2-hNTCP cells and an IFNα dose-dependent increase in the amounts of STAT1, STAT2, IRF9, and USP18 in both cell systems. We hypothesized that the amount of one or several of these proteins may be predictive for the sensitization of the IFNα signal transduction pathway. To test this model-derived hypothesis in primary human hepatocytes, we first investigated the patient-to-patient variability in the abundance of the feedback proteins STAT1, STAT2, IRF9, and USP18 by quantitative immunoblotting in patient-derived primary human hepatocytes that were isolated from tumor-free tissue of six patients (black bars in Fig 6B). The basal levels of STAT1 ranged from $10^5$ to $10^6$ molecules per cell, whereas for STAT2 $10^4$ to $10^5$ molecules per cell were present, similar to untreated Huh7.5 and HepG2-hNTCP cells (Figs 3C and EV5A). IRF9 amounts were similar to STAT2 amounts, and being more than one order of magnitude higher expressed in primary human hepatocytes than in the untreated cell lines. In addition, USP18 amounts highly varied between the patients, the abundance being slightly lower than the abundance of IRF9.

To test whether these patient-specific amounts lead to differences in pathway sensitization, we prestimulated primary human hepatocytes with 2.8 and 1,400 pM IFNα (Patients 1–3) or with 1.2 and 608 pM Roferon (Patients 4–6) or left cells untreated. After 24 h, primary human hepatocytes were stimulated with 1,400 pM IFNα (Patients 1–3) or 608 pM Roferon (Patients 4–6), respectively, and phosphorylation of cytoplasmic (Patients 1–3) or total (Patients 4–6) STAT1 and STAT2 was investigated for up to 4 h by quantitative immunoblotting (Fig 6C). Additionally, the cytoplasmic (Patients 1–3) or total (Patients 4–6) amounts of STAT1, STAT2, IRF9, and USP18 were measured (Appendix Fig S10). We utilized our

calibrated mathematical model of the IFNα signal transduction pathway to analyze the experimental data (Fig 6B and C, and Appendix Fig S10) and defined additional primary human hepatocyte-specific model parameters based on $L_1$ regularization. Altogether 816 data points belonging to three experimental conditions were used for the calibration of these parameters. The following parameters were different in primary human hepatocytes compared to Huh7.5 cells: the amount of the receptor complex (Rec), the basal and induced synthesis rate of *STAT2* mRNA, the degradation rate of *SOCS1* mRNA, and the dissociation rate of nuclear ISGF3. The calibrated mathematical model was able to simultaneously represent the patient-specific abundance of STAT1, STAT2, IRF9, and USP18 (green rectangles in Fig 6B), which was in good agreement with the experimental determinations, and captured the patient-specific dynamics observed in the time-resolved measurements (lines in Fig 6C, Appendix Fig S10). Both experimental data and mathematical model showed that for all patients untreated primary human hepatocytes rapidly respond to IFNα/Roferon stimulation, while primary human hepatocytes that were prestimulated with a high dose of IFNα/Roferon showed qualitatively little response. Interestingly, prestimulation with a low dose of IFNα/Roferon seemed to result in hypersensitization in primary human hepatocytes from patient 1, 3, 4, and 6, but not in the primary human hepatocytes from the other two patients.

To better quantify sensitization of the pathway, we first simulated with the calibrated mathematical model—as a proxy for the activation of an antiviral response—time courses for the amount of occupied GAS and ISRE promoter-bindings sites (active promoters of the antiviral genes) for different Roferon prestimulation doses followed by stimulation with 608 pM or no further stimulation. Second, we defined the absolute antiviral response as the area under the curve from 24 to 28 h after stimulation with 608 pM Roferon and subtracted by the area under the curve from 24 to 28 h with no

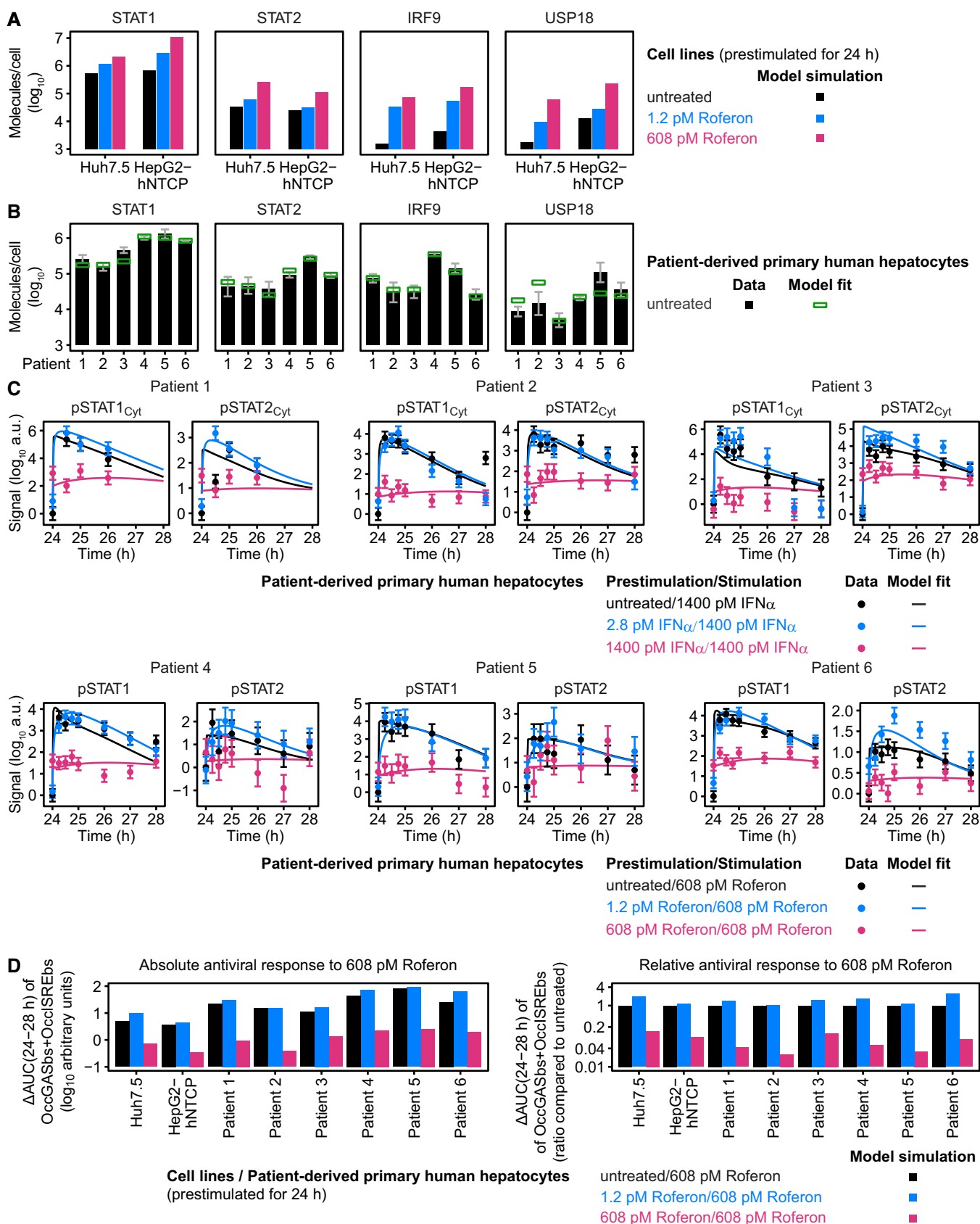

**Figure 6.**

**Figure 6. Cell type-specific basal protein amounts in relation to pathway desensitization in cell lines and primary human hepatocytes.**

A  The number of molecules per cell for STAT1, STAT2, IRF9, and USP18 were simulated by the mathematical model for untreated Huh7.5 and HepG2-hNTCP cells as well as for cells stimulated with 1.2 and 608 pM Roferon for 24 h.

B  The number of molecules per cell for STAT1, STAT2, IRF9, and USP18 were determined experimentally in primary human hepatocytes from six different patients. Cells were harvested without pretreatment, and different amounts of protein calibrators were spiked in total cell lysates and lysates were subjected to immunoblotting. Detection was performed with antibodies specific to STAT1, STAT2, IRF9, or USP18 using chemiluminescence on a CCD camera-based device (ImageQuant). Average of at least $N = 3$ are displayed with standard deviations. Data were used for calibration of a primary human hepatocyte-specific mathematical model, and estimated values are shown with green squares.

C  Experimental data and model fits of IFNα- and Roferon-induced phosphorylation of cytoplasmic or cellular STAT1 and STAT2 in growth factor-depleted primary human hepatocytes prestimulated with 0, 2.8 or 1,400 pM IFNα (patients 1–3) or 0, 1.2 or 608 pM Roferon (patients 4–6). Primary human hepatocytes from the same patients 1–6 as in (B) were used. Experimental data are represented by filled circles ($N = 1$ per patient). Experimental errors were estimated from the signal variance of the hepatocytes prestimulated with 1,400 pM IFNα. Lines indicate model fits.

D  Absolute and relative antiviral response is calculated by the mathematical model for each cell line and patient-derived primary human hepatocytes prestimulated with 0, 1.2, or 608 pM Roferon by simulating the amount of occupied GAS- and ISRE-binding sites upon stimulation with 608 pM Roferon. For the absolute antiviral response, the baseline-subtracted area under curve (ΔAUC) from 24 to 28 h is shown. For the relative antiviral response, the ΔAUC from 24 to 28 h is divided by the corresponding ΔAUC in cells without pretreatment.

stimulation (ΔAUC). To obtain the relative antiviral response, we divided the value of the absolute antiviral response obtained with prestimulated cells by the value of the response in untreated cells. A relative antiviral response of more than one corresponds to pathway hypersensitization, a value lower than one to pathway desensitization. We displayed the resulting values for the Huh7.5 and the HepG2-hNTCP cell lines as well as for the six patient-derived primary human hepatocytes (Fig 6D, left panel). These calculations showed that the extent of pathway hypersensitization upon prestimulation with 1.2 pM Roferon was rather small and variable. HepG2-hNTCP as well as primary human hepatocytes from patients 2 and 5 showed almost no pathway hypersensitization, while primary human hepatocytes from patients 1, 3, and 4 showed intermediate pathway hypersensitization. Interestingly, pathway hypersensitization was prominent in Huh7.5 cells and in primary human hepatocytes from patient 6. Further, the model simulations confirmed that pathway desensitization was rather pronounced in all cellular model systems and not only detected in Huh7.5 and HepG2-hNTCP cells, but also in primary human hepatocytes from all patients. However, the extent of pathway desensitization induced by 608 pM Roferon varied between primary human hepatocytes from different patients and was strongest in primary human hepatocytes from patients 1, 2, and 5 (Fig 6D, right panel). These results indicated the existence of a cell context- and patient-specific desensitization threshold that might be predictable by the amounts of pathway components.

To calculate the Roferon dose dependency of this cell context- and patient-specific pathway desensitization threshold, we plotted the relative antiviral response as a function of the prestimulation amount of Roferon (Fig 7A, left panel). Additionally, we calculated for the hepatoma cell lines and the primary human hepatocytes the Roferon prestimulation dose beyond which the pathway is desensitized, which we term desensitization threshold (Fig 7A, symbols). We again observed very different responses for the distinct cellular systems. While primary human hepatocytes from patients 2, 4, and 5 as well as HepG2-hNTCP showed pathway desensitization between 1–10 pM of Roferon and thus a low desensitization threshold, primary human hepatocytes from patient 3 and Huh7.5 cells showed a desensitization threshold that corresponds to two orders of magnitude higher Roferon doses. To determine which cellular factor is primarily responsible for this behavior, we plotted for each cellular system the desensitization threshold versus the basal amounts of STAT1, STAT2, IRF9 (Appendix Fig S11A), and USP18

(Fig 7A, right panel). For STAT2, only a weak anti-correlation was observed, while for STAT1 and IRF9 almost no correlation was observed (Appendix Fig S11A). However, we observed a linear anti-correlation between the amount of USP18 in untreated cells and the desensitization threshold, revealing that cells with low amounts of USP18 can tolerate a higher Roferon dose before showing desensitization of IFNα signal transduction.

To deduce which factors are predictive *in vivo* for the antiviral response to Roferon, we employed our mathematical model calibrated to primary human hepatocytes to establish a virtual patient cohort. The majority of model parameters was taken from the calibrated model, and these parameters were assumed to be fixed. Five model parameters were assumed to be patient-specific: the abundance of STAT1, STAT2, IRF9, and USP18 as well as the residual amounts of endogenous IFNα in the supernatant (i.e., blood plasma). To determine the patient-relevant IFNα levels and the patient-to-patient variability of IFNα, we analyzed plasma levels of IFNα in a cohort of 36 patients that were chronically infected with HBV (Fig 7B). IFNα levels ranged from 0.00186 to 0.152 pM, with a mean of $0.0440 \pm 0.0362$ pM (concentrations equal to $0.019 \pm 0.016$ pM Roferon). Means and variances of the abundance of STAT1, STAT2, IRF9, and USP18 were determined based on the measured amounts of molecules per cell in patient-derived primary human hepatocytes (Fig 6B) assuming a log-normal distribution. By combining these abundances with the residual IFNα concentration (corresponding to a prestimulation dose) measured in the blood plasma, we generated our virtual patient cohort consisting of 114 patients (Fig 7C). Model simulations were performed by randomly choosing patient-specific parameters assuming log-normal distributions for each of the five parameters. The patient-specific characteristics shown in Fig 7C were monitored after a simulation time of 24 h. For each virtual patient, we evaluated a stimulation with 608 pM Roferon and calculated the relative and absolute antiviral response. To determine which patient-specific factor is predictive for the extent of the antiviral response, we calculated correlations between the relative antiviral response and the cellular abundance of STAT1, STAT2, IRF9, and USP18 as well as the residual amount of Roferon (as a proxy for the prestimulation dose) (Fig 7D and Appendix Fig S11B). For STAT1 and STAT2, we observed a weak positive correlation between the cellular abundance of the respective protein and the relative antiviral response, while no correlation was observed for the abundance of IRF9 and USP18 as well as for

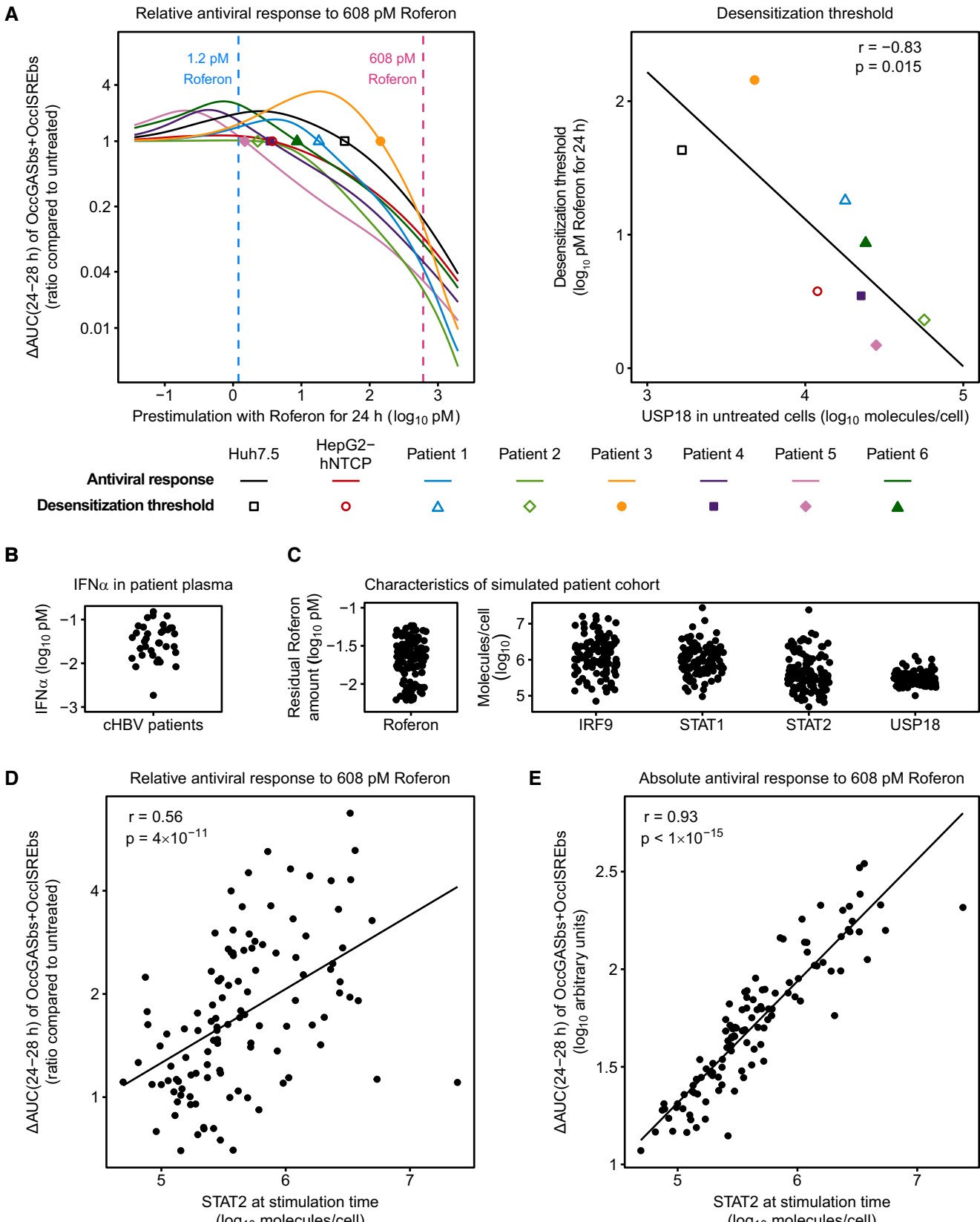

Figure 7.

Figure 7. Basal USP18 amounts determine cell type-specific desensitization threshold, while abundance of STAT2 predicts *in vivo* responsiveness to Roferon.

A   The relative antiviral response upon 608 pM Roferon stimulation (ΔAUC(24–28 h) of OccGASbs + OccISREbs compared to untreated) is shown as a function of the prestimulation dose for the cell lines Huh7.5 and HepG2-hNTCP as well as for primary human hepatocytes derived from six patients (left). Desensitization thresholds (i.e., the prestimulation dose beyond which the relative antiviral response is smaller than one) are indicated by symbols and plotted against the amount of USP18 in corresponding untreated cells (right). Spearman's rank order correlation coefficient (*r*) and *P*-value (*p*) are indicated.

B   Plasma levels of IFNα were measured in a cohort of 36 patients with chronic HBV.

C   A virtual cohort of 114 patients was simulated with the primary human hepatocyte-specific mathematical model. Distributions of IFNα, STAT2, STAT2, IRF9, and USP18 matching experimentally measured values as shown in Figs 6B and 7B are shown.

D   For each patient in the virtual patient cohort, a stimulation with 608 pM Roferon was simulated and the relative antiviral response (ΔAUC(24–28 h) of OccGASbs+OccISREbs compared to untreated) was calculated. The relative antiviral response was plotted against the cellular abundance of STAT2. Spearman's rank order correlation coefficient (*r*) and *P*-value (*p*) are indicated.

E   For each patient in the virtual patient cohort, a stimulation with 608 pM Roferon was simulated and the absolute antiviral response (ΔAUC(24–28 h) of OccGASbs + OccISREbs) was calculated. The absolute antiviral response was plotted against the cellular abundance of STAT2. Spearman's rank order correlation coefficient (*r*) and *P*-value (*p*) are indicated.

the residual amount of Roferon. Next, we calculated correlations between the absolute antiviral response and cellular components. Only a rather weak correlation of the abundance of STAT1, IRF9, and USP18 and no correlation of residual Roferon with the absolute antiviral response was visible (Appendix Fig S11C). However, interestingly, a very strong correlation of the absolute antiviral response with the abundance of STAT2 was observed (Fig 7E). Together, these results indicated that for the *in vivo* situation which was simulated by the patient cohort, the amount of STAT2 is predictive for the antiviral response stimulated by Roferon.

In conclusion, primary human hepatocytes isolated from six different patients showed striking heterogeneity in the abundance of STAT1, STAT2, IRF9, and USP18. The plasma samples from patients chronically infected with HBV revealed a high patient-to-patient variability of IFNα plasma levels. Model simulations revealed that the basal abundance of USP18 controls the threshold between hyper- and desensitization of IFNα signal transduction. By taking into account the measured heterogeneity in protein abundance of pathway components and IFNα, we identified STAT2 as a key predictor for the patient-specific Roferon-induced antiviral response.

## Discussion

Based on quantitative, time-resolved data, we showed that dependent on the IFNα dose prestimulation of hepatoma cell lines and primary human hepatocytes with IFNα leads to desensitization or hypersensitization of pathway activation and thereby the antiviral response. Based on these measurements, we established a mathematical model of IFNα signal transduction and unraveled that USP18 alone is not sufficient to establish pathway desensitization, but also requires the involvement of SOCS1. Our model-based analysis of patient-derived primary cells revealed that the patient-specific abundance of the feedback protein USP18 sets the threshold for pathway desensitization. By simulating the antiviral response upon treatment with different doses of IFNα in a virtual patient cohort, we identified the patient-specific STAT2 level as key predictor for the individual extent of the therapeutically inducible antiviral response.

The phenomenon of pathway sensitization was previously reported for interferon-induced signal transduction, but the underlying mechanisms were only partially understood. Hypersensitization of IFNγ-induced signal transduction as indicated by enhanced

pSTAT1 levels upon stimulation with interferon was observed in monocytes that were primed with IFNγ produced by PBMCs during macrophage development and stimulated with IFNα (Hu *et al*, 2002). Additionally, enhanced levels of transcription factor complexes (ISGF3 and GAF) were shown in IFNγ-prestimulated macrophages upon restimulation with IFNα (Lehtonen *et al*, 1997). So far, the majority of previously published studies focused on IFNα-induced pathway desensitization that is also referred to as ligand refractoriness. By treating fibroblasts with multiple rounds of IFNα, long-term ligand desensitization was observed for up to 3 days as assessed by the analysis of target gene expression. This pathway desensitization was dependent on protein synthesis since treatment with cycloheximide prevented long-term pathway desensitization (Larner *et al*, 1986). Hepatocytes isolated from mice receiving multiple injections of IFNα over time showed only a weak induction of pSTAT1 and reduced amounts of ISGF3 upon the second stimulation (Sarasin-Filipowicz *et al*, 2009). Furthermore, desensitization of IFNα-induced signal transduction can also be established in cells prestimulated with IFNβ, IFNλ1, or IFNλ4. For example, HLLR1-1.4 cells that were prestimulated with IFNβ- or IFNλ1 showed upon IFNα stimulation a reduced induction of pSTAT1, pSTAT2, pTYK2, and pJAK1 and of their target genes (Francois-Newton *et al*, 2011). Likewise, a reduced induction of pSTAT1 and reduced target gene expression upon IFNα stimulation was observed in Huh7 overexpressing IFNλ4, as well as in primary human hepatocytes with elevated IFNλ4 levels due to HCV infection (Sung *et al*, 2017). In U5A cells primed with IFNβ, a reduced induction of pSTAT1 upon stimulation with IFNα2 was detected (Wilmes *et al*, 2015). Interestingly, hepatocytes isolated from mice injected first with IFNβ and subsequently with IFNα showed reduced pSTAT1 levels and ISGF3 DNA binding, but mice injected first with IFNλ2, and later with IFNα only showed a very minor effect on pSTAT1 induction and ISGF3 DNA binding upon exposure to IFNα (Makowska *et al*, 2011). We showed in the presented work that prestimulation of Huh7.5 cells, HepG2-hNTCP cells, and primary human hepatocytes with both the research grade IFNα and the therapeutic agent Roferon results in IFNα signal transduction pathway sensitization. Distinct from previous reports, we provide evidence that prestimulation with IFNα is also able to establish hypersensitization of IFNα-induced signal transduction.

To unravel the mechanism which determines that prestimulation with a low dose of IFNα leads to pathway hypersensitization while prestimulation with a high dose of IFNα leads to pathway

desensitization, we established a model based on coupled ordinary differential equations, which comprises four positive and three negative feedbacks. Previously, Maiwald *et al* (2010) reported an IFNα signal transduction model that focused on the early antiviral response (up to 4 h) and identified IRF9 as a crucial positive feedback. In our model, we additionally incorporated two positive (STAT1 and STAT2) and three negative (SOCS1, SOCS3, and USP18) transcriptional feedbacks, as well as an IRF2-mediated feedback enhancing degradation of the *SOCS1* mRNA. While the positive feedbacks STAT1, STAT2, and IRF9 act at the cytoplasmic level by providing more proteins for formation of heterodimers and ISGF3 complexes, the negative feedbacks SOCS1, SOCS3, and USP18 operate at the receptor level via downregulation of the receptor. In contrast to other signal transduction models of the JAK/STAT pathway (Swameye *et al*, 2003; Bachmann *et al*, 2011), our model covers a comparatively wide time range (up to 32 h) in which two stimulations with a delay of 24 h are performed. Our analysis showed that the transcriptional feedbacks mediated by STAT1 and STAT2 only become relevant after approximately 10 h of ligand addition, while the other transcriptional regulators are already induced within the first 4 h.

Our model allowed us to predict the dynamics of the different IFNα-induced transcription factor complexes and to quantitatively link these to gene expression. IFNα-target genes in Huh7.5 cells can be classified into three groups: Genes that are primarily regulated by GAS sites (*IRF1* and *SOCS3*), genes that are primarily regulated by ISRE sites (*DDX58, HERC5,* and *IFI44L*) and genes that are regulated by both GAS and ISRE sites (*STAT1, STAT2, IRF9, IRF2, USP18,* and *SOCS1*). The connections between transcription factor complexes and mRNA expression of genes can be influenced by cell context-specific alterations in methylation patterns and differences in the activation of signal transduction pathways leading to an impact of additional transcription factors. In our study, we observed for most of the analyzed genes in Huh7.5 and HepG2-hNTCP cells a highly comparable expression dynamics upon stimulation with IFNα. *IRF1* was induced in both cell types as an immediate early gene within 1 h with an approximately 50-fold increase, but showed in HepG2-hNTCP cells a more sustained behavior. This prolonged expression indicates cell type-specific modulation of the *IRF1* mRNA expression, which is potentially due to an impact of the NFκB and MAP-kinase pathways as previously reported (Yarilina *et al*, 2008) or differences in epigenetic modifications present in HepG2-hNTCP cells versus Huh7.5 cells. In line with this hypothesis, we previously reported that cell type-specific differences in the expression dynamics of *SOCS3* can be explained by differences in promoter-binding elements and epigenetic modifications (Merkle *et al*, 2016).

Utilizing model reduction techniques (Maiwald *et al*, 2016), we showed that the data can be described sufficiently if we assume that the first gene group is induced by nuclear pSTAT1 homodimers, the second gene group is regulated by nuclear ISGF3 and the third gene group is also induced by nuclear ISGF3 and, to a lesser extent, by nuclear pSTAT1:pSTAT2 heterodimers. Interestingly, the mathematical model predicted that pSTAT1 homodimers form early after IFNα stimulation, whereas ISGF3 complexes form later, which is corroborated by the experimental evidence obtained by EMSA and co-immunoprecipitation experiments and explains the temporal order of target gene expression. For mathematical modeling approaches, it is of particular importance to combine model-guided experimental design (Kreutz & Timmer, 2009) with careful selection of experimental methods yielding informative data with favorable signal-to-noise ratio to accurately define dynamic behavior. For the detection of the dynamics of the formation of pSTAT1 homodimers in complex with GAS-binding regions, EMSA was most reliable as demonstrated by the small confidence intervals. Several independent EMSA experiments revealed that maximal binding activity is observed within 1 h of IFNα stimulation. However, to examine the dynamics of the formation of the large trimeric ISGF3 complex and the heterodimeric complexes, co-immunoprecipitation experiments were most robust yielding reproducible quantifications in independent experiments. Specifically, we employed co-immunoprecipitation experiments by performing STAT2 immunoprecipitations and quantifying co-immunoprecipitating pSTAT1 to verify the model-predicted dynamics of the sum of IFNα-induced formation of pSTAT1:pSTAT2 heterodimers and pSTAT1:pSTAT2:IRF9 trimers. We cannot exclude that non-canonical complexes such as pSTAT1:STAT2 (Ho *et al*, 2016) are also detected in these experiments. However, Ho *et al* showed that these complexes remain in the cytoplasm and are not relevant for type 1 interferon signaling. Further, since our experiments show that the IFNα-induced dynamics of STAT1 and STAT2 phosphorylation is very similar and this is also reflected by the trajectories of the mathematical model (Fig 3A), we assume that even if a hypothetical pSTAT1:STAT2 complex would form, it would have the same dynamics as the model-simulated pSTAT1:pSTAT2 complex.

The temporal order of INFα-induced formation of pSTAT1-containing transcription factor complexes is highly dependent on the ratio between the components STAT1, STAT2, and IRF9. For example, Huh7.5 cells that are not prestimulated with IFNα harbor per cell approximately $10^6$ molecules STAT1, while the abundance of IRF9 is 100-fold lower. Therefore, the initial amount of ISGF3 that forms early upon INFα stimulation is very limited and only after IRF9 is *de novo* transcribed and translated, ISGF3 can become the dominant transcription factor complex. In HepG2-hNTCP cells, which in comparison with Huh7.5 cells contain lower concentrations of STAT1 and STAT2 but higher amounts of IRF9, ISGF3 complexes can form earlier. Our analysis of primary human hepatocytes demonstrates that the ratio between STAT1, STAT2, and IRF9 varies substantially between different patient-derived samples, suggesting patient-specific dynamics in both ISGF3 formation and antiviral gene response.

Of note, we are not excluding other connections between transcription factors and gene expression, but they are apparently not necessary to describe our experimental data. It has previously been reported that the IFNα-inducible pSTAT1:pSTAT2 heterodimer preferentially binds to sequences that closely resemble GAS elements (Ghislain *et al*, 2001). Our experimental data are not contradicting this notion, as in our model, the pSTAT1:pSTAT2 heterodimer activates genes that are controlled by both GAS and ISRE sites. In our EMSA, we induced a supershift of GAS-binding complexes with antibodies against STAT1, but not against STAT2, thus excluding major contributions of the pSTAT1:pSTAT2 heterodimer to the dynamics of GAS-only regulated genes such as *IRF1* and *SOCS3*.

To the best of our knowledge, our mathematical model for the first time describes sensitization of the IFNα-induced signal transduction pathway quantitatively and provides a mechanistic

understanding of dose-dependent desensitization and hypersensitization of the IFNα pathway. This enabled us to show that the prestimulation dose determines whether the system is in a hypersensitized or a desensitized state and confirmed these findings by applying the model to measurements performed with a second ligand Roferon and another cell line HepG2-hNTCP.

The SOCS proteins play an important role in the negative regulation of signal transduction through cytokine receptors. The transcription of various SOCS proteins is upregulated upon cytokine stimulation, whereof specifically SOCS1 and SOCS3 are known to inhibit IFNα signal transduction (Krebs & Hilton, 2001). As revealed by crystal structure analysis, SOCS1 binds to the JAK1 kinase domain and the Elongin B/C adaptor complex and thereby acts as a direct inhibitor of the catalytic activity of JAK1, JAK2, and TYK2. It was also proposed that besides inhibiting downstream pathway activation, SOCS1 prevents JAK autophosphorylation and it was hypothesized that SOCS1 distinct from SOCS3 might not require receptor binding (Liau *et al*, 2018). In addition to their different mode of action, SOCS proteins also differ in their binding affinities for target proteins, rendering SOCS3 much less potent than SOCS1. We here showed that while *SOCS1* is expressed upon IFNα stimulation for more than 24 h and thereby contributes to long-term pathway desensitization, *SOCS3* expression is transient and returns to basal levels 8 h after IFNα stimulation. This rapid but transient GAS-dependent induction of SOCS3 mRNA is meditated by the early formation of pSTAT1 homodimers, while the GAS- and ISRE-dependent SOCS1 expression is controlled by late formation of pSTAT1:pSTA2 heterodimers and ISGF3 complex. Thus, we observed a temporal division of labor by the feedback proteins SOCS1 and SOCS3 in regulating IFNα signal transduction, reminiscent of the ligand dose-dependent division of labor observed for CIS and SOCS3 in Epo-induced JAK2/STAT5 signal transduction (Bachmann *et al*, 2011).

Besides these negative regulatory effects of the SOCS proteins, it was shown that USP18 inhibits IFNα signal transduction by binding to IFNAR2 and thereby inhibiting downstream substrate phosphorylation (Malakhova *et al*, 2006). A pivotal role of USP18 in desensitization of IFNα signal transduction was established based on knockdown, knockout, and overexpression experiments. It was observed that knockout of the USP18 gene in mouse embryonic fibroblasts and knockdown of USP18 expression in the human hepatoma cell line Huh7.5 lead to prolonged responsiveness to IFNα and enhanced antiviral efficacy against HCV (Randall *et al*, 2006). In complementary experiments, overexpression of the USP18 protein in HLLR1-1.4 cells led to a strong reduction in the levels of pSTAT1 and pSTAT2 upon stimulation with IFNα. Similarly, prestimulation with IFNβ resulted in the induction of USP18 protein and concomitantly prevented the induction of pSTAT1 as wells as pSTAT2 upon stimulation with IFNα (Francois-Newton *et al*, 2011). Further, overexpression of EGFP-USP18 under the control of the CMV promoter resulted in a reduction of the co-localization of HaloTag-IFNAR1 and SNAPf-IFNAR2c in U5A cells treated with IFNα (Wilmes *et al*, 2015). USP18$^{-/-}$ mice injected with IFNα displayed a higher amplitude of STAT1 phosphorylation in the liver compared to control mice, retained responsiveness to a second IFNα injection, and displayed elevated pSTAT1 levels at all examined time points (Sarasin-Filipowicz *et al*, 2009). Taken together, it was proposed that USP18 causes desensitization of IFNα signal transduction by

impairing the formation of functional IFNα-binding sites (Francois-Newton *et al*, 2011). In line with these findings, we observed in Huh7.5 cells that knockdown of USP18 by siRNA transfection in combination with IFNα prestimulation resulted in sustained phosphorylation levels of STAT1 and STAT2 as well as sustained expression of target genes over 24 h, thus supporting the role of USP18 as a negative regulator contributing to long-term desensitization of the IFNα signal transduction pathway. However, our experiments showed that cells that harbor elevated levels of USP18 due to overexpression do not present the same extent of pathway desensitization as parental cells prestimulated with IFNα. These results suggested that USP18 is necessary but not sufficient for desensitization of IFNα signal transduction. Based on our calibrated mathematical model, we proposed a mechanism whereby USP18 and SOCS1 act together on reducing the available receptor copy numbers on the cell surface, suggesting USP18 as a cofactor for SOCS1-mediated receptor degradation.

We have previously shown using Epo-induced AKT and ERK signal transduction as an example that the abundance of pathway components plays a crucial role in cell context-specific pathway regulation (Adlung *et al*, 2017). In the presented study, we observed major differences in the response to repeated IFNα stimulation between primary human hepatocytes isolated from individual patients and detected a high patient-to-patient variability in basal protein levels of the IFNα signal transduction pathway. The mathematical model predicted that the prevailing level of USP18 defines at which interferon dose the turning point between hyper- and desensitization of the pathway is located. It was previously proposed that the initial expression of USP18 serves as a predictor for sustained viral release in chronic HCV patients prior to receiving an interferon-based triple therapy. However, the predictive value of USP18 has been controversially discussed. It was suggested that high initial expression levels of USP18 in peripheral blood mononuclear cells correlate with a better response to the treatment (Frankova *et al*, 2015). On the other hand, the results of another study on chronically HCV-infected patients showed that the response to IFNα-based treatment was linked to cell-specific ISG activation patterns. The ISG baseline in the group of non-responders was lower in macrophages, but higher in hepatocytes, which might explain these discrepancies (Chen *et al*, 2010). We identified by simulating a virtual patient cohort based on measured patient-specific differences the protein abundance of STAT2 as the best predictor for the patient-specific antiviral response. The protein abundance of STAT1, correlated less with the patient-specific hypersensitization of the antiviral response. These results differ from pathway hypersensitization observed by Hu *et al* (2002) upon IFNγ prestimulation, showing that STAT1 is in their context the main transcription factor and accordingly the key component for pathway hypersensitization. In Huh7.5, HepG2-hNTCP and the six patient-derived primary human hepatocytes, the abundance of STAT1 is one order of magnitude higher than the abundance of STAT2, while in the monocytes used to investigate IFNγ-induced pathway hypersensitization STAT1 protein levels were close to the detection limit, revealing cell type-specific differences in the protein abundance of STAT1. In addition, IFNγ signal transduction relies on the pSTAT1 homodimer as the key transcription factor complex, while the amounts of STAT2 are crucial for the formation of the ISGF3 complex. By means of our mathematical model, uncertainties of predictions can be computed

based on the input data of the model (Kreutz *et al*, 2012; Kaschek *et al*, 2019). Of note, the findings derived from the simulation of the virtual patient cohort do not account for uncertainties of clinical measurements.

The experimental data and analyses by the mathematical model demonstrated that control of the IFNα-induced signal transduction is multifactorial and that the different components determine distinct aspects of the dynamics of pathway activation. The amount of IRF9 controls how fast ISGF3 complexes are formed and thereby the speed of the response, whereas the abundance of STAT2 determines how many ISGF3 complexes can be formed and herewith the extent of an antiviral response. As a consequence, USP18 determines pathway sensitization and STAT2 is a predictor of the patient-specific antiviral response.

The activation of the interferon pathway is an important response of the innate immune system to protect from viruses and tumors. In chronic hepatitis C virus (HCV) infection, it has previously been shown that patients with strong ISG-signatures have a lower chance of responding to treatment with IFNα, clearly demonstrating the *in vivo* relevance of interferon pathway desensitization (Chen *et al*, 2005). While interferon therapy is no longer relevant for chronic HCV infection due to the approval of highly effective direct-acting antivirals (Pawlotsky *et al*, 2015), it is an effective therapy for chronic HBV infection with regard to a functional cure, i.e., HBsAg loss (Lok *et al*, 2017). However, only a subset of patients achieves a sustained virological response after interferon therapy. In addition, interferon therapy is associated with considerable side effects. Thus, approaches and biomarkers that are helpful to identify patients that would benefit from interferon therapy are of great importance. Importantly, our findings might also be of relevance for other diseases. Indeed, it was recently shown that the interferon response activated by the PARP inhibitor Niraparib was able to potentiate the efficacy of checkpoint inhibitors used in cancer immunotherapy (Wang *et al*, 2019). Checkpoint inhibitors act by blocking inhibitory immune pathways and are used to treat various types of cancers (Le *et al*, 2015). However, response rates greatly differ between cancer etiologies and in patients with immune-inactive tumors with low heterogeneity of intratumor neo-antigens, responses are mostly poor (McGranahan *et al*, 2016). Thus, mathematical model-based interferon pathway modulation could help to sensitize such tumors to checkpoint blockade. Collectively, we show that the abundance of STAT2 and of USP18, in combination with information on patient-specific IFNα levels allow us to propose with our calibrated mathematical model an IFNα treatment regime to prevent pathway desensitization and to optimize an antiviral or anti-tumor response.

# Materials and Methods

## Reagents and Tools table

| Reagent/Resource | Reference or Source | Identifier or Catalog Number |
|---|---|---|
| **Experimental models** | | |
| Huh7.5 | Blight *et al* (2002) | |
| HepG2–hNTCP | Ni *et al* (2014) | |
| Primary human hepatocytes | University of Leipzig/University Hospital Heidelberg | |
| Plasma samples | University Hospital Freiburg | |
| Huh7/LucUbiNeo | Lohmann and Bartenschlager (2014) | |
| **Recombinant DNA** | | |
| Human USP18 cDNA | Genomics and Proteomics Core Facility of the DKFZ | https://www.ncbi.nlm.nih.gov/nuccore/AL136690 |
| pMOWS-TreT-puro | Pfeifer *et al* (2010) | |
| **Antibodies** | | |
| Phospho-Stat1 (Tyr701) | Cell Signaling | 9167 |
| Phospho-Stat2 (Tyr690) | Cell Signaling | 4441 |
| USP18 | Cell Signaling | 4813 |
| STAT1 | Merck | 06-501 |
| STAT2 | Merck | 06-502 |
| IRF9/ISGF3c | Becton Dickinson | 610285 |
| IRF9 | Abcam | ab126940 |
| IRF9 for IP | Cell Signaling | 76684 |
| Calnexin | Enzo Life Sciences | Adi-SPA-860 |
| HDAC1 | Santa Cruz | sc81598 |
| β-Actin | Sigma-Aldrich | A5441 |

**Reagent and Tools table** (continued)

| Reagent/Resource | Reference or Source | Identifier or Catalog Number |
|---|---|---|
| SOCS3 | Abcam | ab16030 |
| SOCS3 for IP | Merck | 04-004 |
| Goat anti-rabbit HRP | Dianova | 111-035-045 |
| Goat anti-mouse HRP | Dianova | 115-035-146 |
| Goat anti-Rabbit IRDye 800CW | LI-COR Biosciences | 926-32211 |
| **Oligonucleotides and other sequence-based reagents** | | |
| Primers for quantitative RT-PCR of human genes | This study | see Table 1 |
| ON-TARGETplus Human USP18 siRNA | Dharmacon (GE healthcare) | 11274 |
| ON-TARGETplus Non-targeting pool siRNA | Dharmacon (GE healthcare) | D-001810-10-20 |
| GAS-binding region of the IRF1 promoter for EMSA | This study | 5'-CATTTCGGGGAAATCAGGC-3' |
| **Chemicals, Enzymes, and other reagents** | | |
| Human IFN Alpha A (Alpha 2a) | PBL Assay Science | 11000-1 |
| Roferon-A | Roche | PZN 08543409 |
| Human IFN Gamma | R&D Systems | 285-IF-100 |
| Doxycycline | Clontech (Fisher Scientific) | NC0424034 |
| Protein G sepharose beads | GE Healthcare | 17-0618-01 |
| Protein A sepharose beads | GE Healthcare | 17-0780-01 |
| ECL Western Blotting Reagents | GE Healthcare | RPN2235/RPN2236 |
| Pierce BCA Protein Assay Kit | Thermo-Fisher | 23225 |
| RNeasy kit | Qiagen | 74136 |
| QIAshredder spin column | Qiagen | 79656 |
| High Capacity cDNA Reverse Transcription Kit | Applied Biosystems | 4368814 |
| Lipofectamine RNAiMax | Invitrogen | 13778100 |
| $[\gamma\text{-}^{32}P]$-adenosine triphosphate (3000 Ci/mmol) | PerkinElmer | BLU002H250UC |
| T4 polynucleotide kinase | New England Biolabs | M0201S |
| Illustra ProbeQuant G-50 Micro Columns | GE Healthcare | 28903408 |
| poly[dI-dC] | Sigma | 10108812001 |
| **Software** | | |
| GelInspector | Schilling *et al* (2005) | |
| R package dMod version 04 | Kaschek (2017); Kaschek *et al* (2019) | |
| R package blotit version 01 | Kaschek (2011) | |
| Promotor analysis Findpatterns of the GCG sequence analysis package using W2H | Senger *et al* (1998) | |
| Geneious v11.1 | https://www.geneious.com | |
| LightCycler480SW1.5.1 | Roche | |
| trust R package Version 0.1-7 | Geyer (2004) | |
| Image Quant TL | GE Healthcare | |
| **Other** | | |
| InfiniteF200Pro plate reader | Tecan | |
| Sonopuls | Bandelin | |
| ImageQuant LAS4000 | GE Healthcare | |
| Odyssey near-infrared fluorescence scanner | LI-COR Biosciences | |
| Nanodrop2000 | Thermo Scientific | |
| LigthCycler480 | Roche | |

## Methods and Protocols

### Cell culture, primary cells, and plasma samples

The human hepatocellular carcinoma cell lines Huh7.5 and HepG2-hNTCP were kindly provided by Ralf Bartenschlager, University of Heidelberg, Heidelberg, Germany and Stephan Urban, University Hospital Heidelberg, Heidelberg, Germany, respectively (Blight *et al*, 2002; Ni *et al*, 2014). Huh7.5 cells were cultivated in growth medium containing sodium pyruvate (Dulbecco's Modified Eagle Medium #31053 (Gibco), 10% FCS (Gibco), 1% GlutaMAX (Gibco), 1 mM Sodium pyruvate (Gibco), 100 U/ml penicillin∕streptomycin (Gibco)), while HepG2-hNTCP cells were cultivated in the absence of sodium pyruvate and using 1% Glutamin (Gibco) instead of GlutaMAX. Cell line authentication was performed using Multiplex Cell Authentication by Multiplexion (Heidelberg, Germany) as described (Castro *et al*, 2013). The SNP profiles matched known profiles or were unique. Purity of cell lines was validated using the Multiplex cell Contamination Test by Multiplexion (Heidelberg, Germany) as described (Schmitt & Pawlita, 2009). No Mycoplasma, SMRV, or interspecies contamination was detected.

Primary human hepatocytes were prepared at the University of Leipzig and at the University Hospital Heidelberg. Informed consent of the patients for the use of tissue for research purposes was obtained corresponding to the ethical guidelines of University of Leipzig and University Hospital Heidelberg, respectively. Tissue samples were acquired by partial hepatectomy and originate from healthy sections of resected liver tissue. Hepatocytes were isolated as described recently (Kegel *et al*, 2016), and hepatocytes were cultivated in adhesion medium (Williams' Medium E (Biochrom F1115), 10% FCS (Gibco), 0.1 μM dexamethasone, 0.1% insulin, 2 mM L-glutamine (Gibco), 100 U/ml penicillin∕streptomycin (Gibco)).

Plasma samples from patients with chronic HBV infection were obtained after informed consent at the University Hospital Freiburg (ethics committee approval number: 227/15). Plasma levels of IFNα were assessed by a custom multiplex assay (Eve Technologies Corp, Canada).

### siRNAs transfection, plasmids, and retroviral transduction

For knockdown studies, cells were transfected with 50 nM siRNA using Lipofectamine RNAiMax (Invitrogen) according to manufacturer's protocol. Cells were incubated with the siRNA complexes for 20 h. The following siRNAs of Dharmacon (GE healthcare) were used ON-TARGETplus Human USP18 (11274) siRNA—SMARTpool, and ON-TARGETplus Non-targeting pool.

*USP18* cDNA (https://www.ncbi.nlm.nih.gov/nuccore/AL136690) was provided by the Vector and Clone Repository of the Genomics and Proteomics Core Facility of the DKFZ and subsequently cloned in pMOWS-TreT-puro (Pfeifer *et al*, 2010) using MefI/EcoRI and BamHI restriction enzymes (NEB).

For stable transduction, 800,000 phoenix ampho cells were seeded in 6-well plates (TPP) and the next day co-transfected with 8 μg pMOWS-TreT-USP18-puro or pMOWS-TreT and 2 μg pMOWS-TAM2 using calcium phosphate precipitation in growth medium supplemented with 25 μM chloroquine.

After 8-h incubation, medium was replaced by growth medium and incubated overnight. The next day supernatant was harvested, filtered, and supplemented with 8 μg/ml polybrene. 1 ml of supernatant was added to 200,000 Huh7.5 cells seeded in 6-well plates

(TPP), which were transduced by centrifugation for 3 h at 340 × *g* at 37°C. Selection was performed using 0.75 μg/ml puromycin (Sigma), starting 48 h after transduction.

### Time course and sensitization experiments

600,000 Huh7.5 cells or 1,000,000 HepG2-hNTCP were seeded in 6-well plate format (TPP) 1 day in advance. Prior to stimulation, cells were washed three times with DPBS (Pan Biotech) and growth factor-depleted in starvation medium (Dulbecco's Modified Eagle Medium #31053 (Gibco), 1% (*v/v*) GlutaMAX (Gibco), 1 mM Sodium pyruvate (Gibco)) supplemented with 1 mg/ml BSA and 25 mM HEPES (Gibco) for 3 h (Huh7.5 cells). HepG2-hNTCP cells were growth factor-depleted overnight (15 h) in starvation medium without sodium puryvate and HEPES. The Huh7.5 cells were kept in 1.5 ml of the indicated medium, and the HepG2-hNTCP were cultivated in 1 ml of the medium described above. Due to these differences in media volume, the amount of Roferon added per ml was adjusted to ensure that the same total amount of Roferon was added per well. For co-immunoprecipitation experiments, 1.5 million Huh7.5 cells or 2.5 million HepG2-hNTCP were seeded on 60 mm tissue culture dishes (TPP) and kept in 3.7 ml of the respective medium. After growth factor depletion, cells were stimulated on a 37°C heating block by addition of interferon alpha 2a (PBL 11000-1), Roferon (Roche, PZN 08543409), or with recombinant human interferon gamma (R&D, 285-IF-100) and harvested at different time points. For sensitization experiments, cells were prestimulated with IFNα and stimulated 24 h later by addition of IFNα.

For primary human hepatocytes, 1 or 1.5 million viable cells were seeded in 6-well collagen-coated plates (Bio Coat, Corning) 1 day prior to the experiment. The next day, cells were gently washed twice with DPBS (PAN Biotech) before cells were growth factor-depleted for 3 h in starvation medium (Williams' Medium E (Biochrom F1115), 2 mM L-glutamine (Gibco), 100 U/ml penicillin∕streptomycin (Gibco)) prior to the experiment.

### Equipotent concentration of IFNα and Roferon

For model purposes, interferon concentrations given in units/ml and μg/ml were converted to nM based on information supplied by the datasheet (IFNα-2a) or product information (Roferon). In addition, the activity of Roferon (Roche) and IFNα-2a (PBL) were compared using the Huh7/LucUbiNeo cell line (Lohmann & Bartenschlager, 2014), a Huh7-Lunet cell line with a stably replicating HCV genotype 1b (Con1) subgenomic replicon under the selective pressure of G418 (0.5 mg/ml). The replicon contains a neomycin phosphatase as well as a firefly luciferase reporter gene instead of the viral structural genes and harbors replication-enhancing mutations in the nonstructural genes (Con1-ET). To perform the titration, 75,000 cells were seeded in 96 well plate format in the absence of G418 and the next day IFNα and Roferon was added in two-step serial dilutions. After 48 h, cells were lysed in luciferase lysis buffer (1% Triton-x 100, 25 mM glycyl-glycine pH 7.8, 15 mM MgSO$_4$, 4 mM EGTA pH 7.8, 10% glycerol, 1 mM DTT) and stored at −80°C. Luciferase activity was measured on a Mithras2 LB 943 (Berthold Technologies). Signal intensities were normalized to untreated cells and were fitted by four-parameter Hill kinetics to determine the IC$_{50}$ concentrations (Fig EV4A). Based on these data, equipotent concentrations for IFNα and Roferon were calculated (Fig EV4B).

### Cell lysis, immunoprecipitation and quantitative immunoblotting

Cellular fractionation was performed to obtain cytoplasmic and nuclear protein lysates. Lysis buffers were freshly supplemented with the protease inhibitors Aprotinin and AEBSF (Sigma). Cells were lysed in 250 µl cytoplasmic buffer (10 mM Hepes, 10 mM KCl, 0.1 mM EDTA, 0.1 mM EGTA, 1 mM NaF, 1 mM Na$_3$VO$_4$, 0.4% NP40) and gently scraped and transferred to Eppendorf tubes. Lysates were vortexed for 10 s and centrifuged at $1,000 \times g$, at 4°C for 5 min. Supernatants were transferred (cytoplasmic fraction) and pellets were washed with 250 µl washing buffer (10 mM Hepes, 10 mM KCl, 0.1 mM EDTA, 0.1 mM EGTA, 1 mM NaF, 1 mM Na$_3$VO$_4$,) and centrifuged at $1,000 \times g$, at 4°C for 5 min. Supernatants were discarded and 45 µl nuclear lysis buffer was added (20 mM Hepes, 25% Glycerin, 400 mM NaCl, 1 mM EDTA, 1 mM EGTA, 1 mM NaF, 1 mM Na$_3$VO$_4$, 0.4% NP40). Lysates were vortexed 10 s every 2 min for 15 min in total. Nuclear fraction was collected by taking the supernatant after 5 min centrifugation at $20\,817 \times g$ at 4°C.

Total cell lysates were prepared by lysing cells in $1 \times$ RIPA lysis buffer (1% NP40, 0.5% DOC, 0.1% SDS, 250 mM NaCl, 2.5 mM EDTA, 50 mM Tris pH 7.2). Cells were lysed in 250 µl lysis buffer, scraped, transferred to Eppendorf tubes, tumbled for 20 min at 4°C, and subjected to sonication (Sonopuls, Bandelin, for 30 s, with 75% amplitude, 0.1 s on 0.5 s off). Whole cell lysates were collected after 10 min centrifugation at 4°C at $20,817 \times g$.

The concentration of protein lysates was determined by Pierce™ BCA Protein Assay Kit (Thermo-Fisher) and measured on the InfiniteF200Pro plate reader (Tecan). 10 or 20 µg samples were prepared for quantitative immunoblotting. SOCS3 was immunoprecipitated overnight with SOCS3 antibody (Merck) and protein G sepharose beads (GE Healthcare) supplemented with 0.1 ng SBP-SOCS3. For co-immunoprecipitation experiments, 650 µg or 1,200 µg of total cell lysates were incubated overnight with STAT2 antibody (Merck) or IRF9 antibody (Cell Signaling) and protein A sepharose beads (GE Healthcare). Samples were loaded in randomized order to avoid correlated blotting errors (Schilling et al, 2005). Blots were developed using self-produced ECL solutions (Solution 1: 0.1 M Tris pH 8.5, 2.5 mM Luminol, 0.4 mM p-coumaric acid; Solution 2: 0.1 M Tris pH 8.5, 0.018% H$_2$O$_2$) or using ECL Western Blotting Reagents (GE healthcare) on the CCD camera-based ImageQuant LAS4000 (GE Healthcare). To remove previous antibody signals, HRP groups were quenched with H$_2$O$_2$ as described previously (Sennepin et al, 2009) or antibodies were removed by incubation with stripping buffer (0.063 M Tris pH6.8, 2% SDS, 0.7% β-mercaptoethanol) at 65°C. Bands were quantified using ImageQuant software (GE healthcare).

For fluorescence-based detection (Appendix Fig S1), blots were developed using the Odyssey near-infrared fluorescence scanner (LI-COR Biosciences) using a secondary antibody coupled to IRDye 800CW near-infrared dye.

The following antibodies were used Phospho-Stat1 (Tyr701) (58D6) #9167; Phospho-Stat2 (Tyr690) #4441; USP18 (D4E7) #4813, all from Cell Signaling. STAT1, CT #06-501; STAT2, CT #06-502 both from Merck. IRF9/ISGF3c #610285 (Bectin Dickenson), Calnexin #Adi-SPA-860 (Enzo Life Sciences), HDAC1 (10E2) #sc81598 (Santa Cruz), β-Actin #A5441 (Sigma-Aldrich), and SOCS3 #ab16030 (Abcam). IP was performed using the following antibodies SOCS3 #04-004 (1B2), STAT2 CT #06-502, both from Merck, and IRF9 #76684 (D2T8M) from Cell Signaling. Secondary antibodies include rabbit and mouse specific antibodies raised in goat, coupled to HRP (Dianova) and Goat anti-Rabbit IRDye 800CW #926-32211 (LI-COR Biosciences).

### Number of molecules per cell using recombinant proteins

cDNA of human USP18 (https://www.ncbi.nlm.nih.gov/nuccore/AL136690) was provided by Genomics and Proteomics Core Facility of the DKFZ and subsequently cloned in pGEX2T vector (GE Healthcare) using BamHI and EcoRI/MfeI (NEB) restriction sites.

Recombinant proteins were produced upon IPTG addition in transformed BL21-CodonPlus(DE3)-RIL competent bacteria (Agilent) and were purified using GST or SBP isolation as described previously (Raia et al, 2011).

SBP-SOCS3 calibrator was kindly provided by Anja Zeilfelder (Klingmüller lab, DKFZ, Heidelberg). SBP-STAT1ΔN, SBP-STAT2ΔN, and GST-IRF9 were established previously in our laboratory (Maiwald et al, 2010).

Concentration of calibrators was determined using BSA protein standard (Pierce) on SDS-PAGE gel stained with SimplyBlue Safe stain (Invitrogen) according to manufacturer's protocol.

To determine molecules per cell, the cell number of Huh7.5 or HepG2-hNTCP was counted with the Neubauer improved counting chamber for each treated condition. For primary human hepatocytes from Patients 1–3, a dilution curve was established, which was based on protein concentrations derived from different amounts of cells lysed in 250 µl $1 \times$ RIPA lysis buffer. For primary human hepatocytes from Patient 4–6, cells were fixed with paraformaldehyde and nuclei were stained with DAPI. 16 images per condition were taken using a Motorized Widefield Microscope (Zeiss Cell Observer). Using Fiji software, the respective number of cells per dish was quantified.

Different amounts of calibrators were spiked in 10 µg whole-cell lysates and quantitative immunoblotting was performed with the indicated antibodies. The linear regions of the calibration curves were fitted with a linear regression model in R, and the amount of endogenous signal was interpolated. Uncertainties were computed as standard error of the mean of the different samples assuming lognormally distributed signals.

### Quantitative RT–PCR

Total RNA was extracted using RNeasy kit (Qiagen) according to manufacturer's instruction. Clearing of the lysates was achieved using QIAshredder spin column (Qiagen). RNA concentrations were determined by absorbance (Nanodrop2000, Thermo Scientific), and reverse transcription was performed with 1 µg of RNA in 20 µl according to manufacturer's instruction (High Capacity cDNA Reverse Transcription Kit from Applied Biosystems). Quantitative PCR was performed on the LigthCycler480 (Roche) using primers and dual hybridization probes in 2× Probes Master (Roche). Cycling protocol consisted of 5 min pre-incubation at 95°C, 50 amplification cycles (95°C for 10 s, 60°C for 30 sec and acquisition at 72°C for 1 s), and 2 min cooling. Quantification cycles (Cq) were determined by absolute quantification with second derivative maximum method using the software LightCycler480SW1.5.1. Samples for calibration curve were included in each measurement to assess efficiency of primer hybridization.

**Table 1. Primers for quantitative RT-PCR of human genes.**

| Genes | Accession # | Primer fwd | Primer rev | UPL Probe # |
|---|---|---|---|---|
| *CXCL10* | NM_001565.3 | gaaagcagttagcaaggaaaggt | gacatatactccatgtagggaagtga | 34 |
| *CXCL11* | NM_005409.3 | agtgtgaagggcatggcta | tcttttgaacatggggaagc | 76 |
| *DDX58* | NM_014314.3 | tgtgggcaatgtcatcaaaa | gaagcacttgctacctcttgc | 06 |
| *GAPDH* | NM_002046 | agccacatcgctcagacac | gcccaatacgaccaaatcc | 60 |
| *HPRT* | NM_000194.2 | cgagcaagacgttcagtcct | tgaccttgatttattttgcatacc | 73 |
| *HERC5* | NM_016323.3 | cttccagtgaaagtatcatcaagtg | ccagagcaaaatgctttgatt | 67 |
| *IFI44L* | NM_006820.3 | tgacactatggggctagatgg | ttggtttacgggaattaaactgat | 15 |
| *IFI6* | NM_002038.3 | gggctccgtcactagacctt | aaccgtttactcgctgctgt | 40 |
| *IFIT2* | NM_001547.4 | tggtggcagaagaggaagat | gtaggctgctctccaaggaa | 27 |
| *IFITM3* | NM_021034.2 | gatgtggatcacggtggac | agatgctcaaggaggagcac | 76 |
| *IRF1* | NM_002198.1 | ttggccttccacgtcttg | gagctgggccattcacac | 36 |
| *IRF2* | NM_002199.3 | tgaagtggatagtacggtgaaca | cggattggtgacaatctcttg | 56 |
| *IRF9* | NM_006084.4 | aactgcccactctccacttg | agcctggacagcaactcag | 77 |
| *ISG15* | NM_005101.3 | ggcttgaggccgtactcc | ctgttctggctgaccttcg | 24 |
| *ISG56* | NM_001548.3 | gctccagactatccttgacctg | agaacggctgcctaatttacag | 9 |
| *MX1* | NM_30817.1 | gagctgttctcctgcacctc | ctcccactccctgaaatctg | 42 |
| *NMI* | NM_004688.2 | ttcaggcgctgctgtttt | tgtgtcatcttttatcagcttcca | 24 |
| *EIF2AK2* | NM_002759.1 | cggtatgtattaagttcctccatga | gacaaagcttccaaccagga | 62 |
| *SOCS1* | NM_003745.1 | gccccttctgtaggatggta | ctgctgtggagactgcattg | 87 |
| *SOCS3* | NM_003955.3 | tctccttcaattcctcagcttc | gttcagcattcccgaagtgt | 13 |
| *STAT1* | NM_007315.3 | tgagttgatttctgtgtctgaagtt | acacctcgtcaaactcctcag | 32 |
| *STAT2* | NM_005419.2 | ggaacagctggagacatggt | tcctgatagctaaccaggcaac | 17 |
| *TBP* | NM_003194.3 | cggctgtttaacttcgcttc | cacacgccaagaaacagtga | 3 |
| *TRIM21* | NM_003141.3 | tgagcggaaactgaaagtga | tggagacctttagggggttt | 24 |
| *USP18* | NM_017414.3 | tatgtgagccaggcacgat | tcccgacgtggaactcag | 75 |
| *ZNFX1* | NM_021035.2 | tcgctggcagctttatagg | tggcgttcatagctgaggat | 64 |

Primers are listed in 5′- to 3′-end orientation.

Data were normalized with the geometric mean of the reference genes hypoxanthine-guanine phosphoribosyltransferase (*HPRT*), TATA box-binding protein (*TBP*), and glyceraldehyde-3-phosphate dehydrogenase (*GAPDH*).

Primers were designed using the Universal Probe Library Assay Design Center (Roche Applied Biosciences) and manufactured by Eurofins. The utilized UPL probes and primers sequences for human genes are listed in Table 1.

### Transcription factor binding site identification

Promoter analysis of the following human genes was performed of a 3,000 bp region consisting of 1,250 bp in front and 1,750 bp inside the corresponding gene: *IRF1* (>NC_000005.10:c132492039-13248 9039 *Homo sapiens* chromosome 5, GRCh38.p13 Primary Assembly); *SOCS3* (NC_000017.11:78358329-78361329 *Homo sapiens* chromosome 17, GRCh38.p13 Primary Assembly); *DDX58* (>NC_0000 09.12:c32527446-32524446 *Homo sapiens* chromosome 9, GRCh 38.p13 Primary Assembly); *HERC5* (>NC_000004.12:88455354-88458354 *Homo sapiens* chromosome 4, GRCh38.p13 Primary Assembly); and *IFI44L* (>NC_000001.11:78619137-78622137 *Homo*

*sapiens* chromosome 1, GRCh38.p13 Primary Assembly). The promoter analysis was performed using Findpatterns of the GCG sequence analysis package using W2H (Senger *et al*, 1998). The following consensus sequences were tested: GAS (gamma-activated sequence) TTNCNNNAA (Darnell *et al*, 1994); and ISRE (interferon-stimulated response element) TTTCNNTTYY (Dill *et al*, 2014). Gene annotation is displayed using Geneious v11.1 (Kearse *et al*, 2012).

### Electrophoretic mobility shift assay (EMSA)

Electrophoretic mobility shift assays were performed with nuclear protein lysates obtained from cellular fractionation. Oligonucleotides probes used in the assay were synthesized (Sigma) harboring the GAS-binding region of the human IRF1 promoter (5′-CATTTCGGGGAAATCAGGC-3′). After annealing, oligonucleotides were end-labeled with [γ-$^{32}$P]-adenosine triphosphate (3,000 Ci/mmol; PerkinElmer) using the T4 polynucleotide kinase (New England Biolabs) and further purified by gel filtration on Illustra ProbeQuant G-50 Micro Columns (GE Healthcare). Nuclear lysates (5 μg) were normalized in 19-μl reaction mixtures containing 20 mM HEPES, pH 7.9; 1 mM DTT; 0.1 mM

EDTA; 50 mM KCl; 1 mM MgCl$_2$; 5% glycerol; 200 μg/ml BSA; and 1 μg poly[dI-dC] (Sigma). The reaction was placed on ice for 20 min before incubating with $^{32}$P-labeled oligonucleotide probes (10,000 cpm) for 20 min at room temperature. For the supershift samples, the lysate-DNA mixture was incubated with 2 μg of antibodies recognizing either STAT1 (Merck, CT #06-501), STAT2 (Merck, CT #06-502), or IRF9 (Abcam, ab126940) for 10 min at room temperature. Samples were resolved on a 4.5% native polyacrylamide gel (37.5:1) in a 0.5 × Tris-borate-EDTA (TBE) buffer for 6 h at 130 V at 4°C. Gels were dried for 60 min at 80°C and visualized by PhosphorImager.

### Data processing and estimation of uncertainties

Immunoblot data were normalized to housekeepers Calnexin, Actin, HDAC1, or recombinant SBP-SOCS3 using GelInspector (Schilling et al, 2005). For each target, data points were scaled together by means of the R package blotIt (Kaschek, 2011) assuming log-normally distributed signals. Independent experiments that contained more than three overlapping data points with 1,400 pM IFNα treatment were used as a reference for scaling. Different scaling factors for each gel as well as time course values used for parameter estimation were simultaneously determined by generalized least squares estimation, with data points assumed to be log-normally distributed. Uncertainties correspond to 68%-confidence intervals of the estimated data points. For the control model (Huh7.5), 10,945 single data points (Dataset EV1) were scaled by blotIt to obtain 1,902 data points with confidence intervals that served for calibration of the model (Dataset EV2). In addition, the 16 determined amounts of molecules per cell of the feedback proteins were utilized for model calibration. For the validation of the control model with EMSA and protein data (Figs 4B and EV3D), 134 single data points (Dataset EV3) were scaled to obtain 45 data points with confidence intervals (Dataset EV4). For the validation of the control model with qRT-PCR data (Fig 4C), 315 single data points (Dataset EV3) were averaged to obtain 105 data points with standard error of the mean (Dataset EV4). For the Roferon model (Fig 4D), 891 single data points (Dataset EV7) were scaled to obtain 221 data points with confidence intervals (Dataset EV8) and for the HepG2-hNTCP model, 1,274 single data points (Dataset EV9) were scaled to 305 data points with confidence intervals (Dataset EV10). For the validation of the HepG2-hNTCP model with qRT-PCR data (Appendix Fig S7C), 252 single data points (Dataset EV11) were averaged to obtain 84 data points with standard error of the mean (Dataset EV12). For patient-derived primary human hepatocytes, no biological replicates were available. Here, experimental errors were estimated from the signal variance of the hepatocytes prestimulated with 1,400 pM IFNα, assuming that the corresponding underlying time course stays constant after stimulation (Dataset EV13).

### Parameter estimation and model development for the Huh7.5 cell line

The modeling process was performed by means of the R package dMod (Kaschek, 2017; Kaschek et al, 2019). In total, the mathematical model consists of 41 species and 75 reactions that were derived by the law of mass-action and Michaelis–Menten kinetics. The reactions are justified based on published literature (Appendix Table S1). Observables were computed with respect to model states as indicated in Appendix Table S2. Parameter values of the global

optimum for the Huh7.5 core model and profile likelihood-based confidence intervals are shown in Appendix Table S3. Parameters were log-transformed to ensure positivity and enable optimization over a broad range of magnitudes (Raue et al, 2013). Calculation of analytical steady-state expressions (Rosenblatt et al, 2016) and application of model reduction for ODE models (Maiwald et al, 2010) was incorporated by a set of parameter transformations (Appendix Table S4). In some cases during the model simplification procedure, parameter values were fixed instead of changing the model structure to keep a biological meaningful model structure. Parameters were estimated by the method of maximum likelihood performing a deterministic multi-start optimization of 1,000 randomly chosen parameter sets by means of the trust region optimizer trust (Geyer, 2004). Parameter values were not restricted by fixed borders. Instead, in order to prevent the optimizer from finding solutions with very low or high parameter values, we constrained the model parameters with a weak $L_2$ prior that contributed with one to the likelihood, if the parameter differed by five orders of magnitude from 1. When computing the profile likelihood (Appendix Fig S12), these $L_2$ priors were substracted in order to ensure an exclusively data-based identifiability analysis. To show reliability of the optimization, the 200 best optimization runs were displayed as a waterfall plot sorted by their objective values (Appendix Fig S13; Raue et al, 2013). Different local optima were found multiple times, and the global optimum was found in 18 of the 200 cases. To test identifiability of the parameters and to calculate confidence levels for the estimated values, the profile likelihood (Raue et al, 2009) was computed for each parameter. In total, 12 initial values, 17 scaling and offset parameters and 56 dynamical parameters were estimated for the Huh7.5 model. Profile likelihoods (Appendix Fig S12) showed finite confidence intervals for 74 out of 85 parameters. From the remaining parameters, three showed confidence intervals open to minus infinity and eight were open to plus infinity. However, due to the biological significance of the parameters and to ensure that the model selection analysis and the application of the model to different cell systems by $L_1$ regularization was performed without additional constraints, no further model reduction was applied. Estimated parameter values with corresponding confidence intervals and the resulting model parameters obtained after transformation are summarized in Appendix Table S4. For the receptor model, different structures were evaluated by means of the Bayesian information criterion (Schwarz, 1978).

### Application of the Huh7.5 model to HepG2-hNTCP and primary human hepatocytes

To analyze experimental measurements performed with the Roferon, the calibrated Huh7.5 model (control model) was utilized with all parameters being fixed except for scaling and offset parameters as well as the binding affinity of the ligand (parameter BindIFN) that were re-estimated by the method of maximum likelihood.

For analysis of the HepG2-hNTCP data, the previously estimated binding affinity for Roferon was fixed. Scaling and offset parameters as well as parameters corresponding to the molecules per cell in the system, i.e., totSTAT1, totSTAT2, totIRF9, and synthUSP18, were re-estimated. The four parameters, synthUSP18mRNAbasal_OE, synthUSP18_inh, synthUSP18mRNAbasal_inh, and synthUSP18mRNA_inh, that were used for incorporation of inhibitor and overexpression conditions were not considered. The remaining 60

parameters of the model were estimated with an additional $L_1$ constraint as previously described (Merkle *et al*, 2016). In our case, a combination of $L_1$ and $L_2$ prior, i.e., an elastic net (Zou & Hastie, 2005), was applied that penalizes least when parameters take the exact value as in the control model. Penalization strength was chosen such that $L_1$ and $L_2$ prior contribute to the same extent if the parameter value differed by one order of magnitude to the control model. Optimization was performed by means of a trust region optimizer which was adapted to $L_1$ regularization as part of the dynamic modeling framework *dMod*. For each of 25 different regularization strengths, we performed 200 optimization runs starting from randomly chosen parameter sets. The 5,000 resulting fits were evaluated by means of a combination of goodness-of-fit and number of parameters being different from the control model. Based on Bayesian information criterion, we defined the objective value $4 \times \ln(n) \times k - 2 \times \ln(L)$, where $n$ is the number of data points, $k$ the number of parameters different from the control model and $L$ the value of the likelihood function. Compared to classical BIC, our definition favors models with low amounts of necessary $L_1$ parameters. The resulting parameter estimates are summarized in Appendix Tables S5 and S6. For the HepG2-hNTCP model, we obtained two out of the 60 $L_1$ parameters (indicated by ratio_parameter) to be different from Huh7.5.

For the analysis of primary human hepatocyte data, we performed $L_1$ regularization similar to the case of HepG2-hNTCP. For each of six patients, parameters defining the number of molecules per cell of the proteins as well as scaling and offset parameters were re-estimated. The remaining primary human hepatocyte-specific parameters were assumed to be the same for all patients and were estimated by means of $L_1$ regularization (see Appendix Table S7 for parameter estimates). For the primary human hepatocyte model, we obtained five out of 60 $L_1$ parameters to be different from Huh7.5 cells. The gene-specific parameters that were estimated to link the occupied binding sites predicted by the mathematical model to the respective target genes (Fig 4C) are listed in Table S8.

## Data availability

The model structure and equations can be found in Model EV1 (Model_EV1.xml, exemplified for Huh7.5 cells stimulated with 1,400 pM IFNα. This model was deposited in BioModels (Chelliah *et al*, 2015) and assigned the identifier MODEL2005110001 (https://www.ebi.ac.uk/biomodels/MODEL2005110001).

The raw data files from quantitative immunoblots and quantitative RT-PCR data before and after processing with blotIt have been provided as Datasets EV1–EV13.

**Expanded View** for this article is available online.

## Acknowledgements
The authors thank the Genomics and Proteomics Core Facility of the German Cancer Research Center (DKFZ), for providing cDNA of USP18. Furthermore, we would like to thank Agnes Hotz-Wagenblatt for excellent assistance with the GCG sequence analysis package using W2H for SOCS3 promoter analysis. Furthermore, we would like to thank Anja Zeilfelder for providing the SBP-SOCS3 calibrator and Luisa Schwarzmüller for generation of the SBML model file. The human hepatocellular carcinoma cell lines, Huh7.5 and HepG2-hNTCP, were kindly provided by Ralf Bartenschlager and Stephan Urban, respectively. The authors thank Raphael Engesser, Helge Hass, Daniel Kaschek, Christian Tönsing, Clemens Kreutz, and Joep Vanlier for fruitful discussions during the development of the mathematical model. This work was funded by the Deutsche Forschungsgemeinschaft (DFG, German Research Foundation)—Project number 272983813—TRR 179. This work was also supported by the DFG within FOR1202 and within the Germany's Excellence Strategy (CIBSS—EXC-2189—Project ID 390939984) and by the German Ministry of Education and Research (BMBF) within "Multi-Scale Modeling of Drug Induced Liver Injury" (MS_DILI, 031L0074A, 031L0074B), within the Liver Systems Medicine network (LiSyM, 031L0042, 031L0048) and within the EraSysAPP consortium IMOMESIC (031A604A, 031A604B, 031A604C). We further acknowledge support by the Ministerium für Wissenschaft, Forschung und Kunst Baden-Württemberg (MWK) through bwHPC.

## Author contributions
Experimental methods: FK; MT. Main experiments: FK; MT. Mathematical model: MR; TM. Model calibration and analysis: MR. Experiments with Roferon in Huh7.5 and determination of number of molecules per cell of primary human hepatocytes: TN. Electrophoretic mobility shift assays: VGM. Determination of equipotent doses of IFNα and Roferon: CD. Preparation of primary human hepatocytes from patients: AV; MW; ST; KH; GD; DS. Measuring IFNα levels in cHBV patients: TB. Project design: MB; JT; MS; UK. Manuscript writing: FK; MR; MT; MS; UK. Preparation of figures: FK; MR; MT; MS; UK. Manuscript revision and editing: TB; MB; JT; MS; UK. Project funding: MB; JT; UK. Approval of manuscript: all authors.

## Conflict of interest
The authors declare that they have no conflict of interest.

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
