## [Review Process File · Molecular Systems Biology]

Disentangling molecular mechanisms regulating sensitization of interferon alpha signal transduction

Frédérique Kok, Marcus Rosenblatt, Melissa Teusel, Tamar Nizharadze, Vladimir Gonçalves Magalhães, Christopher Dächert, Tim Maiwald, Artyom Vlasov, Marvin Wäsch, Silvana Tyufekchieva, Katrin Hoffmann, Georg Damm, Daniel Seehofer, Tobias Boettler, Marco Binder, Jens Timmer, Marcel Schilling, and Ursula Klingmueller

DOI: 10.15252/msb.20198955

Corresponding author(s): Ursula Klingmueller (u.klingmueller@dkfz.de) , Jens Timmer (jeti@fdm.uni-freiburg.de), Marcel Schilling (m.schilling@dkfz.de)

Review Timeline:

Submission Date:	18th Apr 19
Editorial Decision:	12th Jun 19
Revision Received:	25th Oct 19
Editorial Decision:	10th Dec 19
Revision Received:	20th Mar 20
Editorial Decision:	11th May 20
Revision Received:	29th May 20
Accepted:	16th Jun 20

Editor: Maria Polychronidou

Transaction Report:

Manuscript Number: MSB-19-8955

Title: Disentangling molecular mechanisms regulating sensitization of interferon alpha signal transduction

Thank you again for submitting your work to Molecular Systems Biology. I apologize for the delay in getting back to you with a decision on your manuscript. Unfortunately, after a series of reminders, reviewer #3 communicated to us that they are not going to be able to return a report. In the interest of time, we have decided to proceed with making a decision based on the reports of reviewers #1 and #2. As you will see below, the reviewers acknowledge that the study could be a potentially useful contribution to the field. They raise however a series of concerns, which preclude the publication of your study in its current form.

As you will see below, reviewer #1 expresses strong concerns regarding the conceptual novelty of the study. During our pre-decision cross-commenting process (in which the reviewers are given the chance to make additional comments, including on each other's reports), reviewer #2 pointed out that in their opinion the level of conceptual novelty is not disqualifying criticism. In particular, s/he emphasized that while indeed there is extensive literature on the topic, the value of the study is two fold: first, there is great value in codifying mechanistic knowledge and providing it in the form of a model that allows a field to move forward and second, the conceptual advance in the dimer-switching regulation of Stat1 and ISGF3 activities is not insubstantial (since there have been suggestions of this in the literature, but it is not sorted out). As such, we have decided to offer you a chance to perform a major revision and address the issues raised by the reviewers.

In summary, both reviewers think that further experimental analyses are required in order to better support the main conclusions. In particular they both share concerns regarding the evidence provided to support dimer-switching and the specificity of the feedback/feedforward loops and think that more direct biochemical evidence is needed. We acknowledge that these revisions involve substantial additional experimental work, so we can discuss an extension of the 90 days revision deadline if needed.

All other issues raised by the reviewers need to be satisfactorily addressed. As you may already know, our editorial policy allows in principle a single round of major revision so it is essential to provide responses to the reviewers' comments that are as complete as possible. Please feel free to contact me in case you would like to discuss in further detail any of the issues raised by the reviewers.

REFeree REPORTS

Reviewer #1:

Kok et al. present a copious amount of experimental data and an accompanying first mathematical model describing the signaling by type-1 interferon (IFN) cytokines. This includes experiments using

cancer patient-derived primary hepatocytes and the measurement of interferon concentrations in the plasma of patients chronically infected with hepatitis-b virus.

While the amount of work that went into this study is impressive, the overall conceptual advance is less so. The idea that pre-stimulation of cells with low IFN concentrations (also called priming) sensitizes cells to a subsequent IFN bolus is well established and hardly novel. There is an established literature that links this phenomenon to the upregulation of STAT transcription factors, which are themselves IFN-regulated gene products. For example, papers including Matikainen et al (1997) *Cell Growth Diff* 8, 687, and Melen et al (2000) *J Hepatol* 33, 764, make this point to different extents. The authors' conclusion that STAT2 protein concentration is a critical determinant of IFN sensitivity likewise is hardly surprising, given that this protein is the quintessential and indispensable type-1 IFN mediator.

My main criticism concerns the definitions underlying the mathematical model and some of the central conclusion derived thereof. The authors' cell-based experiments show that priming of cells with low and high IFN concentrations leads to sensitization and de-sensitization of the cells to a subsequent high-dose stimulus, respectively. The authors use their model to understand why this might come about. The model predicts that stimulation with IFN of unprimed cells initially results in STAT1 homodimers, followed by STAT1-STAT2 heterodimers until after about 4 hours ISGF3 becomes the major transcription factor. The model further predicts that the amounts of STAT1 homodimers, STAT1-STAT2 heterodimers and ISGF3 would depend upon the priming IFN concentration, and that this would accordingly determine the repertoire of genes induced, including the positive feedback proteins (STAT1, STAT2, IRF9) and negative feedback proteins (USP18, SOCS1, IRF2) (page 6, top). However, with the exception of IRF2, all feedback proteins considered are induced by both type-1 IFN-induced transcription factors, ISGF3 and STAT1 homodimers. Thus, a shift from predominantly STAT1 homodimer-regulated genes to ISGF3-regulated genes would be expected to have little consequences as far as gene expression is concerned. Nonetheless, a central prediction of their model is the occurrence of the different transcription factors in a temporal order that can be modulated by the priming conditions (page 9, top). Yet, there is no experimental support for this at all. The authors claim in the discussion that this data could not be measured directly (page 16, middle). However, this claim is unconvincing. The authors make no attempt to study the consecutive occurrence of the different transcription factors (eg via DNA binding kinetics), although it apparently takes several hours until there is enough IRF9 protein to assemble significant amounts of ISGF3 complex such that it becomes the dominant transcription factor. They resort to an indirect way to assess this, namely through studying SOCS3 expression, a gene that is induced by STAT1 homodimers but not ISGF3 (Figs 3b and 4c) and which shows a brief upregulation at an early time point. While this conforms to the expected behavior of a gene regulated initially by STAT1 homodimers followed by decreasing expressing as ISGF3 takes over, I do not think that this is adequate to substantiate this important concept, because (i) examining expression of a single gene is insufficient backing for what is considered a general principle; (ii) protein expression is studied in Fig 4c when they ought to look at RNA expression, which is a more direct assessment of gene activity; and (iii) crucially, they do not make the counter test, namely to study genes regulated by ISGF3 only and not STAT1 homodimers.

The conjecture that homodimeric STAT1 is the initial type-1 IFN-induced transcription factor, and STAT2 comes into the picture only at later time points in unprimed cells or needs priming is difficult to reconcile with the observation that active STAT1 homodimers are not formed in cells that lack STAT2 (Stark and Schindler labs have worked on this). In other words, active STAT2 is there first before we have active STAT1, the opposite order from the one proposed here. While different cell types may differ in this regard, this aspect of type-1 IFN functioning cannot be ignored and needs

consideration.

STAT2 additionally functions as an adaptor for USP18. Stat2 is required for recruiting this critical negative IFN regulator to the IFN receptor (Arimoto et al (2017) Nat Struct Mol Biol 24, 279), such that the concentrations of biologically active STAT2 and USP18 are interlinked in more ways than currently incorporated in the model. I am therefore not fully persuaded that the model presented adequately represents knowledge of the molecular underpinnings to faithfully reflect the transcription factor dynamics during type-1 IFN signaling.

The experimental data appear sound and well controlled. I have noticed the authors use enzymatic assays (ECL) for quantitative Western blotting, a method with comparably limited dynamic detection range. I assume the authors included the relevant tests to avoid detection artefacts, but there is nothing mentioned to this effect.

Reviewer #2:

Kok et al address the dose response control of IFN α -responsive ISGF3 as a function of several positive and negative feedback regulators whose expression integrates prior exposure to IFN α . They find that differential dose response relationships of these regulators means that prior IFN α may either lead to sensitization or de-sensitization of the pathway. They identify the regulators that are responsible for each effect. This is a topical study that uses the systems biology approach of iterative experimentation and modeling in an effective manner. There is indeed much molecular knowledge in the literature but what has been lacking is a proper systems understanding that provides for quantitative predictions. The current study is well positioned to close this gap and a substantial amount of work has gone into this manuscript, but there are some important deficiencies or questions that should be addressed. These pertain to the experimental data, the model topology (justification for connections based on literature or own data), and parameter fitting.

Experimental Data:

1. Quantitative Immunoblotting: why do the authors use cytoplasmic extract in 1A and elsewhere? pSTAT1 translocates to the nucleus, so using cytoplasmic extract will not provide a complete picture of how much STAT1 was activated. Whole cell extract seems appropriate.
2. Complex formation: the model describes three STAT1-containing complexes, and two STAT2-containing complexes (simulations are shown in 4A), but no experimental data is provided about these. DNA gel shift studies should be able to provide such quantitative information. Inferring those complex activities from gene expression studies is unreliable, as specificities remain uncertain and the promoter dose response behavior is also.

Model Topology:

1. All connections should be justified clearly based on published literature or own data. Are there uncertainties about some of the connections? If decisions are made without strong experimental evidence, those assumptions should be made explicit and the consequences of alternative formulations should be discussed.
2. For example: ISRE and GAS specificity: I was surprised that the STAT1/STAT2 heterodimer is shown to activate ISRE sites in the model. My understanding is that ISRE binding is mediated by IRF9. Also, is the ISRE/GAS specificity of IFN α -induced regulators well established?

3. For example: Why do complexes only fall apart in the nucleus, and only assemble in the cytoplasm? Why do the reverse reactions not occur in each compartment?

Model Parameterization:

The Method description is extensive, and I apologize if I missed the answers to these questions.

- pSTAT1 is measured: this appears in multiple cytoplasmic and nuclear model species. Did the parameterization take this into account?
- Relative protein concentrations: 50,000 vs 1,000,000 were determined as indicated in Methods, but are the results also shown? It is important that all results pertaining to the parameterization are shown.

The reliability of the model is critical for applying it: it determines the reliability of interpreting measurements of clinical samples. I have not here commented on the application, but if uncertainties in the model are discussed more clearly, then uncertainties in interpreting the clinical samples should also be discussed.

Point-by-point response

Reviewer #1:

Kok et al. present a copious amount of experimental data and an accompanying first mathematical model describing the signaling by type-1 interferon (IFN) cytokines. This includes experiments using cancer patient-derived primary hepatocytes and the measurement of interferon concentrations in the plasma of patients chronically infected with hepatitis-b virus.

My main criticism concerns the definitions underlying the mathematical model and some of the central conclusion derived thereof. The authors' cell-based experiments show that priming of cells with low and high IFN concentrations leads to sensitization and de-sensitization of the cells to a subsequent high-dose stimulus, respectively. The authors use their model to understand why this might come about. The model predicts that stimulation with IFN of unprimed cells initially results in STAT1 homodimers, followed by STAT1-STAT2 heterodimers until after about 4 hours ISGF3 becomes the major transcription factor. The model further predicts that the amounts of STAT1 homodimers, STAT1-STAT2 heterodimers and ISGF3 would depend upon the priming IFN concentration, and that this would accordingly determine the repertoire of genes induced, including the positive feedback proteins (STAT1, STAT2, IRF9) and negative feedback proteins (USP18, SOCS1, IRF2) (page 6, top). However, with the exception of IRF2, all feedback proteins considered are induced by both type-1 IFN-induced transcription factors, ISGF3 and STAT1 homodimers. Thus, a shift from predominantly STAT1 homodimer-regulated genes to ISGF3-regulated genes would be expected to have little consequences as far as gene expression is concerned.

Nonetheless, a central prediction of their model is the occurrence of the different transcription factors in a temporal order that can be modulated by the priming conditions (page 9, top). Yet, there is no experimental support for this at all. The authors claim in the discussion that this data could not be measured directly (page 16, middle). However, this claim is unconvincing. The authors make no attempt to study the consecutive occurrence of the different transcription factors (eg via DNA binding kinetics), although it apparently takes several hours until there is enough IRF9 protein to assemble significant amounts of ISGF3 complex such that it becomes the dominant transcription factor.

We would like to take the opportunity to thank the reviewer for the insightful comments that inspired us to confirm the temporal and dose-dependent order of the formation of transcription factor complexes in the IFN α signal transduction pathway providing insights into the molecular mechanisms controlling pathway sensitization.

To address the reviewer's comment, we performed an electrophoretic mobility shift assay (EMSA) to experimentally examine as requested the potential temporal order of the occurrence of the different transcription factor complexes. To improve the signal to noise ratio, we utilized radioactivity-based EMSAs to analyze nuclear protein lysates obtained by cellular fractionation. Oligonucleotide probes used in the EMSA harbored either the GAS-binding region of the human IRF1 promoter or the ISRE-binding region of the human ISG15 promoter. In line with our model predictions, we detected a protein:DNA complex 1 h post stimulation with IFN α using the GAS probe, which was absent after 4 and 6 h of IFN α stimulation. Co-incubation with antibodies binding to STAT1, STAT2 or IRF9 demonstrated that this complex comprises pSTAT1 homodimers. Additionally, we detected a protein:DNA complex 4 and 6 h post IFN α stimulation using the ISRE probe, which was absent at 1 h post stimulation. Co-incubation with antibodies demonstrated that this complex comprises the ISGF3 transcription factor. These results are shown in the new Figure 4B and are described on page 9 of the manuscript:

"To verify the model-predicted consecutive occurrence of the different transcription factor complexes, we performed electrophoretic mobility shift assays (EMSA) as previously reported (Forero et al, 2019). Experiments using a probe comprising the GAS-binding region of the IRF1 promoter showed that an early

DNA:protein complex is induced by IFN α , which is absent at four and six hours post IFN α treatment (Figure 4B). Incubation of this complex with an antibody recognizing STAT1 led to a supershift, which was not observed upon incubation with antibodies detecting STAT2 or IRF9. These results experimentally confirmed a rapid but transient formation of pSTAT1 homodimers in response to stimulation with IFN α and thereby validated the model-predicted early occurrence of pSTAT1 homodimers in response to IFN α stimulation. In accordance with our assumption in the mathematical model, no major binding of the pSTAT1:pSTAT2 heterodimer to the GAS region was observed. Furthermore, a probe comprising the ISRE-binding region of the ISG15 promoter was tested and although this probe produced higher background signals, the data indicated that a late DNA:protein complex is formed upon stimulation with IFN α , which is absent at one hour post IFN α stimulation but present at four and six hours post IFN α treatment (Figure 4B). This late DNA:protein complex disappeared by co-incubation with antibodies recognizing either STAT1, STAT2 or IRF9. These results confirmed our model predictions that whereas pSTAT1 homodimers are rapidly and transiently formed in response to IFN α stimulation, it takes several hours until enough IRF9 protein has been produced to assemble significant amounts of the ISGF3 complex before it becomes the dominant transcription factor complex."

They resort to an indirect way to assess this, namely through studying SOCS3 expression, a gene that is induced by STAT1 homodimers but not ISGF3 (Figs 3b and 4c) and which shows a brief upregulation at an early time point. While this conforms to the expected behavior of a gene regulated initially by STAT1 homodimers followed by decreasing expressing as ISGF3 takes over, I do not think that this is adequate to substantiate this important concept, because (i) examining expression of a single gene is insufficient backing for what is considered a general principle; (ii) protein expression is studied in Fig 4c when they ought to look at RNA expression, which is a more direct assessment of gene activity; and (iii) crucially, they do not make the counter test, namely to study genes regulated by ISGF3 only and not STAT1 homodimers.

To substantiate our concept we followed the advice of the reviewer: We first performed a bioinformatics promoter analysis and identified several genes that were likely to be only GAS- or ISRE-activated (new Figure EV3A). To experimentally confirm this dependencies, we tested the activation of the respective candidate gene expression (i) upon stimulation with IFN γ , which only induces STAT1 phosphorylation and therefore the formation of pSTAT1 homodimers, and (ii) upon stimulation with IFN α , which results in transient pSTAT1 homodimer formation and sustained ISGF3 complex formation. As another gene harboring like SOCS3 only GAS sites in the promoter sequence we identified *IRF1*. In line with our expectations IFN γ stimulation resulted in a sustained expression of *SOCS3* and *IRF1*, whereas IFN α stimulation only caused a transient response (new Figure EV3B) corroborating the model-predicted transient pSTAT1 homodimer formation in response to IFN α . Additionally, we examined three genes that are primarily regulated by ISRE sites: *DDX58*, *HERC5* and *IFI44L*. While the bioinformatics promoter analysis suggested also putative GAS sites for these genes, our experimental evidence showed that the expression of these genes was not induced by IFN γ stimulation, whereas IFN α stimulation induced a sustained response during the investigated timeframe of four hours. Therefore, we identified two genes as indicators of pSTAT1 homodimer formation and three genes as indicators of ISGF3 formation. In addition to predicting the occupancy of GAS binding sites in response to prestimulation with low (28 pM) and intermediate (280 pM) doses of IFN α , we now also predicted the behavior of the occupancy of the ISRE binding sites (new Figure 4C). The mathematical model predicted that prestimulation with 28 and 280 pM IFN α for 24 h reduced the the formation of pSTAT1 homodimers upon stimulation with 1400 pM IFN α and consequently the occupancy of GAS binding sites in a dose-dependent manner. This model prediction was experimentally verified by the observed expression dynamics of *IRF1* and *SOCS3*. The model prediction of the dynamics of the formation of ISGF3 and consequently of the occupancy of ISRE binding sites upon prestimulation with 28 and 280 pM IFN α was distinctly different. The model predicted that prestimulation with 28 pM IFN α resulted in a higher occupancy at 24 h and in an accelerated increase of the occupancy of ISRE binding sites, which can be explained by the increased abundance of IRF9. Upon prestimulation with 280 pM the occupancy of ISRE binding sites was predicted to be even higher at 24 h, while the maximum occupancy of ISRE binding sites upon stimulation with 1400 pM IFN α

was similar to cells that were not prestimulated. This model predictions were experimentally verified by the analyzing the expression dynamics of *DDX58*, *HERC5* and *IFI44L*. By estimating the gene-specific transcription dynamics, our model was able to also link the measured gene expression to the predicted binding site occupancy. These results are shown in the new Figure 4C and described on page 10 of the manuscript:

“To experimentally validate this model-based hypothesis, we had to first identify interferon target genes whose expression is primarily regulated by the presence of GAS- or ISRE-binding sites. As shown in Figure EV3A, bioinformatics analyses revealed that the promoter regions of the *IRF1* and *SOCS3* genes harbor primarily putative GAS sequences, while the *DDX58*, *HERC5* and *IFI44L* genes contain primarily putative ISRE sites in proximity to the transcription start site. To experimentally verify that these genes are primarily driven by either GAS or ISRE sites, we stimulated Huh7.5 cells with 5000 IU/ml of either IFN γ or IFN α (corresponding to 1400 pM IFN α) (Figure EV3B). In line with the promoter analyses, stimulation with IFN γ , which only induces phosphorylation of STAT1 and therefore the formation of pSTAT1 homodimers, resulted in a sustained expression of *IRF1* and *SOCS3*, but not of the other genes. In agreement with our model-based prediction that IFN α triggers the transient formation of pSTAT1 homodimers and the sustained formation of ISGF3 complexes, IFN α induced a transient expression of *IRF1* and *SOCS3* with a peak around one hour after IFN α stimulation, while it induced a sustained expression for *DDX58*, *HERC5* and *IFI44L* in the timeframe of four hours. In sum these experiments established that the expression of *IRF1* and *SOCS3* is controlled by the presence of GAS sequences, whereas the expression of *DDX58*, *HERC5* and *IFI44L* is primarily dependent on the presence of ISRE sites.

To further evaluate the distinct IFN α dose-dependent formation of IFN α -induced transcription factor complexes as predicted by the mathematical model (Figure 4A), we simulated the dynamics of the occupancy of GAS binding sites upon stimulation with 1400 pM IFN α after prestimulation with 28 and 280 pM IFN α . Prestimulation with these IFN α doses was predicted by the mathematical model to reduce pSTAT1 homodimer formation and consequently the occupancy of GAS binding sites in a dose-dependent manner upon stimulation with 1400 pM IFN α (Figure 4C, upper left panel). The corresponding 68%-confidence intervals (shaded areas in Figure 4C) were computed as proposed in Kreutz et al. (2012). To verify this model prediction, we examined the dynamics of the production of the GAS-dependent transcripts *IRF1* and *SOCS3* upon stimulation with 1400 pM IFN α in untreated cells and after prestimulation with 28 and 280 pM IFN α (Figure 4C, symbols in upper right panels). In accordance with the mathematical model, the dynamics of the expression of these genes was reduced by the prestimulation with 28 and 280 pM IFN α and reflected the predicted reduced formation of the pSTAT1 homodimers. Conversely, the dynamics of the formation of ISGF3 complexes and consequently the occupancy of ISRE binding sites upon stimulation with 1400 pM IFN α after prestimulation with 28 and 280 pM IFN α was predicted. In contrast to the GAS/ISRE binding sites controlling the expression of *STAT1*, *STAT2*, *IRF9*, *IRF2*, *USP18* and *SOCS1* that are occupied by ISGF3 and pSTAT1:pSTAT2 heterodimers, the ISRE binding sites are only occupied by ISGF3. The model predicted that the prestimulation with 28 pM IFN α resulted in a higher initial level of occupied ISRE binding sites at 24 hours and an accelerated increase of occupied ISRE binding sites upon stimulation with 1400 pM IFN α . Prestimulation with 280 pM IFN α was predicted to further increase the initial occupancy of ISRE binding sites after 24 hours and the maximum occupancy of ISRE binding sites upon stimulation with 1400 pM IFN α was predicted to be similar as in cells that were not prestimulated (Figure 4C, lower left panel). The experimental analysis of the dynamics of the ISRE-dependent transcripts *DDX58*, *HERC5* and *IFI44L* confirmed upon prestimulation with 28 pM IFN α a higher basal expression at 24 h and an accelerated gene induction compared to cells that were not prestimulated. Upon prestimulation with 280 pM IFN α , the basal expression of *DDX58*, *HERC5* and *IFI44L* was even higher at 24 h and the maximum expression upon stimulation with 1400 pM IFN α was similar to cells that were not prestimulated, reflecting the predicted dynamics of the occupied ISRE binding sites (Figure 4C, symbols in lower right panels). These measured transcripts were linked to the amount of occupied binding sites predicted by the mathematical model by estimating gene-specific parameters, i.e. mRNA synthesis and degradation rate, time delay of mRNA

production and the Hill coefficient, while all remaining model parameters were fixed. This allowed to overlay the measured dynamics of this gene set with the simulated model trajectories (Figure 4C, lines in upper and lower right panels)."

These new results are also discussed on page 18 of the manuscript:

"Our model allowed us to predict the dynamics of the different IFN α -induced transcription factor complexes and to quantitatively link these to gene expression. IFN α -target genes can be classified into three groups: Genes that are primarily regulated by GAS sites (*IRF1* and *SOCS3*), genes that are primarily regulated by ISRE sites (*DDX58*, *HERC5* and *IFI44L*) and genes that are regulated by both GAS and ISRE sites (*STAT1*, *STAT2*, *IRF9*, *IRF2*, *USP18* and *SOCS1*). Utilizing model reduction techniques (Maiwald et al, 2016), we showed that the data can be described sufficiently if we assume that the first gene group is induced by nuclear pSTAT1 homodimers, the second gene group is regulated by nuclear ISGF3 and the third gene group is also induced by ISGF3 and, to a lesser extent, by pSTAT1:pSTAT2 heterodimers. Interestingly, the mathematical model predicted that pSTAT1 homodimers form early after IFN α stimulation, whereas ISGF3 complexes form later, which is corroborated by the experimental evidence obtained by EMSA and explains the temporal order of target gene expression. Of note, we are not excluding other connections, but they are apparently not necessary to describe our experimental data. It has previously been reported that the IFN α -inducible pSTAT1:pSTAT2 heterodimer preferentially binds to sequences that closely resembles GAS elements (Ghislain et al, 2001). Our experimental data is not contradicting this notion, as in our model, the pSTAT1:pSTAT2 heterodimer activates genes that are controlled by both GAS and ISRE sites. In our EMSA, we induced a supershit of GAS binding complexes with antibodies against STAT1, but not against STAT2. Thus, major contributions of the pSTAT1:pSTAT2 heterodimer to the dynamics of GAS-only regulated genes such as *IRF1* and *SOCS3* were excluded."

The conjecture that homodimeric STAT1 is the initial type-1 IFN-induced transcription factor, and STAT2 comes into the picture only at later time points in unprimed cells or needs priming is difficult to reconcile with the observation that active STAT1 homodimers are not formed in cells that lack STAT2 (Stark and Schindler labs have worked on this). In other words, active STAT2 is there first before we have active STAT1, the opposite order from the one proposed here. While different cell types may differ in this regard, this aspect of type-1 IFN functioning cannot be ignored and needs consideration.

We apologize for being unclear regarding this point. We did not intend to claim that STAT2 needs priming before being activated. Because STAT1 and STAT2 are both very rapidly phosphorylated upon IFN α stimulation, we are unable to resolve the temporal order of pSTAT1 and pSTAT2 activation given our experimental data (Figure 1D). In our mathematical model, pSTAT1:pSTAT2 heterodimers are predicted to be formed immediately upon stimulation with IFN α . We concluded that pSTAT1 homodimers are formed faster than pSTAT1:pSTAT2 heterodimers, which is not in contrast to phosphorylated monomeric STAT2 being formed earlier than phosphorylated monomeric STAT1. In other words, our mathematical model summarizes the different steps in the (i) formation of phosphorylated monomeric STAT2, (ii) formation of phosphorylated monomeric STAT1 and (iii) formation of homo- and heterodimers in only two reactions. To make this point more understandable in the manuscript, we added the following sentence to the description of the mathematical model in the manuscript (page 6):

"In the mathematical model, phosphorylation and dimerization of STAT1 and STAT2 were approximated by single reactions in which the active receptor complex forms and induces dimer formation directly."

We also re-phrased the sentence "the mathematical model indicated that subsequently pSTAT1:pSTAT2 heterodimers are formed" in the manuscript on page 9:

"The mathematical model indicated that simultaneously pSTAT1:pSTAT2 heterodimers are formed, albeit with slower dynamics."

STAT2 additionally functions as an adaptor for USP18. Stat2 is required for recruiting this critical negative IFN regulator to the IFN receptor (Arimoto et al (2017) Nat Struct Mol Biol 24, 279), such that the concentrations of biologically active STAT2 and USP18 are interlinked in more ways than

currently incorporated in the model. I am therefore not fully persuaded that the model presented adequately represents knowledge of the molecular underpinnings to faithfully reflect the transcription factor dynamics during type-1 IFN signaling.

Inspired by the reviewer's comment, we analyzed two different mathematical models utilizing our complete set of available data. The first model comprised an interaction term between USP18 and STAT2, while the second one did not. It turned out that these models both described the data equally well, which was shown by the same likelihood value after fitting the models with a multi-start optimization starting from 500 randomly chosen parameter vectors. These studies do not exclude that the interaction of USP18 and STAT2 exists, however, it is not necessary to describe our data. Similarly, we tested and rejected two other mechanisms: A cytoplasmic phosphatase dissociating pSTAT1 homodimers, pSTAT1:pSTAT2 heterodimers and ISGF3 in the cytoplasm and the impact of pSTAT1 homodimers in the nucleus on the OccGAS/ISRE-induced genes. The results are shown in the new Appendix Figure S5 and are described on page 6 of the manuscript:

„We tested three additional mechanisms, (i) a cytoplasmic phosphatase dissociating pSTAT1dimc, pSTAT1pSTAT2c and ISGF3c, (ii) STAT2 functioning as an adapter for USP18 (Arimoto et al, 2017) and (iii) pSTAT1dimn inducing OccGAS/ISREbs by formulating alternative mathematical models (Appendix Figure S5A). We re-estimated the model parameters for each of these three hypotheses and calculated the Bayesian information criterion (BIC). In all three cases, the goodness-of-fit was nearly the same, however, due to the additional parameters, the BIC was significantly worse and these additional mechanisms were rejected (Appendix Figure S5B).“

The experimental data appear sound and well controlled. I have noticed the authors use enzymatic assays (ECL) for quantitative Western blotting, a method with comparably limited dynamic detection range. I assume the authors included the relevant tests to avoid detection artefacts, but there is nothing mentioned to this effect.

We entirely agree with the reviewer and always take extra care to ensure reliable quantification in our immunoblotting experiments. To provide more evidence, we now included an additional experiment. We stimulated Huh7.5 cells with increasing amounts of IFN α and detected pSTAT1 in total cell lysates with both chemiluminescence using a CCD camera device (ImageQuant) and with fluorescence employing a near-infrared fluorescence scanner (Odyssey). As shown in the new Appendix Figure S1B-D, the results obtained with both methods were highly comparable. This evidence is described on page 4 of the manuscript:

“To ensure the linearity of detection in the enzymatic assays (chemiluminescence) employed for quantitative immunoblotting, we not only measured the abundance of pSTAT1 in total cellular lysates by chemiluminescence employing a CCD camera device (Appendix Figure S1B), but also by fluorescence using a near-infrared fluorescence scanner (Appendix Figure S1C). The comparison of the chemiluminescence-based quantifications with the fluorescence-based quantifications revealed a Pearson correlation coefficient of 0.99, showing a comparable detection range for both methods (Appendix Figure S1D).”

Reviewer #2:

Kok et al address the dose response control of IFN α -responsive ISGF3 as a function of several positive and negative feedback regulators whose expression integrates prior exposure to IFN α . They find that differential dose response relationships of these regulators means that prior IFN α may either lead to sensitization or de-sensitization of the pathway. They identify the regulators that are responsible for each effect. This is a topical study that uses the systems biology approach of iterative experimentation and modeling in an effective manner. There is indeed much molecular knowledge in the literature but what has been lacking is a proper systems understanding that provides for quantitative predictions. The current study is well positioned to close this gap and a substantial amount of work has gone into this manuscript, but there are

some important deficiencies or questions that should be addressed. These pertain to the experimental data, the model topology (justification for connections based on literature of own data), and parameter fitting.

Experimental Data:

1. Quantitative Immunoblotting: why do the authors use cytoplasmic extract in 1A and elsewhere? pSTAT1 translocates to the nucleus, so using cytoplasmic extract will not provide a complete picture of how much STAT1 was activated. Whole cell extract seems appropriate.

We agree with the reviewer and repeated the IFN α dose response of STAT1 phosphorylation for total cell lysates, resulting in a comparable dose-response curve for pSTAT1 as obtained with cytoplasmic extracts (new Appendix Figure S1B). Additionally, we now show the corresponding dose response plot of pSTAT1 using nuclear lysates in the new Appendix Figure S1A. The results are described on page 4 of the manuscript:

“Since STAT proteins translocate to the nucleus upon activation, we additionally measured pSTAT1 in nuclear lysates (Appendix Figure S1A) as well as in total cell lysates (Appendix Figure S1B), showing a comparable dose-response behavior.”

2. Complex formation: the model describes three STAT1-containing complexes, and two STAT2-containing complexes (simulations are shown in 4A), but no experimental data is provided about these. DNA gel shift studies should be able to provide such quantitative information. Inferring those complex activities from gene expression studies is unreliable, as specificities remain uncertain and the promoter dose response behavior is also.

We thank the reviewer for this comment that helped us to confirm the IFN α induced formation of the three transcription factor complexes and thereby provide deeper insights into the molecular mechanisms controlling sensitization of the IFN α signal transduction pathway.

To address the reviewer’s comment, we performed an electrophoretic mobility shift assay (EMSA) to experimentally examine as requested the potential temporal order of the occurrence of the different transcription factor complexes. To improve the signal to noise ratio, we utilized radioactivity-based EMSAs to analyze nuclear protein lysates obtained by cellular fractionation. Oligonucleotide probes used in the EMSA harbored either the GAS-binding region of the human IRF1 promoter or the ISRE-binding region of the human ISG15 promoter. In line with our model predictions, we detected a protein:DNA complex 1 h post stimulation with IFN α using the GAS probe, which was absent after 4 and 6 h of IFN α stimulation. Co-incubation with antibodies binding to STAT1, STAT2 or IRF9 demonstrated that this complex comprises pSTAT1 homodimers. Additionally, we detected a protein:DNA complex 4 and 6 h post IFN α stimulation using the ISRE probe, which was absent at 1 h post stimulation. Co-incubation with antibodies demonstrated that this complex comprises the ISGF3 transcription factor. These results are shown in the new Figure 4B and are described on page 9 of the manuscript:

“To verify the model-predicted consecutive occurrence of the different transcription factor complexes, we performed electrophoretic mobility shift assays (EMSA) as previously reported (Forero et al, 2019). Experiments using a probe comprising the GAS-binding region of the IRF1 promoter showed that an early DNA:protein complex is induced by IFN α , which is absent at four and six hours post IFN α treatment (Figure 4B). Incubation of this complex with an antibody recognizing STAT1 led to a supershift, which was not observed upon incubation with antibodies detecting STAT2 or IRF9. These results experimentally confirmed a rapid but transient formation of pSTAT1 homodimers in response to stimulation with IFN α and thereby validated the model-predicted early occurrence of pSTAT1 homodimers in response to IFN α stimulation. In accordance with our assumption in the mathematical model, no major binding of the pSTAT1:pSTAT2 heterodimer to the GAS region was observed. Furthermore, a probe comprising the ISRE-binding region of the ISG15 promoter was tested and although this probe produced higher background signals, the data indicated that a late DNA:protein complex is formed upon stimulation with IFN α , which is absent at one hour post IFN α stimulation but present at four and six hours post IFN α treatment (Figure 4B). This late DNA:protein complex disappeared by co-incubation with antibodies

recognizing either STAT1, STAT2 or IRF9. These results confirmed our model predictions that whereas pSTAT1 homodimers are rapidly and transiently formed in response to IFN α stimulation, it takes several hours until enough IRF9 protein has been produced to assemble significant amounts of ISGF3 complex before it becomes the dominant transcription factor complex.”

Model Topology:

1. All connections should be justified clearly based on published literature or own data. Are there uncertainties about some of the connections? If decisions are made without strong experimental evidence, those assumptions should be made explicit and the consequences of alternative formulations should be discussed.

We followed the reviewer’s suggestion and now included in Appendix Table S1 a column to give citations for all reactions based on biochemical evidence. We also report in this table reactions that were included or excluded based on our result (see answer to point 3 below).

2. For example: ISRE and GAS specificity: I was surprised that the STAT1/STAT2 heterodimer is shown to activate ISRE sites in the model. My understanding is that ISRE binding is mediated by IRF9. Also, is the ISRE/GAS specificity of IFN α -induced regulators well established?

We apologize for being unclear on this point. The term ISRE sites in the original manuscript referred to promoters of genes that are primarily controlled by ISGF3, but which also contain GAS sites. To clarify this point, we now re-named these as occupied GAS/ISRE binding sites (OccGAS/ISREbs). These OccGAS/ISREbs control the expression of *STAT1*, *STAT2*, *IRF9*, *IRF2*, *USP18* and *SOCS1* and are induced by ISGF3 and, to a lesser extent, by pSTAT1:pSTAT2 heterodimers. The impact of pSTAT1 homodimers on these binding sites was shown to be negligible by model selection (see answer to point 3 below). We clarify this aspect on page 6 of the manuscript:

“The positive feedback transcripts *IRF9* mRNA, *STAT1* mRNA and *STAT2* mRNA as well as the negative feedback transcripts *USP18* mRNA, *SOCS1* mRNA and *IRF2* mRNA harbor both GAS binding sites as well as interferon-stimulated-response element (ISRE) binding sites. Therefore, pSTAT1:pSTAT2n and ISGF3n both contribute to formation of these occupied GAS/ISRE binding sites (OccGAS/ISREbs).”

Additionally, we now discuss the ISRE/GAS specificity of IFN α -induced regulators on page 18 of the manuscript:

“Our model allowed us to predict the dynamics of the different IFN α -induced transcription factor complexes and to quantitatively link these to gene expression. IFN α -target genes can be classified into three groups: Genes that are primarily regulated by GAS sites (*IRF1* and *SOCS3*), genes that are primarily regulated by ISRE sites (*DDX58*, *HERC5* and *IFI44L*) and genes that are regulated by both GAS and ISRE sites (*STAT1*, *STAT2*, *IRF9*, *IRF2*, *USP18* and *SOCS1*). Utilizing model reduction techniques (Maiwald et al, 2016), we showed that the data can be described sufficiently if we assume that the first gene group is induced by nuclear pSTAT1 homodimers, the second gene group is regulated by nuclear ISGF3 and the third gene group is also induced by nuclear ISGF3 and, to a lesser extent, by nuclear pSTAT1:pSTAT2 heterodimers. Interestingly, the mathematical model predicted that pSTAT1 homodimers form early after IFN α stimulation, whereas ISGF3 complexes form later, which is corroborated by the experimental evidence obtained by EMSA and explains the temporal order of target gene expression. Of note, we are not excluding other connections, but they are apparently not necessary to describe our experimental data. It has previously been reported that the IFN α -inducible pSTAT1:pSTAT2 heterodimer preferentially binds to sequences that closely resemble GAS elements (Ghislain et al, 2001). Our experimental data is not contradicting this result, as in our model, the pSTAT1:pSTAT2 heterodimer activates genes that are controlled by both GAS and ISRE sites. In our EMSA, we inhibited formation of GAS binding complexes with antibodies against STAT1, but not against STAT2. Thus, major contributions of the pSTAT1:pSTAT2 heterodimer to the dynamics of GAS-only regulated genes such as *IRF1* and *SOCS3* were excluded.”

Concerning the GAS and ISRE dependency of the expression of IFN α -induced genes, we combined bioinformatics promoter analyses with experiments testing the stimulation with IFN α or IFN γ to identify two genes that are only regulated by GAS sites as well as three genes that are only regulated by ISRE sites. These results are shown in the new Figure 4C and are described on page 10 of the manuscript:

“To experimentally validate this model-based hypothesis, we had to first identify interferon target genes whose expression is primarily regulated by the presence of GAS or ISRE binding sites. As shown in Figure EV3A, bioinformatics analyses revealed that the promoter regions of the *IRF1* and *SOCS3* genes harbor primarily putative GAS sequences, while the *DDX58*, *HERC5* and *IFI44L* genes contain primarily putative ISRE sites in proximity to the transcription start site. To experimentally verify that these genes are primarily driven by either GAS or ISRE sites, we stimulated Huh7.5 cells with 5000 IU/ml of either IFN γ or IFN α (corresponding to 1400 pM IFN α) (Figure EV3B). In line with the promoter analyses, stimulation with IFN γ , which only induces phosphorylation of STAT1 and therefore the formation of pSTAT1 homodimers, resulted in a sustained expression of *IRF1* and *SOCS3*, but not of the other genes. In agreement with our model-based prediction that IFN α triggers the transient formation of pSTAT1 homodimers and the sustained formation of ISGF3 complexes, IFN α induced a transient expression of *IRF1* and *SOCS3* with a peak around one hour after IFN α stimulation, while it induced a sustained expression for *DDX58*, *HERC5* and *IFI44L* in the timeframe of four hours. In sum these experiments established that the expression of *IRF1* and *SOCS3* is controlled by the presence of GAS sequences, whereas the expression of *DDX58*, *HERC5* and *IFI44L* is primarily dependent on the presence of ISRE sites.”

3. For example: Why do complexes only fall apart in the nucleus, and only assemble in the cytoplasm? Why do the reverse reactions not occur in each compartment?

To test the proposition of the reviewer, we calibrated an alternative model where complexes also fall apart in the cytoplasm. It turned out that both models described the data equally well which was shown by the same likelihood value after fitting the models with a multi-start optimization starting from 500 randomly chosen parameter vectors. By computing the Bayesian information criterion (BIC), we rejected the model extension. We would like to emphasize that we are not excluding these reactions from occurring, but we do not need them to describe our data. Similarly, we tested and rejected two other mechanisms: STAT2 functioning as an adapter for USP18 as suggested by Arimoto et al, 2017 and the impact of pSTAT1 homodimers in the nucleus on the OccGAS/ISRE-induced genes. The results are shown in the new Appendix Figure S5 and are described on page 6 of the manuscript:

„We tested three additional mechanisms, (i) a cytoplasmic phosphatase dissociating pSTAT1dimc, pSTAT1pSTAT2c and ISGF3c, (ii) STAT2 functioning as an adapter for USP18 (Arimoto et al, 2017) and (iii) pSTAT1dimn inducing OccGAS/ISREbs by formulating alternative mathematical models (Appendix Figure S5A). We re-estimated the model parameters for each of these three hypotheses and calculated the Bayesian information criterion (BIC). In all three cases, the goodness-of-fit was nearly the same, however, due to the additional parameters, the BIC was significantly worse and these additional mechanisms were rejected (Appendix Figure S5B).“

Model Parameterization:

The Method description is extensive, and I apologize if I missed the answers to these questions.

- pSTAT1 is measured: this appears in multiple cytoplasmic and nuclear model species. Did the parameterization take this into account?

Indeed, different model species including pSTAT1 were considered in the model as observation functions summing up the different model species that contain pSTAT1. These functions are listed in Appendix Table S2. Please compare e.g. the observation function for pSTAT1_{Nuc}, pSTAT1_{Cyt} and pSTAT1.

- Relative protein concentrations: 50,000 vs 1,000,000 were determined as indicated in Methods, but are the results also shown? It is important that all result pertaining to the parameterization are shown.

The results of the determination of the protein concentrations in Huh7.5 cells are shown in Figure 3C. The source data is provided as "DatasetEV02_Huh75_merged". For example, the number of molecules per cell for STAT1 in unstimulated cells is given by "Experimental Condition = 0_0", "Name of observable = mc_STAT1".

The reliability of the model is critical for applying it: it determines the reliability of interpreting measurements of clinical samples. I have not here commented on the application, but if uncertainties in the model are discussed more clearly, then uncertainties in interpreting the clinical samples should also be discussed.

We completely agree with the reviewer on the importance of assessing the uncertainties of predictions. To now better emphasize this point we added the following paragraph to the discussion (page 20):

"By means of our mathematical model, uncertainties of predictions can be computed based on the input data of the model (Kaschek et al, 2019; Kreutz et al, 2012). However, the findings derived from the simulation of the virtual patient cohort do not account for uncertainties of clinical measurements. These will have to be addressed first before serving as an input for the mathematical model to predict patient-specific pathway sensitization."

10th Dec 2019

Manuscript Number: MSB-19-8955R, Disentangling molecular mechanisms regulating sensitization of interferon alpha signal transduction

Thank you for sending us your revised manuscript. We have now heard back from the two reviewers who were asked to evaluate your study. As you will see below, the reviewers think that the study has significantly improved as a result of the performed revisions. However, they still raise a series of concerns, which we would ask you to address in an exceptional second round of revision.

As you will see below, the most substantial remaining issues are the following:

- The EMSAs shown in support of the dimer switching are not of sufficient quality, and as both reviewers point out, further and better quality replicates need to be included.
- Reviewer #1 questions several of the model's assumptions, which need to be better justified.

All other issues raised need to be convincingly addressed.

REFEREE REPORTS

Reviewer #1:

I thank the authors for addressing the points I had raised and I acknowledge that they have added useful experimental evidence. However, regarding the authors' key conclusion that dimer-switching is underlying the observed transcriptional changes during IFN treatment they fall way short of providing experimental evidence that can be considered "strong". The addition of Figure 4B is a beginning to address this critical point, but the data remain incomplete and unconvincing. Incomplete, because only one situation (untreated cells) is considered, pSTAT1pSTAT2 dimers are not detected, and pre-treated cells are not included. However, the model predicts two consequences of IFN pre-treatment, namely (i) the disappearance of STAT1 homodimers (GAF) and (ii) the rapid appearance of ISGF3, whereby the former is particularly interesting. Both of which are experimentally accessible but have not been investigated. The additional data are not fully convincing, since the quality of some of the EMSA data shown in Figure 4B is poor (ISGF3 binding activity). It is noteworthy that untreated Huh7.5 hepatoma cells show ISGF3 activity only after several hours of IFN stimulation (Figure 4B) -in agreement with other studies using these cells- which the authors' take as strong evidence in support of their dimer-switching model. However, essentially all studies using other human and mouse cell types including liver cells (fibroblasts, hepatocytes, macrophages) do not show a delayed ISGF3 activity. As an example, I refer to mouse primary hepatocytes, where ISGF3 activity is induced within 30 min of IFN treatment (J Leukoc Biol. 93: 377). While there are likely cell type and species differences in IFN signaling, to me these differences indicate that the inability to detect ISGF3 early on is not indicative of a generally applicable biological mechanism (i.e. GAF activity followed by ISGF3), but rather reflects a limitation of the experimental approach chosen. This, of course, raises questions also about the biological significance of the model that the authors propose to describe IFN signaling. This latter point is also relevant in the following. To support the dimer-switching hypotheses the authors' use bioinformatics tools to identify genes that are regulated primarily by GAS or ISRE binding sites. Subsequent transcription analyses of the respective genes are then correlated with the model-predicted occurrences of STAT1 homodimers and ISGF3 transcription factors. However, the authors make assumptions whose validity is questionable, i.e. considering only 3kB from the transcription start site as relevant for gene regulation; and the stringency of the GAS and ISRE consensus sequences used for the identification of binding sites. Accordingly, their conclusions, which are based on these assumptions, are likewise not beyond reproach. Specifically, they disagree with previous findings. IRF1 and SOCS3 are studied as the examples for primarily STAT1 homodimer-induced genes. However, SOCS3 induction does not require STAT1, because cells that lack STAT1 retain IFN-gamma induced SOCS3 expression, contradicting the author's assignment of SOCS3 as a STAT1 homodimer-regulated gene (PNAS 98, 6674). Likewise, IFN alpha-induced IRF1 expression involves recruitment of ISGF3 to an ISRE (Nat Immunol. 15, 168). Thus, the authors' explanation for the transient expression of some genes by the transient presence of STAT1 homodimers is not in line with established knowledge of IFN-dependent gene regulation. Their explanation for the delayed but sustained expression of another set of genes by a switch to ISGF3 is similarly flawed. The authors' bioinformatics analyses identify three other genes as being regulated by ISGF3, not STAT1 homodimers. Accordingly, these genes are demonstrated to be unresponsive to IFN-gamma in the Huh cells, a strong inducer of STAT1 homodimers (Figure EV3B). I do not doubt the observations made for these cells. However, two of the three genes tested by the authors (DDX58, IFI44L) are well-established IFN gamma target genes in human hepatocytes (Gastroenterology 143, 777) and other cell types

http://software.broadinstitute.org/gsea/msigdb/cards/HALLMARK_INTERFERON_GAMMA_RESPONSE

. The mentioned variabilities/disagreements with published results are important, because they call into question the validity and general biological significance of key assumptions and undermine central conclusions of this work.

In summary, the authors present a large and credible volume of experimental data that undoubtedly add to our knowledge about IFN signaling, and their model is able to replicate their experimental findings. Nonetheless, the progress in knowledge is convincing only insofar as experimentally derived

details are concerned. On a conceptual level, and regarding biological significance at large, this work does not provide sufficient insight in order to constitute a major step ahead in the understanding of IFN signaling.

Reviewer #2:

I appreciate the additional work the authors undertook to address my critique points. The authors have done a lot and I feel overall this is a strong manuscript. Most points were addressed to satisfaction. However, one issue remains, but I feel that it can be sidestepped, so long as the relative strength of the evidence is fairly discussed.

The authors address the question about TF complexes by providing an EMSA. However, its quality is really so low that it cannot really be used to draw conclusions with confidence. It is really below the publishable threshold. Is this a key result that must be included, or can the paper not stand without it? If the authors feel that this result is critical, they need to redo the EMSAs, and provide a quantitative evaluation of replicates. In my mind it may be sufficient to discuss the model predictions and indicate that they await experimental confirmation.

Also, the nomenclature of occupied ISRE/GAS sites is very cumbersome and confusing. The ISRE or GAS sequences either exist or do not. If ISGF3 binds the ISRE, then this is a ISGF3-ISRE complex. If the GAS site is occupied it is a GAF-GAS complex. Are the authors suggesting that there may be GAF-ISRE complexes? I suggest that the description is revised to use conventional nomenclature of sites and complexes.

Point-by-point response

Reviewer #1:

I thank the authors for addressing the points I had raised and I acknowledge that they have added useful experimental evidence. However, regarding the authors' key conclusion that dimer-switching is underlying the observed transcriptional changes during IFN treatment they fall way short of providing experimental evidence that can be considered "strong". The addition of Figure 4B is a beginning to address this critical point, but the data remain incomplete and unconvincing. Incomplete, because only one situation (untreated cells) is considered, pSTAT1:pSTAT2 dimers are not detected, and pre-treated cells are not included. However, the model predicts two consequences of IFN pre-treatment, namely (i) the disappearance of STAT1 homodimers (GAF) and (ii) the rapid appearance of ISGF3, whereby the former is particularly interesting. Both of which are experimentally accessible but have not been investigated. The additional data are not fully convincing, since the quality of some of the EMSA data shown in Figure 4B is poor (ISGF3 binding activity).

To strengthen our conclusions, we followed the advice of the reviewer and repeated the EMSA experiments using GAS probes to detect pSTAT1 homodimer not only with untreated cells, but now also included experiments with prestimulated cells. These experiment were performed in triplicates and were quantified. The obtained results and the model predictions are now presented in the new Fig. 4B (left panel). An exemplary image of the EMSA using the GAS probes is displayed in Fig. EV3A (left panel) and the specificity of the detection of pSTAT1 homodimers was confirmed by super-shift experiments shown in Fig. EV3A (right panel). The results shown in the new Fig. 4B (left panel) confirmed as predicted the appearance of pSTAT1 homodimers at early time points after IFN α stimulation in untreated cells and the disappearance of the pSTAT1 homodimers in prestimulated cells. Additionally, we performed multiple experiments to improve the quality of the EMSA using ISRE probes and even contacted authors of other publications for advice. But it turned out that the detection of the ISGF3 complex binding to ISRE probes in EMSA experiments in human hepatocytes is particularly challenging and the quality of the obtained results – even though being in line with our model predictions – remained unsatisfactorily. Therefore, we rather performed experiments to directly quantify the dynamics of the IFN α -induced formation of the ISGF3 complex (pSTAT1:pSTAT2:IRF9). We performed co-immunoprecipitation (co-IP) experiments by conducting immunoprecipitation experiments with antibodies recognizing IRF9 and detecting the co-immunoprecipitating pSTAT1 by immunoblotting. The quantifications of triplicate experiments performed with untreated and prestimulated cells as well as the model predictions for the formation of ISGF3 complexes are shown in the new Figure 4B, right panel and an exemplary immunoblot is shown in Fig. EV3C. These results confirm, in line with the model prediction, the occurrence of ISGF3 at later time points after IFN α stimulation, whereas in prestimulated cells ISGF3 is already detectable at earlier time points. Further, we performed immunoprecipitation experiments with antibodies recognizing STAT2 and detecting co-immunoprecipitating pSTAT1, which reflects pSTAT1 present in pSTAT1:pSTAT2 homodimers and ISGF3 complexes. The quantifications of triplicate experiments as well as the model-predicted sum of the pSTAT1:pSTAT2 homodimers and ISGF3 complexes are shown in the new Figure 4B, middle panel and an exemplary immunoblot is shown in Fig. EV3B. These results are described in the revised results section on page 9 of the manuscript:

“To experimentally verify the model-predicted consecutive occurrence of the different transcription factor complexes, we performed electrophoretic mobility shift assays (EMSA) as previously reported (Forero et al, 2019). Experiments using a probe comprising the GAS-binding region of the IRF1 promoter (Figure EV3A, left panel) showed that in untreated Huh7.5 cells an early DNA:protein complex is induced after one hour (corresponding to 25 hours after mock prestimulation) of stimulation with 1400 pM IFN α . This DNA:protein complex is absent at four and six hours post IFN α stimulation of Huh7.5 cells (corresponding to 28 and 30 hours after mock prestimulation, respectively). As shown in Figure EV3A, right panel, incubation of the lysate-DNA mixture with an antibody recognizing STAT1 led to a supershift, which was

absent upon addition of antibodies detecting STAT2 or IRF9, confirming the specificity of the detected complex as pSTAT1 homodimer. In accordance with our assumption in the mathematical model, no major binding of the pSTAT1:pSTAT2 heterodimer to the GAS region was observed. On the contrary to untreated Huh7.5 cells, formation of pSTAT1 homodimeric complexes induced by stimulation with 1400 pM IFN α is much reduced in cells prestimulated with 280 pM IFN α for 24 h. To quantitatively compare the obtained results with our model predictions, we predicted the dynamics of occupied GAS binding sites induced by 1400 pM IFN α in untreated Huh7.5 cells and in Huh7.5 cells prestimulated with 280 pM IFN α for 24 h (Figure 4B, left panel). The corresponding 68%-confidence intervals were computed as proposed in Kreutz et al. (2012). The simulation showed a steep increase of occupied GAS binding sites within the first hour after stimulation, which was suppressed upon prestimulation of cells with IFN α . As shown in Figure 4B, left panel, the mean values of pSTAT1 homodimeric complexes detected by EMSA (N=3) were in agreement with the model prediction and experimentally confirmed a rapid but transient formation of pSTAT1 homodimers in response to stimulation with IFN α , which was much reduced upon IFN α prestimulation, validating the model-predicted early occurrence of pSTAT1 homodimers in response to IFN α stimulation. To investigate the IFN α -induced dynamics of the formation of the other STAT1-containing transcription factor complexes, we performed co-immunoprecipitation (co-IP) experiments. We stimulated non-prestimulated Huh7.5 cells or Huh7.5 cells prestimulated with 280 pM IFN α for 24 h for one to six hours (corresponding to 25 to 30 hours after prestimulation) with 1400 pM IFN α . The cellular lysates were used for immunoprecipitation experiments using antibodies recognizing STAT2 and co-immunoprecipitated pSTAT1 was detected by quantitative immunoblotting (Figure EV3B). With these co-IP experiments the dynamics of the sum of IFN α -induced formation of pSTAT1:pSTAT2 heterodimers and pSTAT1:pSTAT2:IRF9 trimers (ISGF3) was detected. In untreated Huh 7.5 cells the signal for co-immunoprecipitating pSTAT1 was maximal after one hour of stimulation with 1400 pM IFN α and slowly decreased thereafter but not reaching baseline after six hours of stimulation (25 to 28 hours after mock prestimulation). Upon prestimulation of Huh7.5 cells with 280 pM IFN α for 24 h, higher levels of total STAT2 were observed, while the overall signal of co-immunoprecipitated pSTAT1 was lower. Distinctively, it was already detectable after 24 hours of prestimulation and increased to a much lower extent by stimulation with 1400 pM IFN α compared to the amount detected in untreated cells. To compare the experimental results obtained by the quantification of co-immunoprecipitating pSTAT1 (N=3) to the predictions by our mathematical model, we calculated the dynamics of the sum of pSTAT1:pSTAT2 heterodimers and the pSTAT1:pSTAT2:IRF9 trimers (ISGF3) induced by 1400 pM IFN α in untreated Huh7.5 cells and Huh7.5 cells prestimulated with 280 pM IFN α for 24 h and computed 68%-confidence intervals. As shown in Figure 4B, middle panel, these results experimentally confirmed the broad peak in the model-predicted dynamics of the sum of pSTAT1:pSTAT2 heterodimers and ISGF3, which was reduced to around one third upon prestimulation of the cells. Additionally, to quantify in untreated and prestimulated Huh7.5 cells the dynamics of IFN α -induced ISGF3 complex formation, co-IP experiments were performed using antibodies recognizing IRF9 and co-immunoprecipitated pSTAT1 was detected by quantitative immunoblotting (Figure EV3C). The signal for IRF9-precipitated pSTAT1 increased upon stimulation of untreated Huh7.5 cells with 1400 pM IFN α with a peak at four hours post IFN α treatment (28 hours after mock prestimulation). Upon prestimulation of Huh7.5 cells with 280 pM IFN α for 24 h, IRF9 levels were strongly increased and co-immunoprecipitated pSTAT1 was now already peaking at around one hour after IFN α stimulation (25 hours after prestimulation). The mean values for pSTAT1 (N=3) at the different time points of IFN α stimulation of untreated and prestimulated Huh7.5 cells were in line with the model-predicted dynamics of ISGF3 complex formation in response to IFN α stimulation confirming the late increase of ISGF3 transcription factor complexes in untreated cells and acceleration of the formation to one hour after IFN α stimulation in cells prestimulated with 280 pM IFN α .”

It is noteworthy that untreated Huh7.5 hepatoma cells show ISGF3 activity only after several hours of IFN stimulation (Figure 4B) -in agreement with other studies using these cells- which the authors' take as strong evidence in support of their dimer-switching model. However, essentially all studies using other human and mouse cell types including liver cells (fibroblasts, hepatocytes,

macrophages) do not show a delayed ISGF3 activity. As an example, I refer to mouse primary hepatocytes, where ISGF3 activity is induced within 30 min of IFN treatment (J Leukoc Biol. 93: 377). While there are likely cell type and species differences in IFN signaling, to me these differences indicate that the inability to detect ISGF3 early on is not indicative of a generally applicable biological mechanism (i.e. GAF activity followed by ISGF3), but rather reflects a limitation of the experimental approach chosen. This, of course, raises questions also about the biological significance of the model that the authors propose to describe IFN signaling.

To clarify the issue raised by the reviewer we re-inspected our model based prediction and performed experiments with another hepatoma cell line. As visible in our model simulations Figure 4A (purple dashed line in top left panel), the model indicates that ISGF3 complex formation is already slightly increasing in the first 30 min after IFN α -stimulation. This is due to the amount of IRF9 already being present in IFN α -stimulation naïve cells. However the model suggests that only after four hours of IFN α -stimulation IRF9 is induced to a larger extent and as a consequence ISGF3 becomes the dominant transcription factor. These model based effects are now supported by our new co-immunoprecipitation experiments using an antibody that recognizes IRF9 for the immunoprecipitation and an anti-IRF9 antibody as well as an anti-pSTAT1 antibody for detection by quantitative immunoblotting (new Figure EV3C lower and upper panel). The corresponding quantifications of pSTAT1 from triplicate experiments are displayed in the new Figure 4B, right panel. To broaden the applicability of our studies, we in addition examined with our model the IFN α -induced dynamics of ISGF3 complex formation in HepG2-hNTCP cells (new Appendix Figure S7A, left panel) that have – if they are not prestimulated with IFN α – lower abundances of STAT1 and STAT2, but higher amounts of IRF9. Therefore, in these cells ISGF3 is formed more rapidly in response to IFN α stimulation, which is similar to the observations in other cell types as indicated by the reviewer. Likewise, our model analysis reveals that upon prolonged pre-stimulation with IFN α the levels of IRF9 are elevated and ISGF3 complexes form more rapidly in both Huh7.5 and HepG2-hNTCP cells. Thus, while the molecular mechanism we propose is independent of the cell type, the cell type-specific abundance of IRF9 (in relation to STAT1 and STAT2) determines the dynamics of the formation of the ISGF3 complex providing a possible explanation for the observation that in primary mouse hepatocytes ISGF3 activity can be induced within 30 min after FN α stimulation. We describe the new results in the revised results section page 9 of the manuscript as described above and discuss the implications of our insights in the revised discussion section on page 20 of the manuscript:

“This temporal order of IFN α -induced formation of pSTAT1-containing transcription factor complexes is highly dependent on the ratio between the components STAT1, STAT2 and IRF9. For example Huh7.5 cells that are not prestimulated with IFN α harbor per cell approximately 10^6 molecules STAT1, while the abundance of IRF9 is 100-fold lower. Therefore, the initial amount of ISGF3 that forms early upon IFN α stimulation is very limited and only after IRF9 is *de novo* transcribed and translated, ISGF3 can become the dominant transcription factor complex. In HepG2-hNTCP cells, which in comparison to Huh7.5 cells contain lower concentrations of STAT1 and STAT2 but higher amounts of IRF9, ISGF3 complexes can form earlier. Our analysis of primary human hepatocytes demonstrates that the ratio between STAT1, STAT2 and IRF9 varies substantially between different patient derived samples, suggesting patient-specific dynamics in both ISGF3 formation and antiviral gene response.”

This latter point is also relevant in the following. To support the dimer-switching hypotheses the authors' use bioinformatics tools to identify genes that are regulated primarily by GAS or ISRE binding sites. Subsequent transcription analyses of the respective genes are then correlated with the model-predicted occurrences of STAT1 homodimers and ISGF3 transcription factors. However, the authors make assumptions whose validity is questionable, i.e. considering only 3kB from the transcription start site as relevant for gene regulation; and the stringency of the GAS and ISRE consensus sequences used for the identification of binding sites. Accordingly, their conclusions, which are based on these assumptions, are likewise not beyond reproach. Specifically, they disagree with previous findings. IRF1 and SOCS3 are studied as the examples for primarily STAT1 homodimer-induced genes. However, SOCS3 induction does not require STAT1, because cells

that lack STAT1 retain IFN-gamma induced SOCS3 expression, contradicting the author's assignment of SOCS3 as a STAT1 homodimer-regulated gene (PNAS 98, 6674). Likewise, IFN alpha-induced IRF1 expression involves recruitment of ISGF3 to an ISRE (Nat Immunol. 15, 168). Thus, the authors' explanation for the transient expression of some genes by the transient presence of STAT1 homodimers is not in line with established knowledge of IFN-dependent gene regulation. Their explanation for the delayed but sustained expression of another set of genes by a switch to ISGF3 is similarly flawed. The authors' bioinformatics analyses identify three other genes as being regulated by ISGF3, not STAT1 homodimers. Accordingly, these genes are demonstrated to be unresponsive to IFN-gamma in the Huh cells, a strong inducer of STAT1 homodimers (Figure EV3B). I do not doubt the observations made for these cells. However, two of the three genes tested by the authors (DDX58, IFI44L) are well-established IFN gamma target genes in human hepatocytes (Gastroenterology 143, 777) and other cell types http://software.broadinstitute.org/gsea/msigdb/cards/HALLMARK_INTERFERON_GAMMA_RESPONSE. The mentioned variabilities/disagreements with published results are important, because they call into question the validity and general biological significance of key assumptions and undermine central conclusions of this work.

We agree that the bioinformatics analysis has to be taken with caution, but we would like to point out that we merely utilized it to select potential candidates as reporters to monitor target gene expression induced by pSTAT1 homodimers or ISGF3. We carefully validated the selected genes as proxies for pSTAT1 homodimer activity or ISGF3 activity by stimulating Huh7.5 cells with IFN α or with IFN γ and by quantifying gene expression. To broaden the applicability we now performed the same experiments in HepG2-hNTCP cells. While in Huh7.5 cells the dynamics of *IRF1* was very transient, in HepG2-hNTCP cells IFN α stimulation resulted in a more sustained gene expression pattern (new Appendix Figure S7B). Because the expression dynamics of *IRF1* was previously shown to be complex and also regulated by NF κ B and MAP-Kinase pathways (Yarilina et al., Nat Immunol. 2008;9:378-87), we focused our studies in HepG2-hNTCP cells on the *SOCS3* expression dynamics. The gene expression pattern of *SOCS3* in these cells indeed mirrored the pattern observed in Huh7.5 cells: It was induced by both IFN α and IFN γ and the stimulation by IFN α resulted in a transient peak in the first hour after stimulation. Importantly, the reduced expression dynamics of *SOCS3* observed in HepG2-hNTCP cells prestimulated with IFN α confirmed our model prediction in this cell type (new Appendix Figure S7C). Further, we performed new experiments with HepG2-hNTCP cells to confirm target genes that we identified in Huh7.5 cells as reliable proxies for ISGF3 activity. We stimulated HepG2-hNTCP cells with IFN α and with IFN γ . As in Huh7.5 cells all three selected genes (*DDX58*, *HERC5* and *IFI44L*) were only induced by IFN α , but not by IFN γ , demonstrating that these genes are also ISRE-dependent in HepG2-hNTCP cells (new Appendix Figure S7B). The observed impact of IFN α prestimulation on the expression dynamics of these genes was in line with our model predictions, validating the models capacity to predict target gene expression also HepG2-hNTCP cells (new Appendix Figure S7C). The model prediction and experimental validation are now described in the revised results section on page 13 of the manuscript:

“To investigate the impact of the different ratio between the STAT proteins and IRF9 in HepG2-hNTCP cells on formation of pSTAT1-containing transcription factor complexes, we simulated with our mathematical model the dynamics of pSTAT1:pSTAT1 homodimers, pSTAT1:pSTAT2 heterodimers and ISGF3 complexes in the nucleus of HepG2-hNTCP cells (Appendix Figure S7A). Unlike Huh7.5 cells, the mathematical model predicted a rapid formation of ISGF3 complexes within 30 min due to the higher amounts of IRF9 compared to STAT1 being present in untreated HepG2-hNTCP cells. Stimulation with IFN α results in a gradual increase in IRF9 production and therefore in a further increase in the formation of ISGF3 complexes two hours later. Further, the mathematical model suggested that in HepG2-hNTCP cells pSTAT1:pSTAT1 homodimers and pSTAT1:pSTAT2 heterodimers are formed with a similar dynamics as in Huh7.5 cells in response to IFN α stimulation, but the amounts of these complexes are lower. Additionally, our mathematical model simulations indicated that prestimulation of HepG2-hNTCP cells with 28 or 280 pM IFN α for 24 h reduces the formation of these complexes in a dose-dependent manner, while the formation of the ISGF3 complex is, similar to Huh7.5 cells, much accelerated.

To experimentally validate in HepG2-hNTCP cells the impact of the formation of these transcription factor complexes on the expression dynamics of the target genes selected for Huh7.5 cells, we first stimulated HepG2-hNTCP cells with 5000 IU/ml of either IFN γ or IFN α (corresponding to 1400 pM IFN α) (Appendix Figure S7B). Similar to Huh7.5 cells, stimulation with IFN γ induced the expression of *IRF1* and *SOCS3*, but not of the other genes. As in Huh7.5 cells, IFN α induced within four hours a sustained expression of *DDX58*, *HERC5* and *IFI44L* and a transient expression of *SOCS3* with a peak at around one hour after IFN α stimulation. The dynamics of *IRF1* expression deviated in HepG2-hNTCP cells with a more sustained response induced by IFN α stimulation. Due to this difference and because the expression dynamics of *IRF1* was previously shown to be complex and also regulated by the NF κ B and MAP-kinase pathways (Yarilina et al, 2008), we excluded this gene from further analysis.

We simulated with our mathematical model the dynamics of the occupancy of GAS binding sites induced in HepG2-hNTCP cells either untreated or prestimulated with 28 or 280 pM IFN α by stimulation with 1400 pM IFN α . Prestimulation with 28 pM IFN α was predicted to have little impact on the peak amplitude of the occupancy of GAS binding sites, while prestimulation with 280 pM IFN α reduced the peak amplitude of occupied GAS binding sites by an order of magnitude (Appendix Figure S7C, upper left panel). In accordance with the model prediction, the dynamics of the expression of experimentally measured expression of *SOCS3* was only slightly reduced by the prestimulation with 28 pM IFN α , but almost completely abolished by prestimulation with 280 pM IFN α (Appendix Figure S7C, upper right panel). Moreover, similar to Huh7.5 cells, the mathematical model predicted for HepG2-hNTCP cells a higher initial level of occupied ISRE binding sites at 24 hours of prestimulation and an accelerated increase of occupied ISRE binding sites upon stimulation with 1400 pM IFN α (Appendix Figure S7C, lower left panel), which was in agreement with the experimentally observed expression of the ISRE-dependent transcripts *DDX58*, *HERC5* and *IFI44L* (Appendix Figure S7C, lower right panel). These experiments demonstrated that the mathematical model can also predict the dynamics of IFN α -induced transcription factor complex formation in another liver cell line and thus confirm the broader applicability of the mathematical model.”

We further discuss these findings and explicitly mention the cell context-specific aspects in the revised discussion on page 20 of the manuscript:

“IFN α -target genes in Huh7.5 cells can be classified into three groups: Genes that are primarily regulated by GAS sites (*IRF1* and *SOCS3*), genes that are primarily regulated by ISRE sites (*DDX58*, *HERC5* and *IFI44L*) and genes that are regulated by both GAS and ISRE sites (*STAT1*, *STAT2*, *IRF9*, *IRF2*, *USP18* and *SOCS1*). The connections between transcription factor complexes and the expression of genes can be specific for a certain cell type, as methylation patterns may vary and other pathways may additionally contribute to the regulation of gene expression. In our study we observed for most of the genes with the exception of *IRF1* in Huh7.5 and HepG2-hNTCP cells similar dependencies of their expression on IFN α .”

In summary, the authors present a large and credible volume of experimental data that undoubtedly add to our knowledge about IFN signaling, and their model is able to replicate their experimental findings. Nonetheless, the progress in knowledge is convincing only insofar as experimentally derived details are concerned. On a conceptual level, and regarding biological significance at large, this work does not provide sufficient insight in order to constitute a major step ahead in the understanding of IFN signaling.

We hope that the reviewer shares our view that given the new experimental data collected on several levels and in an additional cell line, we now not only validate our conclusions, but also provide novel conceptual insights into the regulation of temporal order in IFN signaling.

Reviewer #2:

I appreciate the additional work the authors undertook to address my critique points.

The authors have done a lot and I feel overall this is a strong manuscript. Most points were addressed to satisfaction. However, one issue remains, but I feel that it can be sidestepped, so long as the relative strength of the evidence is fairly discussed.

The authors address the question about TF complexes by providing an EMSA. However, its quality is really so low that it cannot really be used to draw conclusions with confidence. It is really below the publishable threshold. Is this a key result that must be included, or can the paper not stand without it? If the authors feel that this result is critical, they need to redo the EMSAs, and provide a quantitative evaluation of replicates. In my mind it may be sufficient to discuss the model predictions and indicate that they await experimental confirmation.

We followed the advice of the reviewer, repeated the EMSA using GAS probes to detect pSTAT1 homodimers in triplicates, quantified the bands and compared the results to our model predictions. Additionally, we performed multiple experiments to improve the quality of the EMSA using ISRE probes and even contacted authors of other publications for advice. But it turned out that the detection of the ISGF3 complex binding to ISRE probes in EMSA experiments in hepatocytes is particularly challenging and the quality of the obtained results - even though being in line with our model predictions - remained unsatisfactorily. Therefore, we rather performed experiments to directly quantify the dynamics of the IFN α -induced formation of the ISGF3 complex (pSTAT1:pSTAT2:IRF9). We performed co-immunoprecipitation (co-IP) experiments by conducting immunoprecipitation experiments with antibodies recognizing IRF9 and detecting the co-immunoprecipitating pSTAT1 by immunoblotting. The quantifications of triplicate experiments performed with untreated and pretreated cells as well as the model predictions for the formation of ISGF3 complexes are shown in the new **Figure 4B, right panel** and an exemplary immunoblots are shown in Fig. EV3C. These results confirm in line with the model prediction the occurrence of ISGF3 at later time points after IFN α stimulation, whereas in pretreated cells ISGF3 is already detectable at earlier time points. Further, we performed immunoprecipitation experiments with antibodies recognizing STAT2 and detecting co-immunoprecipitating pSTAT1, which reflects pSTAT1 present in pSTAT1:pSTAT2 homodimers and ISGF3 complexes. The quantifications of triplicate experiments as well as the model-predicted sum of the pSTAT1:pSTAT2 homodimers and ISGF3 complexes are shown in the **new Figure 4B, middle panel**. These results are described in the revised results section on page 9 of the manuscript:

“To experimentally verify the model-predicted consecutive occurrence of the different transcription factor complexes, we performed electrophoretic mobility shift assays (EMSA) as previously reported (Forero et al, 2019). Experiments using a probe comprising the GAS-binding region of the IRF1 promoter (Figure EV3A, left panel) showed that in untreated Huh7.5 cells an early DNA:protein complex is induced after one hour (corresponding to 25 hours after mock prestimulation) of stimulation with 1400 pM IFN α . This DNA:protein complex is absent at four and six hours post IFN α stimulation of Huh7.5 cells (corresponding to 28 and 30 hours after mock prestimulation, respectively). As shown in Figure EV3A, right panel, incubation of the lysate-DNA mixture with an antibody recognizing STAT1 led to a supershift, which was absent upon addition of antibodies detecting STAT2 or IRF9, confirming the specificity of the detected complex as pSTAT1 homodimer. In accordance with our assumption in the mathematical model, no major binding of the pSTAT1:pSTAT2 heterodimer to the GAS region was observed. On the contrary to untreated Huh7.5 cells, formation of pSTAT1 homodimeric complexes induced by stimulation with 1400 pM IFN α is much reduced in cells prestimulated with 280 pM IFN α for 24 h. To quantitatively compare the obtained results with our model predictions, we predicted the dynamics of occupied GAS binding sites induced by 1400 pM IFN α in untreated Huh7.5 cells and in Huh7.5 cells prestimulated with 280 pM IFN α for 24 h (Figure 4B, left panel). The corresponding 68%-confidence intervals were computed as proposed in Kreutz et al. (2012). The simulation showed a steep increase of occupied GAS binding sites within the first hour after stimulation, which was suppressed upon prestimulation of cells with IFN α . As shown in Figure 4B, left panel, the mean values of pSTAT1 homodimeric complexes detected by EMSA (N=3) were in agreement with the model prediction and experimentally confirmed a rapid but transient formation of pSTAT1 homodimers in response to stimulation with IFN α , which was much reduced upon IFN α prestimulation, validating the model-predicted early occurrence of pSTAT1 homodimers in response to IFN α stimulation. To investigate the IFN α -induced dynamics of the formation of the other STAT1-containing transcription factor complexes, we performed co-immunoprecipitation (co-IP) experiments. We stimulated non-prestimulated Huh7.5 cells or Huh7.5 cells prestimulated with 280 pM IFN α for 24 h for one to six hours (corresponding to 25 to 30 hours after prestimulation) with 1400 pM IFN α . The cellular lysates were used

for immunoprecipitation experiments using antibodies recognizing STAT2 and co-immunoprecipitated pSTAT1 was detected by quantitative immunoblotting (Figure EV3B). With these co-IP experiments the dynamics of the sum of IFN α -induced formation of pSTAT1:pSTAT2 heterodimers and pSTAT1:pSTAT2:IRF9 trimers (ISGF3) was detected. In untreated Huh 7.5 cells the signal for co-immunoprecipitating pSTAT1 was maximal after one hour of stimulation with 1400 pM IFN α and slowly decreased thereafter but not reaching baseline after six hours of stimulation (25 to 28 hours after mock prestimulation). Upon prestimulation of Huh7.5 cells with 280 pM IFN α for 24 h, higher levels of total STAT2 were observed, while the overall signal of co-immunoprecipitated pSTAT1 was lower. Distinctively, it was already detectable after 24 hours of prestimulation and increased to a much lower extent by stimulation with 1400 pM IFN α compared to the amount detected in untreated cells. To compare the experimental results obtained by the quantification of co-immunoprecipitating pSTAT1 (N=3) to the predictions by our mathematical model, we calculated the dynamics of the sum of pSTAT1:pSTAT2 heterodimers and the pSTAT1:pSTAT2:IRF9 trimers (ISGF3) induced by 1400 pM IFN α in untreated Huh7.5 cells and Huh7.5 cells prestimulated with 280 pM IFN α for 24 h and computed 68%-confidence intervals. As shown in Figure 4B, middle panel, these results experimentally confirmed the broad peak in the model-predicted dynamics of the sum of pSTAT1:pSTAT2 heterodimers and ISGF3, which was reduced to around one third upon prestimulation of the cells.

Additionally, to quantify in untreated and prestimulated Huh7.5 cells the dynamics of IFN α -induced ISGF3 complex formation, co-IP experiments were performed using antibodies recognizing IRF9 and co-immunoprecipitated pSTAT1 was detected by quantitative immunoblotting (Figure EV3C). The signal for IRF9-precipitated pSTAT1 increased upon stimulation of untreated Huh7.5 cells with 1400 pM IFN α with a peak at four hours post IFN α treatment (28 hours after mock prestimulation). Upon prestimulation of Huh7.5 cells with 280 pM IFN α for 24 h, IRF9 levels were strongly increased and co-immunoprecipitated pSTAT1 was now already peaking at around one hour after IFN α stimulation (25 hours after prestimulation). The mean values for pSTAT1 (N=3) at the different time points of IFN α stimulation of untreated and prestimulated Huh7.5 cells were in line with the model-predicted dynamics of ISGF3 complex formation in response to IFN α stimulation confirming the late increase of ISGF3 transcription factor complexes in untreated cells and acceleration of the formation to one hour after IFN α stimulation in cells prestimulated with 280 pM IFN α .”

Also, the nomenclature of occupied ISRE/GAS sites is very cumbersome and confusing. The ISRE or GAS sequences either exist or do not. If ISGF3 binds the ISRE, then this is a ISGF3-ISRE complex. If the GAS site is occupied it is a GAF-GAS complex. Are the authors suggesting that there may be GAF-ISRE complexes? I suggest that the description is revised to use conventional nomenclature of sites and complexes.

We are grateful to the reviewer for pointing this out. We intended to convey that some genes harbor both GAS and ISRE sites in their promoters. Of course, these sites can only be bound by their respective transcription factor complexes (GAS-GAS, ISGF3-ISRE). For clarification, we updated the model scheme (Figure 2) and now indicate that the transcripts of IRF9, STAT1, STAT2, IRF2, SOCS1 and USP18 are induced by both occupied (pSTAT1 homodimers bound) GAS bindings sites and occupied (ISGF3 bound) ISRE binding sites (OccGASbs+OccISREbs). We clarified this nomenclature in the results section on page 6 of the manuscript as follows:

“The promoters of the genes encoding the positive feedback proteins IRF9, STAT1 and STAT2 as well as the negative feedback proteins USP18, SOCS1 and IRF2 harbor gamma interferon activated sites (GAS) in combination with interferon-stimulated response elements (ISRE). Since pSTAT1:pSTAT2 heterodimers and ISGF3 bind to these combined GAS and ISRE sites, both, nuclear pSTAT1pSTAT2n and ISGF3n contribute to the formation of occupied GAS and ISRE binding sites (OccGASbs+OccISREbs) in the promoters of these genes.”

11th May 2020

Manuscript Number: MSB-19-8955RR

Title: Disentangling molecular mechanisms regulating sensitization of interferon alpha signal transduction

Thank you for sending us your revised manuscript .

We have now heard back from reviewer #1 who was asked to evaluate your study. As you will see below, the reviewer appreciates the addition of the new data. However, they feel that the new data raise some questions about the reported conclusions. We have consulted with reviewer #2 regarding these remaining issues. Reviewer #2 mentioned: "The quantity of data is impressive. When so much data is presented, not all of it will neatly support the overall conclusions. Any Mathematical model, or even word model or word interpretation of data, is necessarily a simplification of the true complexity of the biological network that control phenotypes. I feel that at this point the paper provides a lot of honest experimental data and a fairly rigorous way to try to parameterize a model to the data. That is useful to the field. Discrepancies do occur, and I would agree that pointing those discrepancies out would be useful, as each of those would then be the starting point for a future study of refining the model."

Taken together, we do not think that the comments of reviewer #1 preclude the publication of the paper. We would only ask you to make sure that a cautious interpretation of the related findings is presented in the text, by including some (minor) text changes in a final round of minor revision.

REFeree REPORTS

Reviewer #1:

The authors have added additional data to this voluminous work yet its novelty and biological significance remains limited. In fact, some of the novel data raise new troubling questions, such as the gene expression data of IRF1, one of the two genes previously used to link the transient formation of GAF to the transient expression of genes supposedly regulated by GAF. This, however, does not hold true for the additional cell line tested, where this gene shows sustained expression. The authors drop this gene from the short list that supports their model, because IRF1 expression dynamics was previously shown to be "complex". I am sure it is, but this unexpected

behaviour questions the general significance of the findings and underscores that key results in this work are founded on weak evidence. Another example is the conjecture that precipitation of STAT2 and detection of co-precipitating phosphor-STAT1 reflects activated heterodimers, pSTAT:pSTAT2 (page 10, Figure EV3B). However, pSTAT1 binds strongly also to latent STAT2 (Plos Biol 14: e2000117), such that we cannot know the composition of those heterodimers. Given that "non-canonical" complexes are proposed to contribute to IFN signalling, this is a potentially serious limitation. The authors now include EMSAs that show a transient presence of GAF in untreated cells but not pre-treated cells, which addresses an important point. While this is welcome, it is still not fully convincing, because (i) the GAF signal seems to be rather faint (unbound radioactivity or an IFN γ control are not shown for comparison), which (ii) raises questions as to the relevance of GAF for IFN-induced gene transcription. In addition, the concurrent ISGF3 signal cannot be detected, and is detected indirectly by the mentioned precipitation assays, the limitations of which I have mentioned. The authors' statement regarding this part of their work "But it turned out that ... the quality of the obtained (ISGF3-EMSA) results - even though being in line with our model predictions - remained unsatisfactorily" is nonsensical if not outright disconcerting. The authors may well have worked out a relevant point, but the current data are simply not convincing of general mechanisms, it is rather a sort of "cherry-picking" approach in which data that fit expectations are included and those that don't are deposited. Finally, the authors' additional experiments with Hep cells point to IRF9 levels as the decisive difference in regards ISGF3 assembly and hence IFN-mediated gene regulation. Yet, the abstract states that "the abundance of STAT2 predicts the patient-specific IFN α signal response".

Point-by-point response

Reviewer #1:

The authors have added additional data to this voluminous work yet its novelty and biological significance remains limited. In fact, some of the novel data raise new troubling questions, such as the gene expression data of *IRF1*, one of the two genes previously used to link the transient formation of GAF to the transient expression of genes supposedly regulated by GAF. This, however, does not hold true for the additional cell line tested, where this gene shows sustained expression. The authors drop this gene from the short list that supports their model, because *IRF1* expression dynamics was previously shown to be "complex". I am sure it is, but this unexpected behaviour questions the general significance of the findings and underscores that key results in this work are founded on weak evidence.

A closer view of the results displayed in Appendix Figure S7B show that - as in Huh7.5 cells - stimulation of HepG2-hNTCP cells with IFN α results within one hour in a 50 fold induction of *IRF1* mRNA expression. These results confirm *IRF1* as an immediate early gene induced by IFN α . As in Huh7.5 cells, the IFN α -induced expression of *IRF1* in HepG2-hNTCP cells is similar to *SOCS3*, but different from *DDX58*, *HERC5* and *IFI44L*, which are only upregulated after 3 to 5 hours of IFN α stimulation. However, in HepG2-hNTCP cells the IFN α -induced expression of *IRF1* is more sustained than in Huh7.5 cells, indicating cell type-specific modulation of the down regulation of *IRF1* expression, possibly due to cell-context specific methylation patterns or the impact of other transcription factors. To clarify the issue raised by the reviewer we improved the description of these observations in the results section of the revised manuscript on page 14:

"Similar to Huh7.5 cells, stimulation with IFN γ induced in HepG2-hNTCP cells the expression of *IRF1* and *SOCS3*, but not of *DDX58*, *HERC5* and *IFI44L*. In line with the mRNA expression dynamics observed in Huh7.5 cells, IFN α stimulation of HepG2-hNTCP cells resulted after four hours in the induction of the expression of *DDX58*, *HERC5* and *IFI44L* and in an immediate increase in the expression of *SOCS3* with a peak at around one hour after IFN α stimulation. For *IRF1* an approximately 50-fold increase of mRNA expression was observed within 1 hour after IFN α stimulation, which was in line with the mRNA expression dynamics in Huh7.5 cells and confirmed *IRF1* as an immediate early gene of IFN α -induced responses. However, in HepG2-hNTCP cells the IFN α -induced expression of *IRF1* was more sustained suggesting that the down regulation of the *IRF1* expression in HepG2-hNTCP is potentially modulated by the cell-context specific activation of other transcription factors. This is in line with previous reports that the expression of *IRF1* can also be regulated by the NF κ B and MAP-kinase pathways (Yarilina et al, 2008) and therefore we did not include *IRF1* in our further analyses."

Further, to improve clarity we rephrased the discussion of the cell type-specific expression dynamics of *IRF1* in the discussion section of the revised manuscript on page 20:

"The connections between transcription factor complexes and mRNA expression of genes can be influenced by cell context-specific alterations in methylation patterns (Merkle et al, 2016) and differences in the activation of signal transduction pathways leading to an impact of additional transcription factors. In our study we observed for most of the analyzed genes in Huh7.5 and HepG2-hNTCP cells a highly comparable expression dynamics upon stimulation with IFN α . *IRF1* was induced in both cell types as an immediate early gene within 1 hour with an approximately 50-fold increase, but showed in HepG2-hNTCP cells a more sustained behavior. This prolonged expression indicates cell type-specific modulation of the *IRF1* mRNA expression, which is potentially due to an impact of the NF κ B and MAP-kinase pathways as previously reported (Yarilina et al, 2008) or differences in epigenetic modifications present in HepG2-hNTCP cells versus Huh7.5 cells. In line with this hypothesis, we previously reported that cell type-specific differences in the expression dynamics of *SOCS3* can be explained by differences in promoter binding elements and epigenetic modifications (Merkle et al, 2016)."

Another example is the conjecture that precipitation of STAT2 and detection of co-precipitating phosphor-STAT1 reflects activated heterodimers, pSTAT:pSTAT2 (page 10, Figure EV3B). However, pSTAT1 binds strongly also to latent STAT2 (Plos Biol 14: e2000117), such that we cannot know the composition of those heterodimers. Given that "non-canonical" complexes are proposed to contribute to IFN signalling, this is a potentially serious limitation.

To clarify the point raised by the reviewer we rephrased the paragraph in the discussion section of the revised manuscript on page 21:

"Specifically, we employed co-immunoprecipitation experiments by performing STAT2 immunoprecipitations and quantifying co-immunoprecipitating pSTAT1 to verify the model-predicted dynamics of the sum of IFN α -induced formation of pSTAT1:pSTAT2 heterodimers and pSTAT1:pSTAT2:IRF9 trimers. We cannot exclude that non-canonical complexes such as pSTAT1:STAT2 (Ho et al, 2016) are also detected in these experiments. However, Ho et al showed that these complexes remain in the cytoplasm and are not relevant for type 1 interferon signaling. Further, since our experiments show that the IFN α -induced dynamics of STAT1 and STAT2 phosphorylation is very similar and this is also reflected by the trajectories of the mathematical model (Figure 3A), we assume that even if a hypothetical pSTAT1:STAT2 complex would form, it would have the same dynamics as the model-simulated pSTAT1:pSTAT2 complex."

The authors now include EMSAs that show a transient presence of GAF in untreated cells but not pre-treated cells, which addresses an important point. While this is welcome, it is still not fully convincing, because (i) the GAF signal seems to be rather faint (unbound radioactivity or an IFN γ control are not shown for comparison), which (ii) raises questions as to the relevance of GAF for IFN-induced gene transcription. In addition, the concurrent ISGF3 signal cannot be detected, and is detected indirectly by the mentioned precipitation assays, the limitations of which I have mentioned. The authors' statement regarding this part of their work "But it turned out that ... the quality of the obtained (ISGF3-EMSA) results - even though being in line with our model predictions - remained unsatisfactorily" is nonsensical if not outright disconcerting. The authors may well have worked out a relevant point, but the current data are simply not convincing of general mechanisms, it is rather a sort of "cherry-picking" approach in which data that fit expectations are included and those that don't are deposed.

As shown below in the images of the three independent EMSA experiments using GAS binding probes including the unbound radioactivity, in each experiment a strong band of a DNA:protein complex (GASprobe:pSTAT1:pSTAT1) was reproducibly detected already 1 hour after IFN α stimulation and declined thereafter. In Huh7.5 cells that had been prestimulated with 280 pM IFN α the intensity of the band was much reduced but also readily detectable. In each of the experiments the band could be reliably quantified yielding average values with small confidence intervals (shown in Figure 4B, left panel), which is essential to reliably define dynamic behavior and apply mathematical modelling approaches.

Huh7.5 (nuclear lysates, EMSA with GAS of IRF1)

As shown in previous versions of the manuscript, we also explored the use of EMSA to examine the IFN α induced dynamics of the formation of ISGF3 complexes and performed ISRE-EMSA:

Huh7.5 (nuclear lysates, EMSA with ISRE of ISG15)

These experiments show that the ISRE-binding ISGF3 trimeric complexes become detectable at 4 to 6 hours post IFN α stimulation and thus confirm our model based-hypothesis that ISGF3 form later than pSTAT1:pSTAT1 homodimers and are accelerated and reduced upon prestimulation with 280 pM IFN α .

However, despite many repetitions we felt that this assay is not robust enough to warrant reliable quantification of detailed dynamic behavior and noted that most of the published studies using ISRE-EMSA have been performed with murine cells. As our studies demonstrate the importance of the abundance of pathway components in determining the dynamics of complex formation, it is conceivable that in the human cells we are studying lower amounts of ISGF3 form as compared to many murine cells. Given these challenges we employed co-immunoprecipitation studies that yielded highly robust data with small confidence intervals confirming our initial results on the dynamics of complex formation obtained by EMSA experiments and display these results in the manuscript.

To clarify the issue raised by the reviewer, we now discuss the importance of adequate experimental methods with favorable signal-to-noise ratio to define dynamic behavior in the discussion section of the revised manuscript on page 21:

“For mathematical modeling approaches it is of particular importance to combine model-guided experimental design (Kreutz & Timmer, 2009) with careful selection of experimental methods yielding informative data with favorable signal-to-noise ratio to accurately define dynamic behavior. For the detection of the dynamics of the formation of pSTAT1 homodimers in complex with GAS binding regions EMSA was most reliable as demonstrated by the small confidence intervals and revealed in several independent experiments that maximal binding activity is observed within one hour of IFN α stimulation. However, to examine the dynamics of the formation of the large trimeric ISGF3 complex and the heterodimeric complexes, co-immunoprecipitation experiments were most robust yielding reproducible quantifications in independent experiments.”

Finally, the authors' additional experiments with Hep cells point to IRF9 levels as the decisive difference in regards ISGF3 assembly and hence IFN-mediated gene regulation. Yet, the abstract states that "the abundance of STAT2 predicts the patient-specific IFN α signal response".

Indeed, IRF9 levels determine how fast ISGF3 complexes are formed and thus the speed of the response. For example the model predicts that ISGF3 complexes are formed more rapidly in untreated HepG2-hNTCP cells (Appendix Figure S7A) than in untreated Huh7.5 cells (Figure 4A), since already prior to IFN α stimulation IRF9 is more abundant in HepG2-hNTCP cells. However, the maximal amount of ISGF3 formed is not only determined by the abundance of IRF9 but also by the availability of pSTAT1 and in particular of pSTAT2 that is present in lower amounts and therefore limiting. In fact, the determined absolute antiviral response to 608 pM Roferon is comparable between untreated Huh7.5 and HepG2-hNTCP cells (Figure 6D). Thus, IRF9 and STAT2 control different aspects of the signal transduction network. To further clarify this point we revised the manuscript on page 3:

“Model simulations and experimental evidence revealed that not only USP18 but also SOCS1 are required for pathway desensitization, while induction of IRF9 and STAT2 contributes to pathway hypersensitization and the basal amount of IRF9 controls the dynamics of the ISGF3 transcription factor complex formation.”

Additionally, we now differentiate between the different control mechanisms in the discussion section of the revised manuscript on page 23:

“The experimental data and analyses by the mathematical model demonstrated that control of the IFN α -induced signal transduction is multifactorial and that the different components determine distinct aspects of the dynamics of pathway activation. The amount of IRF9 controls how fast ISGF3 complexes are formed and thereby the speed of the response, whereas the abundance of STAT2 determines how many ISGF3 complexes can be formed and herewith the extent of an antiviral response. As a consequence USP18 determines pathway sensitization and STAT2 is a predictor of the patient-specific antiviral response.”

Manuscript number: MSB-19-8955RRR, Disentangling molecular mechanisms regulating sensitization of interferon alpha signal transduction

Thank you again for sending us your revised manuscript. We are now satisfied with the modifications made and I am pleased to inform you that your paper has been accepted for publication.

Corresponding Author Name: Ursula Klingmüller
Journal Submitted to: Molecular Systems Biology
Manuscript Number: MSB-19-8955